# Generalizing Beyond Suboptimality:
# Offline Reinforcement Learning Learns Effective Scheduling through Random Solutions

## Abstract

Online reinforcement learning (RL) approaches have demonstrated strong performance on Job Shop Scheduling (JSP) and Flexible JSP (FJSP) problems by learning scheduling policies through direct interaction with simulated environments. However, these methods often require extensive training interactions, limiting their sample efficiency and practical applicability.

Motivated by this challenge, we introduce **Conservative Discrete Quantile Actor-Critic (CDQAC)**, an offline RL algorithm that learns effective scheduling policies directly from static, suboptimal datasets. CDQAC couples a quantile-based critic with delayed policy updates to estimate the return distribution of machine–operation pairs. Extensive experiments on JSP and FJSP benchmarks demonstrate that CDQAC matches or outperforms the data-generating heuristics, outperforms recent offline and online RL baselines for JSP and FJSP, and is highly sample efficient, requiring only 1 to 5% of the original dataset to learn high-quality policies. Our analysis suggests that, for JSP and FJSP, offline RL performance depends more on state-action coverage than on the quality of individual trajectories. FJSP and JSP couples a dense reward aligned with the makespan objective with equal-length trajectories across heuristics, enabling effective learning from a broad range of behaviors. Consistent with this observation, datasets generated by a simple random heuristic with broader coverage let it outperform policies trained on datasets produced by stronger heuristics such as Genetic Algorithms.

## 1 Introduction

The Job Shop Scheduling Problem (JSP) and Flexible Job Shop Scheduling Problem (FJSP) are fundamental challenges in manufacturing and industrial operations (Bhatt & Chauhan, 2015), where the goal is to optimally schedule *jobs* on available *machines* to minimize objectives such as total completion time (makespan). Exact methods such as Constraint Programming (CP) (Da Col & Teppan, 2022) and Mathematical Programming (Fan & Su, 2022) guarantee optimality but face scalability issues for large-sized instances. Therefore, in practice, heuristic methods such as Genetic Algorithms (GA) (Bhatt & Chauhan, 2015) and Priority Dispatching Rules (PDRs) (Veronique Sels & Vanhoucke, 2012) are often preferred, as they can find acceptable solutions in reasonable time.

Deep reinforcement learning (RL) has emerged as a promising approach for learning dispatching rules; methods like Learning-to-Dispatch (L2D) (Zhang et al., 2020) can outperform standard dispatching rules, infer high quality solutions orders of magnitude faster than exact solvers, and generalize to larger instances. However, because they train from scratch via trial-and-error, these online methods suffer from sample inefficiency, often requiring a vast amount of simulator interactions to converge. Conversely, heuristics such as PDR and Genetic Algorithms (GA) are widely used and can easily generate substantial, albeit inherently suboptimal, training data. Offline RL offers a paradigm to leverage these static datasets by estimating value rather than merely imitating actions, enabling it to learn a new policy not present in the training dataset. Remmerden et al.

(2025) recently introduced Offline-LD as the first offline RL method for JSP, where it outperformed L2D using only 100 instances. Crucially, the reliance of Offline-LD on computationally expensive CP solutions limits its scalability, leaving the potential of learning from vast amounts of suboptimal heuristic data underutilized.

We argue that JSP and FJSP are particularly well suited to learning from suboptimal—even random—data. Unlike prior offline RL applications such as robotics (Levine et al., 2020; Figueiredo Prudencio et al., 2024) where the reward is sparse or an approximation of the true objective, FJSP and JSP provide a dense reward signal that is aligned exactly with the optimization objective through partial makespan increases (Smit et al., 2025), allowing every transition to carry a calibrated value signal. Furthermore, constructive scheduling policies for FJSP/JSP, whether expert, heuristic, or random, explore from the same states and always generate trajectories of identical length, equal to the number of operations to be scheduled. As a result, JSP/FJSP avoid the trajectory-length bias that Li et al. (2023) shows can make offline RL favor expert data. Together, suggest that offline RL performance in FJSP and JSP depends more on *state–action coverage* (Schweighofer et al., 2022) than on *data quality*.

With this motivation, we propose **Conservative Discrete Quantile Actor-Critic** (CDQAC), a novel offline RL method for JSP and FJSP, which learns effective scheduling policies from **suboptimal examples** generated by a wide range of heuristics. CDQAC learns an approximated representation of the value of each action from which it can generalize a new policy that can outperform the heuristic that generated the data. CDQAC achieves this through a quantile-based critic, with a new dueling architecture. This critic provides value estimates that guide the actor, while a delayed policy update prevents the propagation of early noisy critic predictions, ensuring stable joint learning of the policy and value function.

Our contributions are as follows: (1) We introduce CDQAC, a offline RL method that demonstrates strong performance on both JSP and FJSP, by solely training on suboptimal examples. CDQAC outperforms the tested online RL methods while using only suboptimally generated random data, removing the need for both a training simulator and expert demonstrations. (2) CDQAC integrates a quantile critic with a dueling architecture that uses *separated* inputs for the value and advantage streams — tailored to JSP and FJSP's variable-size action spaces — and a delayed policy update that prevents bootstrapped error propagation in offline discrete RL. (3) We show that random data can be the *strongest* training source for offline RL on FJSP and JSP. This follows from two structural properties of these problems: a dense, objective-aligned reward and equal-length trajectories across all heuristics. Under these conditions, state–action coverage rather than data quality significantly correlates with performance, and a random policy maximizes per-state action diversity by construction. (4) CDQAC is highly sample-efficient, achieving competitive performance with 10–25 training instances, which is typically 1–5% of the data used by online RL baselines.

## 2 Preliminaries

**JSP & FJSP.** We formulate the Job Shop Scheduling (JSP) and Flexible Job Shop Scheduling Problem (FJSP) as follows. Given a set of $n$ jobs, represented as $\mathcal{J}$, and a set of $m$ machines, represented as $\mathcal{M}$, each job $J_i \in \mathcal{J}$ has $n_i$ operations. These operations $\mathcal{O}_i = \{O_{i,1}, O_{i,2}, ..., O_{i,n_i}\}$ must be processed in order, forming a precedence constraint. In JSP, each operation $O_{i,j}$ can only be processed by a single machine, whereas in FJSP, $O_{i,j}$ can be processed on any machine in its set of compatible available machines $\mathcal{M}_{i,j} \subseteq \mathcal{M}$. Each machine $M_k \in \mathcal{M}_{i,j}$ has a specific processing time for an operation $O_{i,j}$ denoted as $p_{i,j}^k$, where $p_{i,j}^k > 0$. The objective is to minimize the makespan, defined as the completion of the last operation $C_{\max} = \max_{O_{i,j} \in \mathcal{O}} C(O_{i,j})$, where $C(O_{i,j})$ represents the completion time of operation $O_{i,j}$.

**Offline Reinforcement Learning.** We formalize FJSP and JSP as a Markov Decision Process (MDP) denoted as $\mathbf{M}_{\mathrm{MDP}} = \langle \mathcal{S}, \mathcal{A}(s_t), P, R, \gamma \rangle$. A state $s_t \in \mathcal{S}$ represents the progress of the current schedule in the timestep $t$, and includes all operations $O_{i,j} \in \mathcal{O}_t$ that are available to be scheduled on machines $M_k \in \mathcal{M}_t$, whereby $\mathcal{M}_t$ only contains machines that are free at timestep $t$. The action space $a_t \in \mathcal{A}(s_t)$ corresponds to all available machine-operation pairs $(O_{i,j}, M_k)$ at $t$. $P$ is the transition function and determines the next state $s_{t+1}$ on the selected machine-operation pair $(O_{i,j}, M_k)$, whereby unavailable pairs, due to $M_k$ being selected, being removed and new available pairs added. The reward $r_t$ is the negative increase in the (partial) makespan resulting from action $a_t$: $r_t = \max_{O_{i,j} \in \mathcal{O}} C(O_{i,j}, s_t) - \max_{O_{i,j} \in \mathcal{O}} C(O_{i,j}, s_{t+1})$, $\gamma$ is the discount

factor that determines the importance of future rewards. We set $\gamma = 1$. In offline RL, a policy $\pi(a|s)$ is learned through a static dataset $\mathcal{D} = \{(s, a, r(s, a), s')_i\}$, where $s'$ is the next state. $\mathcal{D}$ is generated through one or more behavioral policies $\pi_\beta$.

# 3 Related Work

**Learning-based methods for FJSP/JSP** Deep reinforcement learning (DRL) has emerged as a competitive alternative to traditional dispatching rules and exact solvers for JSP and FJSP, offering near-real-time inference, strong cross-instance generalization, and consistently outperforming standard priority dispatching rules (Smit et al., 2025). The dominant paradigm in this line of work is online DRL: a constructive policy is trained from scratch through repeated interaction with a simulated scheduling environment. L2D (Zhang et al., 2020) established this paradigm for JSP, formulating scheduling on a disjunctive graph and training a GNN policy with PPO to surpass classical heuristics, with subsequent extensions to multi-agent settings (Park et al., 2021). The constructive online setup was later adapted to FJSP, most notably by FJSP-DRL (Song et al., 2023), which uses a heterogeneous graph encoder, and DANIEL (Wang et al., 2023), which jointly reasons over operations and machines through a dual attention network. Beyond purely constructive policies, complementary directions augment the online loop with curriculum strategies (Iklassov et al., 2023), behavior cloning losses against CP demonstrations (Tassel et al., 2023; Jesus et al., 2026), additional changes to the network architecture (Zhao et al., 2025), RL-guided improvement heuristics (Zhang et al., 2024a;b), and self-supervised constructive learning (Corsini et al., 2024; Pirnay & Grimm, 2024). All of these methods have shown their competitive performance as the traditional heuristics on JSP and FJSP benchmarks. However, they still require a simulated environment during training and therefore inherit the sample inefficiency that characterizes online RL. Building on the demonstrated effectiveness of online DRL for JSP and FJSP, this paper investigates whether an offline approach can attain comparable performance while training purely on static, suboptimal datasets — removing the need for any environment interaction. The only prior offline RL method in this space, Offline-LD (Remmerden et al., 2025), partially addresses this question but is restricted to JSP and requires (near-)optimal CP solutions as training data, a limitation we revisit in the next subsection.

**Offline Reinforcement Learning.** Most offline RL methods target continuous action spaces (An et al., 2021; Kostrikov et al., 2021; Jackson et al., 2025), while discrete-action methods are typically benchmarked on settings with fixed state and action sizes (Kumar et al., 2023). Both assumptions are incompatible with FJSP and JSP, whose action spaces are discrete, variable-size, and instance-dependent. This rules out continuous-action methods such as TD3+BC and SPOT, as well as transformer-based sequence models (Chen et al., 2021; Janner et al., 2021), which require fixed state/action dimensions. Among methods compatible with our setting, Conservative Q-Learning (CQL) (Kumar et al., 2020) prevents overestimation of out-of-distribution actions through value regularization and has been applied to large discrete-action problems (Kumar et al., 2023). Distributional offline RL was explored by CODAC (Ma et al., 2021), which combines quantile-based distributional RL with conservative penalties for risk-sensitive continuous control. CDQAC differs from CODAC in three respects: (1) we target discrete, variable-size scheduling actions rather than continuous control; (2) we apply CQL conservatism on the scalar mean of the quantile distribution rather than penalizing the full return distribution, since our objective is risk-neutral makespan minimization; and (3) we introduce a separated-input dueling architecture and delayed policy updates tailored to JSP and FJSP. A consistent finding across these benchmarks is a data-quality ordering: noisy-expert datasets dominate, while uniformly random data is the weakest training source (Schweighofer et al., 2022; Kumar et al., 2022). This ordering reflects the structural properties of the domains in which offline RL is usually studied—robotics and Atari—where rewards are sparse or proxy and random policies produce short, failure-terminated trajectories. FJSP and JSP are a different setting: it provides a dense, objective-aligned reward and equal-length trajectories regardless of the generating heuristic. Under these conditions, state–action coverage rather than data quality becomes the more important factor, which makes diverse and even random data a natural strength rather than a liability. We develop this argument in detail in Sect. 5.2 and confirm it empirically. Offline RL has only recently been applied to JSP. Offline-LD (Remmerden et al., 2025) first demonstrated offline RL by using (near-)optimal constraint programming solutions, but is limited to JSP and requires expert-quality data. CDQAC extends offline RL to both JSP and FJSP and, unlike prior work, learns effective policies from

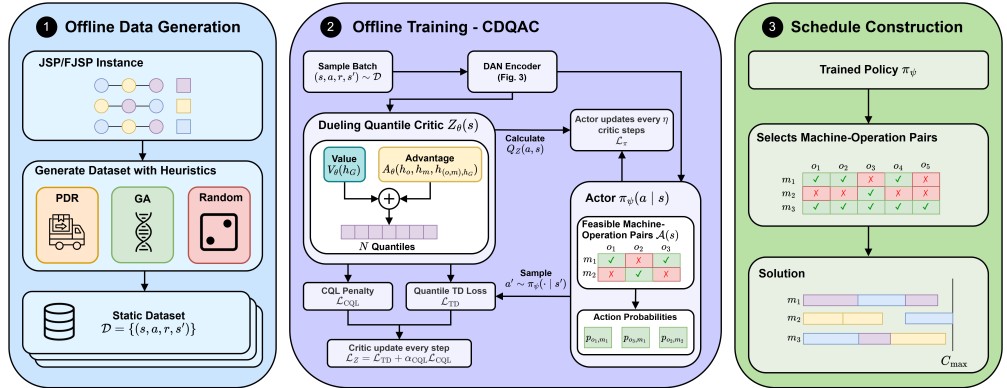

Figure 1: **(1)**: A static dataset $\mathcal{D}$ is generated for a JSP/FJSP instance using *any* heuristic—priority dispatching rules (PDR), genetic algorithms (GA), or even a random heuristic. **(2)**: From this offline data, CDQAC trains a dueling quantile critic $Z_\theta$ under a quantile TD loss with a conservative Q-learning (CQL) penalty, together with a delayed actor $\pi_\psi$ over feasible machine–operation pairs. **(3)**: At inference, the learned policy $\pi_\psi$ constructs a schedule that minimizes the makespan $C_{\max}$.

purely suboptimal data — including data generated by a random policy. This distinguishes our setting from imitation learning, or behavioral cloning, which imitates near-optimal demonstrations and cannot improve beyond the data generator (Luo et al., 2023; Drakulic et al., 2023).

## 4 Conservative Discrete Quantile Actor-Critic for JSP and FJSP

Our goal is to learn a scheduling policy $\pi_\psi$ from a static dataset $\mathcal{D}$ that surpasses the behavioral policies $\pi_\beta$ that generated it. $\pi_\beta$ may be any (possibly non-Markovian) heuristic, such as PDRs, genetic algorithms, or random schedulers, as shown in Fig 1. To outperform $\pi_\beta$, the learner must estimate accurate state–action values $Q_\theta(s, a)$ in $\mathcal{D}$ and "stitch" high-value segments into a better policy. This differs from Behavioral Cloning, which learns to imitate the actions of $\pi_\beta$. Because $\pi_\psi$ is updated solely via $Q_\theta$, we employ a critic that must (1) model the *return distribution* for state–action pairs observed in $\mathcal{D}$ and (2) remain *conservative* on out-of-distribution (OOD) actions to avoid overestimation under distributional shift, while still enabling improvement beyond the data. For this purpose, we propose Conservative Discrete Quantile Actor-Critic (CDQAC), an offline RL approach for JSP and FJSP. CDQAC integrates a **quantile critic** with a delayed policy update, enabling the learning of a scheduling policy from a dataset $\mathcal{D}$ composed of suboptimal examples, while still discovering policies that outperform those contained in $\mathcal{D}$.

**Quantile Critic.** Although both JSP and FJSP are deterministic, a state–action pair can induce a non-degenerate return distribution under a stochastic continuation policy. Assigning operation $O_{i,j}$ to machine $M_k$ changes which machine–operation pairs subsequently become feasible, and different downstream decisions can therefore produce substantially different makespans. We represent this action-dependent return variation through the random return $Z^\pi(s, a) = \sum_{t=0}^\infty \gamma^t r(s_t, a_t)$ directly, rather than its expectation $Q^\pi(s, a) = \mathbb{E}[Z^\pi(s, a)]$, which has been shown to yield more accurate value estimates than standard Q-learning (Bellemare et al., 2017; Dabney et al., 2018). Although approximating the return as $Z^\pi(s, a)$ is not directly required, we argue that this approach results in more accurate learning of the return distribution and in return learning a more optimal scheduling policy $\pi_\psi$, which is trained through bootstrapped targets of the critic.

To mitigate value overestimation, we parameterize the critic with two heads $Z_{\theta_i}$, $i \in \{1, 2\}$, and use $Z_\theta = \min(Z_{\theta_1}, Z_{\theta_2})$ in target computation (Christodoulou, 2019; Zhou et al., 2024). Each head approximates the return as a uniform mixture of $N$ Dirac deltas at fixed fractions $\tau_n = (2n - 1)/(2N)$, $n \in [1, \dots, N]$:

$$Z_{\theta_i}(s, a) = \frac{1}{N} \sum_{j=1}^N \delta\left(\theta_i^j(s, a)\right), \tag{1}$$

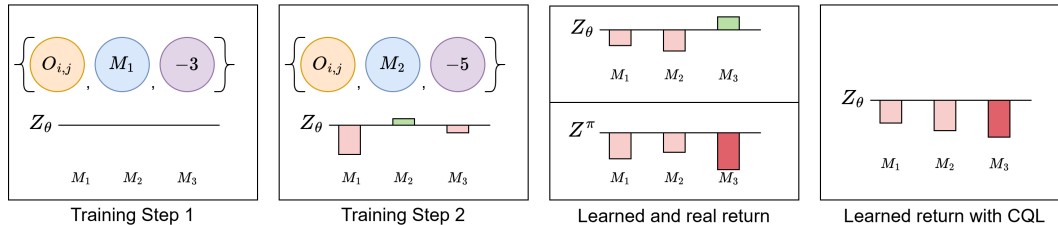

Figure 2: Illustrative example of overestimating OOD actions. In training steps 1 and 2 examples are shown of negative outcomes of pairing operation $O_{i,j}$ with either machine $M_1$, with a reward of $-3$, or $M_2$, with a reward of $-5$, learning that $M_3$ results in the best outcome, since the combination $(O_{i,j}, M_3)$ does not exist in the dataset. The real return $Z_\pi$ shows that $M_3$ results in the worst outcome. CQL ensures OOD actions are not overestimated, in comparison to actions in the dataset.

where $\theta_i^j(s,a)$ predicts the $j$-th quantile and $\delta$ is the Dirac delta. The corresponding distributional Bellman target is

$$\mathcal{T}Z(s,a) = r(s,a) + \gamma Z_{\hat{\theta}}(s',a'), s' \sim D, a' \sim \pi_\psi(\cdot \mid s'), \tag{2}$$

where $\hat{\theta}$ denotes the target network and $a'$ is sampled from the current policy, tying the learned distribution to the policy being optimized. The temporal-difference (TD) loss is

$$\mathcal{L}_{\text{TD}}(\theta) = \mathbb{E}_{\substack{(s,a,r,s')\sim\mathcal{D} \\ a'\sim\pi_\psi(\cdot|s')}} \left[ \rho_\tau^H(\mathcal{T}Z(s,a) - Z_\theta(s,a)) \right], \tag{3}$$

where $\rho_\tau^H$ is the asymmetric quantile Huber loss (Dabney et al., 2018) applied jointly across all fractions $\tau$. The target network is updated by Polyak averaging, $\hat{\theta} \leftarrow (1-\rho)\hat{\theta} + \rho\theta$. A scalar $Q$-value is recovered as $Q_\theta^Z(s,a) = \mathbb{E}[Z_\theta(s,a)]$.

**Conservative Q-Learning.** In the offline setting, CDQAC updates $Z_\theta$ using targets that can involve actions without support in the static dataset $\mathcal{D}$. This support mismatch, i.e. *distributional shift* between the state–action distribution in $\mathcal{D}$ and that induced by the learned policy, leads the critic to overestimate the values for out-of-distribution (OOD) actions, as illustrated in Fig. 2. This overestimation is not an issue for online RL, since it can explore these actions during training; however, offline RL cannot due to learning from a static dataset. To avoid this overestimation, we add Conservative Q-learning (CQL) (Kumar et al., 2020) to the loss of the critic. CQL penalizes overestimation of OOD actions, by introducing a regularization term used in combination with standard critic loss:

$$\mathcal{L}_Z(\theta) = \alpha_{\text{CQL}} \mathbb{E}_{s\sim\mathcal{D}} \left[ \log \sum_{a'\in\mathcal{A}(s)} \exp(Q_\theta^Z(s,a')) - \mathbb{E}_{a\sim\mathcal{D}}[Q_\theta^Z(s,a)] \right] + \mathcal{L}_{TD}(\theta), \tag{4}$$

where, $\alpha_{\text{CQL}}$ determines the strength of the penalty, and $\mathcal{L}_{TD}(\theta)$ is the loss in Eq. 3.

**Delayed Policy Updates.** CDQAC updates the critic using bootstrapped targets whose successor actions are sampled from the current policy $\pi_\psi(\cdot \mid s')$, while the policy is optimized against the critic's current estimates. Early in training, this coupling can amplify critic errors: noisy value estimates may increase the probability of poorly supported machine–operation pairs, which are then reused in subsequent Bellman targets. This issue is particularly harmful in offline scheduling, where the static dataset provides no online interaction to correct such errors.

We therefore update the policy only every $\eta$ critic steps. This gives the quantile critic more updates to fit the return distribution and apply conservative regularization before it is used for policy improvement. Delayed policy updates are established in online continuous-control RL (Fujimoto et al., 2018); here, we use them as a stabilization mechanism for offline discrete actor–critic learning with variable-size combinatorial action spaces. Our ablations show that this mechanism is important for preventing premature policy improvement

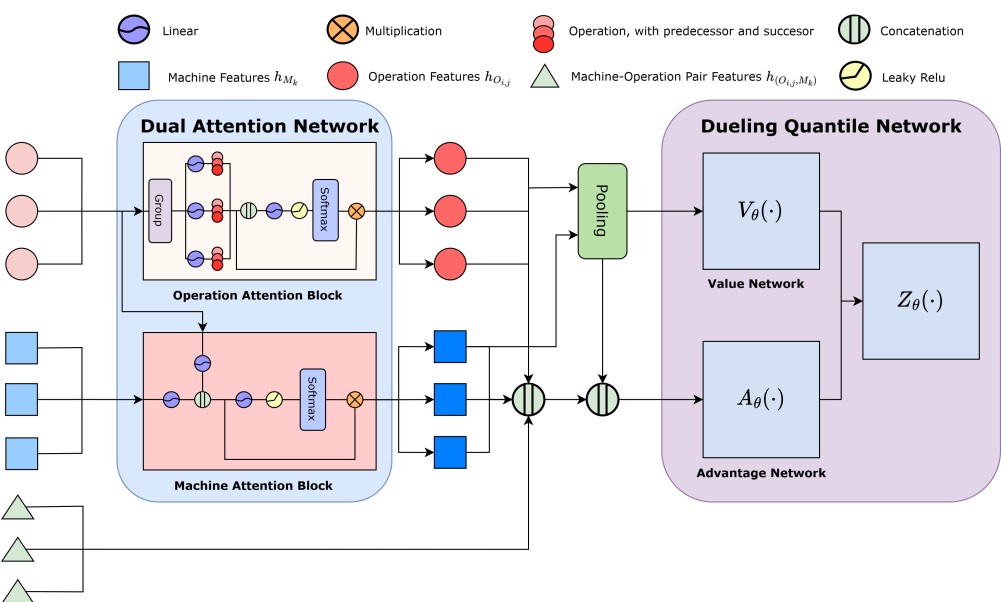

Figure 3: The network architecture. (Left) The Dual Attention Network (DAN) encodes the operations and machines. (Right) The Dueling Quantile Network uses these embeddings to learn the machine-operation pair, whereby it combines the Value $V_\theta$ and Advantage $A_\theta$ streams through Eq. 6.

from an unreliable critic. The policy loss is

$$\mathcal{L}_\pi(\psi) = \mathbb{E}_{\substack{s \sim \mathcal{D} \\ a \sim \pi_\psi(\cdot|s)}} \left[ -Q_\theta^Z(s, a) - \lambda \, \mathcal{H}[\pi_\psi(\cdot \mid s)] \right], \tag{5}$$

where the entropy bonus encourages diversity over feasible machine–operation pairs and prevents premature policy collapse.

## 4.1 Network Architecture

To encode an FJSP or JSP instance, we use a dual attention network (DAN), adapted from DANIEL (Wang et al., 2023), for both the policy network $\pi_\psi$ and the quantile critic $Z_\theta$. Fig. 3 shows our network architecture. DAN processes two parallel attention streams over the relevant operations $O_{i,j} \in \mathcal{O}_t$ and machines $M_k \in \mathcal{M}_t$, learning the complex relation between each machine-operation pair at timestep $t$ and embedding them as $h_{O_{i,j}}$ and $h_{M_k}$. A detailed explanation of DAN and the input features is provided in App. B.

From the machine embeddings $h_{M_k}$ and operation embeddings $h_{O_{i,j}}$, we compute a global embedding $h_G = \left[ \left( \frac{1}{|\mathcal{O}_t|} \sum_{O_{i,j} \in \mathcal{O}_t} h_{O_{i,j}} \right) \parallel \left( \frac{1}{|\mathcal{M}_t|} \sum_{M_k \in \mathcal{M}_t} h_{M_k} \right) \right]$, where $\parallel$ denotes concatenation. The actor network takes $h_G$, the operation and machine embeddings $h_{O_{i,j}}, h_{M_k}$, and the pair-specific features $h_{(O_{i,j}, M_k)}$ as input for the policy $\pi_\psi$, enabling it to select a machine-operation pair based on its embedding in relation to the global state.

The quantile critic in CDQAC uses a dueling architecture with separated value/advantage inputs, adapted to variable-size scheduling action spaces based on Wang et al. (2016), decomposing the state-action value into a value stream $V(s)$ and an advantage stream $A(s, a)$. We adopt this as an architectural mitigation of the propagation problem that also motivates the delayed policy update: as the action space grows, only the selected pair's advantage is updated per transition, so noisy critic estimates increasingly propagate to unsampled pairs through subsequent Bellman targets. $V_\theta$, by contrast, is updated at every step regardless of the sampled action, providing a stable shared state representation that anchors generalization across the action space and yields a more accurate $Z_\theta$. The two mechanisms are complementary: the delayed update stabilizes propagation temporally, the dueling decomposition stabilizes it architecturally. Whereas Wang et al. (2016) feed $V_\theta$ and $A_\theta$ the same input, we use separate inputs: $V_\theta$ receives only $h_G$, while $A_\theta$ additionally

receives the operation-, machine-, and pair-specific embeddings (Fig. 3). This lets $V_\theta$ focus on the state value while $A_\theta$ specializes to each individual machine-operation pair $(O_{i,j}, M_k)$, resulting in the following formulation:

$$Z_\theta(h_{O_{i,j}}, h_{M_k}, h_{(O_{i,j}, M_k)}, h_G) = V_\theta(h_G) + \Big( A_\theta(h_{O_{i,j}}, h_{M_k}, h_{(O_{i,j}, M_k)}, h_G)$$
$$- \frac{1}{|\mathcal{A}(s_t)|} \sum_{(O', M') \in \mathcal{A}(s_t)} A_\theta(h_{O'}, h_{M'}, h_{(O', M')}, h_G) \Big) \qquad (6)$$

where $\mathcal{A}(s_t)$ is the set of available machine-operation pairs $(O', M')$ in state $s_t$. In Eq. 6, we subtract the mean advantage from each action's advantage, which is required because the value and advantage streams are not uniquely identifiable. Both streams output $N$ quantile values, and Eq. 6 is applied independently at each fraction; as in QR-DQN, monotonicity is not strictly enforced, and only learned via the loss function (Eq. 3).

## 5  Experiments

**Generated & benchmark instances.**   For both problems we train on 500 generated instances and evaluate on a mix of held-out generated instances and standard benchmarks. **FJSP:** we train on instances of sizes $\{10 \times 5, 15 \times 10, 20 \times 10\}$, where each job has between $\lfloor 0.8m \rfloor$ and $\lfloor 1.2m \rfloor$ operations ($m$ being the number of machines) and processing times are integers drawn from $[1, 99]$. Evaluation uses 100 generated instances per size in $\{10 \times 5, 15 \times 10, 20 \times 10, 30 \times 10, 40 \times 10\}$, together with the Brandimarte (mk) (Brandimarte, 1993) and Hurink (edata, rdata, vdata) (Hurink et al., 1994) benchmarks. **JSP:** we train on 500 instances at $10 \times 5$ and $15 \times 10$ generated following Taillard (1993), and evaluate on the Taillard (Taillard, 1993) and Demirkol (Demirkol et al., 1998) benchmarks. Full benchmark details are given in App. C.

**Training dataset generation.**   Offline RL trains on a fixed dataset $\mathcal{D}$. We collect trajectories using three kind of heuristics: (i) Priority Dispatching Rules (**PDR**)—for FJSP, 4 job-selection $\times$ 4 machine-selection rules (16 trajectories per instance); for JSP, 4 job rules (machines fixed). (ii) Genetic Algorithms **GA** (Reijnen et al., 2023)—use the entire final population (typically higher quality, lower diversity than PDRs). (iii) **Random**—uniformly sample feasible actions. Consequently, we build four datasets: **PDR** (16 FJSP / 4 JSP trajectories), **GA** (200 trajectories per instance), **PDR–GA** (union), and **Random** (100 trajectories per instance). These datasets matches the setup used in offline RL work (Fu et al., 2020), where datasets with different qualities are used. From each trajectory, we extract the transitions on which CDQAC and offline RL baselines are trained. Heuristic details are in App. D.

**Metrics.**   We report the *optimality gap*: Gap $= \frac{C^j_{\max} - C_{\mathrm{ub}}}{C_{\mathrm{ub}}} \times 100$, which measures the difference between $C^j_{\max}$, the makespan found by the method $j$ and $C_{\mathrm{ub}}$, which is the optimal or best-known makespan for the given instance. For generated instances, we used solutions generated by OR tools (Perron et al., 2023), with a solving time limit of 30 minutes per instance, as reported in  Wang et al. (2023). For the benchmark instances, Taillard, Demirkol, Brandimarte, and Hurink, we used the best known solutions noted in the literature (Reijnen et al., 2023).

**Baselines.**   We benchmark CDQAC against both offline and online RL approaches and strong heuristics. Each learning-based policy is evaluated in two modes: **greedy** (argmax) and **sampling** (100 solutions sampled; best kept), averaging over three evaluation seeds (1, 2, 3).

(1) **Offline RL**: We compare CDQAC with Offline-LD (Remmerden et al., 2025), Behavioral Cloning (BC), Implicit Q-Learning (IQL) (Kostrikov et al., 2021), and a discrete TD3-AWR (Jackson et al., 2025). Offline-LD was originally developed for JSP and has not been applied to FJSP, IQL and TD3-AWR have not been applied to neither JSP or FJSP, and while BC has been used in related scheduling settings, it has not been evaluated in our setup. We therefore adapt all three baselines to FJSP using DAN (Wang et al., 2023) as the encoder network, matching CDQAC and isolating the contribution of its novel components. For Offline-LD, we additionally implement both the maskable QRDQN (mQRDQN) and discrete maskable SAC (d-mSAC) variants from (Remmerden et al., 2025). Each method is trained on the four datasets (PDR, GA, PDR-GA,

Table 1: Average gap (%) on all FJSP evaluation sets, for all training sizes ($10 \times 5$, $15 \times 10$, and $20 \times 10$). $\pi_\beta$ best performance of heuristics that generated dataset. **Bold** is best result of the method (row) for each training dataset (column).

|  |  | PDR | GA | PDR-GA | Random |
|---|---|---|---|---|---|
| **Greedy** | BC | $29.13 \pm 3.2$ | $13.91 \pm 0.6$ | $22.37 \pm 2.24$ | $21.85 \pm 2.51$ |
|  | Offline-LD (mQRDQN) | $22.26 \pm 2.43$ | $30.85 \pm 3.57$ | $21.80 \pm 3.64$ | $21.49 \pm 2.62$ |
|  | Offline-LD (d-mSAC) | $23.28 \pm 3.06$ | $21.02 \pm 2.13$ | $25.94 \pm 2.29$ | $16.91 \pm 1.89$ |
|  | IQL | $19.93 \pm 1.83$ | $20.66 \pm 2.18$ | $19.24 \pm 2.34$ | $21.34 \pm 3.54$ |
|  | TD3-AWR | $18.63 \pm 2.58$ | $19.91 \pm 1.08$ | $15.29 \pm 1.44$ | $14.72 \pm 0.50$ |
|  | **CDQAC (Ours)** | $\mathbf{12.34 \pm 1.72}$ | $\mathbf{13.06 \pm 2.10}$ | $\mathbf{11.31 \pm 1.33}$ | $\mathbf{10.68 \pm 0.51}$ |
| **Sampling** | BC | $10.71 \pm 0.99$ | $8.3 \pm 0.15$ | $9.49 \pm 0.56$ | $13.15 \pm 0.09$ |
|  | Offline-LD (mQRDQN) | $13.64 \pm 0.20$ | $14.26 \pm 0.26$ | $13.68 \pm 0.17$ | $13.63 \pm 0.23$ |
|  | Offline-LD (d-mSAC) | $11.61 \pm 1.32$ | $8.83 \pm 0.69$ | $11.69 \pm 1.23$ | $7.79 \pm 0.86$ |
|  | IQL | $10.01 \pm 0.58$ | $9.19 \pm 0.56$ | $9.48 \pm 0.59$ | $10.79 \pm 0.74$ |
|  | TD3-AWR | $8.86 \pm 0.52$ | $8.87 \pm 0.34$ | $7.86 \pm 0.27$ | $8.93 \pm 0.16$ |
|  | **CDQAC (Ours)** | $\mathbf{6.57 \pm 0.76}$ | $\mathbf{6.43 \pm 0.87}$ | $\mathbf{5.87 \pm 0.51}$ | $\mathbf{5.86 \pm 0.30}$ |
| $\pi_\beta$ |  | 14.13 | 6.74 | 6.74 | 28.16 |

Random) and three instance sizes ($10 \times 5$, $15 \times 10$, $20 \times 10$), as relative offline RL performance can vary substantially between datasets. Full details are provided in App. E.

(2) **Online RL**: We benchmark CDQAC against state-of-the-art online RL methods: DANIEL (Wang et al., 2023), and Residual (Ho et al., 2024) because, similar to CDQAC, they can solve both JSP and FJSP instances directly. In addition, for FJSP, we also compare with FJSP-DRL (Song et al., 2023) and two recent methods: DOAGNN (Zhao et al., 2025), an online PPO approach that decouples the disjunctive graph into separate machine–operation and operation–operation graphs, each processed by a dedicated GNN, and densifies the reward with minimum-completion-time and machine-idle-time terms; and PRISMA (Jesus et al., 2026), a hybrid multi-agent framework that pretrains decentralized policies on CP-generated expert demonstrations via behavioral cloning and guides subsequent PPO training with a CP-derived potential function for reward shaping. Both require online environment interaction, and PRISMA additionally relies on CP-SAT demonstrations, in contrast to CDQAC's purely offline training on random data. For JSP, L2D (Zhang et al., 2020) is added for comparison. We use published results of DANIEL, FJSP-DRL, DOAGNN and PRISMA directly and retrain Residual on the same distribution (1,000 instances, 20 runs). We compare against GA from (Reijnen et al., 2023), dispatching rules (MOR-SPT/EST), CP (30 minutes time limit), and a specialized GA algorithm 2SGA (Defersha & Rooyani, 2020). Additional JSP baselines include MOR, MWKR, MIP, and CP, with a 30 minute time limit for both MIP and CP.

**Training Setup.** We evaluate the stability of CDQAC by running all experiments with four different seeds (1, 2, 3, 4). Although this is standard practice in offline RL (Fu et al., 2020), online RL methods for FJSP (Song et al., 2023; Wang et al., 2023) typically report results from a single seed. Consequently, we present mean and standard deviation for our offline RL comparisons, but only single seed results (seed 1) when comparing with online methods. Our hardware is a NVIDIA A100 GPU, Intel Xeon CPU, and 360GB of RAM. Detailed descriptions of the hyperparameters and the network architecture can be found in App. G.

### 5.1 Offline RL Methods Comparison with Different Training Datasets

Table 1 shows the results of our method compared to other offline approaches. CDQAC outperforms both versions of Offline-LD, IQL, TD3-AWR and BC by a significant margin. Furthermore, CDQAC consistently outperforms all heuristics that generated the datasets ($\pi_\beta$). In contrast, the other offline RL baselines, Offline-LD, IQL, TD3-AWR never outperformed *GA* with the sampling evaluation, or even the PDR heuristics with greedy evaluation. The second highest performance was achieved with BC, when trained on *GA* (Greedy:13.91%, Sampling: 8.3%); however, BC still performed worse than CDQAC, even when trained on the same *GA* dataset (Greedy: 13.06%, Sampling: 6.43%). Additional results of our offline RL comparison are in App. I.2.

CDQAC (Greedy: 10.68%, Sampling: 5.86%), Offline-LD (d-mSAC) (Greedy: 16.91%, Sampling: 7.79%), and TD3-AWR, with the greedy evaluation (Greedy: 14.72%), achieve their best performance when trained

on the Random dataset. For these methods, a diverse but suboptimal dataset is thus preferred over a narrow expert one, a pattern that differs from the data-quality ordering reported on standard offline-RL benchmarks (Schweighofer et al., 2022; Kumar et al., 2022). We explain this in detail in the following section.

## 5.2 Why FJSP/JSP favors broad-coverage data

Standard offline-RL benchmarks exhibit a clear data-quality ordering, in which random data is the weakest training source and noisy-expert data dominates (Schweighofer et al., 2022; Kumar et al., 2022). We argue that this ordering is not intrinsic to offline RL but follows from properties of the benchmarks on which it is typically evaluated. In the case of JSP and FJSP, it has two structural properties. First, the reward $r_t$ is the negative partial-makespan increase, so the undiscounted return equals the makespan: every transition carries a calibrated, objective-aligned signal rather than a sparse or proxy reward. Second, constructive scheduling policies share a fixed horizon — the number of operations — so random and expert heuristics produce equal-length trajectories, removing the trajectory-length bias that can favor expert data in other settings (Li et al., 2023). This allows an expert and random heuristic to explore the same states. Under these two conditions, we hypothesize that offline performance depends less on data *quality* than on state—action *coverage*: a learner that estimates accurate returns and stitches high-value segments should prefer the dataset with the broadest

Table 2: Average **State-Action Coverage** (SACo, ↑), **Trajectory Quality** (TQ), and **embedding-space $k$-NN density proxy** ($k$=10, ↓) of the FJSP training datasets, averaged over the three training sizes. PDR is the reference dataset for SACo. **Bold** indicates the best value per metric. For each metric, we measure the Spearman Correlation $\rho_s$, with significance $p_s$, against the performance of CDQAC as reported in Table 1, and the $\epsilon$-PDR experiments in Table 10.

| Dataset | SACo (↑) | TQ (↑) | $k$-NN dist. (↓) |
|---|---|---|---|
| PDR | $1 \pm 0$ | $0.58 \pm 0.02$ | $0.033 \pm 0.011$ |
| GA | $2.96 \pm 0.49$ | $\mathbf{0.89 \pm 0.01}$ | $0.030 \pm 0.009$ |
| PDR-GA | $3.96 \pm 0.49$ | $0.81 \pm 0.01$ | $\mathbf{0.029 \pm 0.009}$ |
| Random | $\mathbf{7.70 \pm 0.77}$ | $0.58 \pm 0.02$ | $0.031 \pm 0.009$ |
| $\rho_s$ | -0.946 | 0.157 | 0.578 |
| $p_s$-value | 0.0006 | 0.711 | 0.14 |

coverage of the state–action space, even when every individual trajectory is poor. Since a uniformly random policy maximizes per–state action diversity by construction, we now test empirically whether broad–coverage data is the *strongest* source for this JSP/FJSP setting due to the structural properties of these problems.

**Broad-coverage data is the strongest source.** Table 1 shows the performance on different source datasets. As expected, CDQAC achieves its best gaps on the Random dataset (Greedy 10.68%, Sampling 5.86%) and on PDR-GA (11.31%, 5.87%), and its worst on the narrow PDR (12.34%, 6.57%) and expert GA (13.06%, 6.43%) datasets. The same ranking holds for Offline-LD (d-mSAC), which is also best on Random, indicating that the effect is not specific to CDQAC. IQL (Kostrikov et al., 2021), which never evaluates OOD actions on its targets, does not show the preference (worst in Random in both modes, Table 1), which is consistent with coverage that matters specifically where OOD actions enter the Bellman target.

**Coverage best predicts performance.** We characterize each dataset using three complementary metrics (Table 2). State–Action Coverage (SACo) measures support breadth, $\text{SACo}(\mathcal{D}) = u_{s,a}(\mathcal{D})/u_{s,a}(\mathcal{D}_{\text{ref}})$, where $u_{s,a}(\mathcal{D})$ is the number of unique state–action pairs and PDR is the reference dataset. Trajectory Quality (TQ) measures normalized return, $\text{TQ}(\mathcal{D}) = (R - R_{\min})/(R_{\max} - R_{\min})$ (Schweighofer et al., 2022). Finally, an embedding-space $k$-NN distance measures local support around high-quality behavior. For each instance, we use the best of 1,000 stochastic policy rollouts as a near-optimal reference trajectory, embed its transitions and the dataset transitions with a trained DAN encoder, and compute the mean distance to the $k$=10 nearest dataset transitions. Thus, higher SACo and TQ are preferable, whereas a lower $k$-NN distance indicates denser local support.

Random attains the highest coverage, with a SACo 7.7× that of PDR, while ranking only third in local density ($0.031 \pm 0.009$). Across the original and $\varepsilon$-PDR datasets, SACo is the only metric significantly correlated with CDQAC's gap ($\rho_s = -0.946$, $p_s = 0.0006$): broader coverage is associated with better performance. The density proxy has the expected direction; larger distances are associated with larger gaps but is not significant ($\rho_s = 0.578$, $p_s = 0.140$). TQ is also uncorrelated with performance ($\rho_s = 0.157$, $p_s = 0.711$): GA has the highest TQ yet is among the weakest training sources, whereas PDR matches Random's mean TQ despite far stronger best trajectories ($\pi_\beta$ gaps of 14.13% vs. 28.16%, Table 1), indicating that some dispatching rules perform no better than a random policy.

Table 3: Results on ablation study of dataset properties. Standard is the setup used in all other experiments. Trained on $10 \times 5$. **Bold**: indicates best gap (%).

| | Training Dataset | Standard | Early Termination | Proxy Reward | Sparse Reward |
|---|---|---|---|---|---|
| Greedy | PDR | $11.92 \pm 0.49$ | $\mathbf{27.41 \pm 3.88}$ | $19.77 \pm 5.02$ | $13.73 \pm 0.67$ |
| | GA | $11.94 \pm 0.38$ | $38.04 \pm 2.44$ | $\mathbf{17.6 \pm 2.2}$ | $\mathbf{12.87 \pm 0.82}$ |
| | PDR-GA | $11.94 \pm 0.53$ | $33.78 \pm 3.92$ | $17.74 \pm 3.31$ | $12.89 \pm 0.34$ |
| | Random | $\mathbf{11.61 \pm 0.38}$ | $39.11 \pm 3.72$ | $20.48 \pm 3.32$ | $13.74 \pm 0.81$ |
| Sampling | PDR | $\mathbf{6.42 \pm 0.18}$ | $\mathbf{13.14 \pm 2.54}$ | $10.06 \pm 2.8$ | $7.85 \pm 0.53$ |
| | GA | $6.58 \pm 0.29$ | $19.57 \pm 3.05$ | $9.66 \pm 1.51$ | $\mathbf{6.68 \pm 0.39}$ |
| | PDR-GA | $6.57 \pm 0.31$ | $17.45 \pm 2.49$ | $\mathbf{9.56 \pm 2.05}$ | $7.35 \pm 0.2$ |
| | Random | $6.43 \pm 0.21$ | $20.6 \pm 3.1$ | $11.41 \pm 2.48$ | $7.56 \pm 0.51$ |

Broad coverage can benefit conservative offline RL through two complementary mechanisms. First, suboptimal trajectories provide *negative data* that reveal the outcomes of alternative actions (Kumar et al., 2022). Together with scheduling's dense, objective-aligned reward, these observations help calibrate pessimism by distinguishing actions that are empirically poor from those that are merely unobserved. Second, coverage of decision points relevant to high-quality policies reduces *intrinsic uncertainty* caused by missing transitions (Jin et al., 2021) and provides useful transitions for trajectory stitching. Random therefore combines adequate local support around high-quality behavior with substantially broader evidence about alternative decisions. App. F provides further analysis and controlled coverage interventions.

**Confirming the two preconditions.** As stated earlier, JSP and FJSP have unique structural properties: a dense, objective-aligned reward and equal-length trajectories. If either fails, the coverage advantage of random data should disappear. We test this through three ablations (Table 3): early termination (10% chance per timestep), proxy rewards (negative rewards adjusted to $-1$), and sparse rewards (makespan given only at the terminal state). The predictions hold. Early termination breaks the equal-length property and is harmful on all datasets, and most so on Random: without access to later states, the intrinsic uncertainty is maximal and the coverage benefit vanishes, mirroring why random policies fail on Atari, where they terminate early at failure states (Li et al., 2023), which results in a lower density around the optimal policy for these problem settings. FJSP and JSP do not have this issue, given that density has the lowest correlation to performance according to Table 2. Proxy rewards break objective alignment and degrade Random most relative to GA or PDR-GA, since stitching requires a reward that distinguishes good transitions from bad ones. With a sparse reward the degradation is smallest overall, but Random again degrades most, indicating that expert structure can partially substitute for a dense signal whereas coverage cannot. Together, these results match the predictions derived from the two properties identified in advance, both of which the standard JSP and FJSP reward function (Zhang et al., 2020; Wang et al., 2023; Song et al., 2023; Iklassov et al., 2023; Remmerden et al., 2025; Smit et al., 2025) satisfies by construction, placing JSP and FJSP in a different regime from the settings where offline RL is usually studied — one where state–action coverage rather than data quality is the dataset property most strongly associated with performance.

### 5.3 Comparison with online RL on FJSP

Online RL remains the dominant learning paradigm for JSP and FJSP, and offline methods generally underperform when online interaction is available (Fu et al., 2020; Fujimoto et al., 2019; Kumar et al., 2023). The only previously reported scheduling exception, Offline-LD (Remmerden et al., 2025), outperforms L2D but relies on (near-)optimal CP demonstrations. We therefore investigate whether CDQAC can compete with online RL while learning from static random-policy data, without solver supervision or environment interaction during training.

On the standard FJSP benchmarks (Table 4), CDQAC achieves the lowest gap in five of the eight benchmark–inference-mode combinations against FJSP-DRL (Song et al., 2023), Residual (Ho et al., 2024), DOAGNN (Zhao et al., 2025), DANIEL (Wang et al., 2023), and PRISMA (Jesus et al., 2026). The exceptions are both evaluations on Hurink edata, where PRISMA performs best, and the $10 \times 5$ sampling evaluation on rdata, where DANIEL narrowly leads (4.95% vs. 5.59%). Against PRISMA specifically, CDQAC obtains lower gaps in six of eight comparisons despite using only 500 training instances, whereas PRISMA uses 1,000

Table 4: Results FJSP benchmarks sets. CDQAC trained on Random dataset; all models on 10×5 or 15×10 instances, except DOAGNN and PRISMA, who are trained on mixed datasets. **Bold** indicates best performance. The makespan and gap of 2SGA on la(vdata) are computed on instances la01–la30, as reported in Defersha & Rooyani (2020).

| | Method | mk | | edata | | rdata | | vdata | |
|---|---|---|---|---|---|---|---|---|---|
| | | Gap(%) | Time(s) | Gap(%) | Time(s) | Gap(%) | Time(s) | Gap(%) | Time(s) |
| | DOAGNN | 31.64 | 6.73 | 16.21 | 12.81 | 11.83 | 12.98 | 4.32 | 11.31 |
| | PRISMA | 13.80 | 0.394 | **13.70** | 0.407 | 11.40 | 0.435 | 4.90 | 0.464 |
| Greedy 10×5 | FJSP-DRL | 28.52 | 1.26 | 15.53 | 1.4 | 11.15 | 1.4 | 4.25 | 1.37 |
| | Residual | 25.53 | 0.68 | 15.97 | 0.5 | 11.78 | 0.63 | 2.8 | 0.8 |
| | DANIEL | 13.58 | 1.29 | 16.33 | 1.37 | 11.42 | 1.37 | 3.28 | 1.37 |
| | **CDQAC (Ours)** | 13.04 | 1.1 | 13.86 | 1.18 | **10.10** | 1.18 | **2.75** | 1.18 |
| 15×10 | FJSP-DRL | 26.77 | 1.25 | 15 | 1.4 | 11.14 | 1.4 | 4.02 | 1.37 |
| | Residual | 25.22 | 0.68 | 16.99 | 0.5 | 11.19 | 0.62 | 4.04 | 0.79 |
| | DANIEL | 12.97 | 1.3 | 14.41 | 1.38 | 12.07 | 1.36 | 3.75 | 1.37 |
| | **CDQAC (Ours)** | **12.64** | 1.08 | 14.74 | 1.15 | 10.47 | 1.14 | 3.13 | 1.14 |
| | DOAGNN | 18.62 | 41.42 | 8.64 | 50.21 | 5.83 | 49.85 | 1.44 | 52.12 |
| | PRISMA | 10.28 | 1.968 | **7.72** | 1.923 | 6.83 | 2.008 | 1.69 | 2.092 |
| Sampling 10×5 | FJSP-DRL | 18.56 | 4.13 | 8.17 | 4.91 | 5.57 | 4.81 | 1.32 | 4.71 |
| | Residual | 21.65 | 65.01 | 13.61 | 49.84 | 7.42 | 60.75 | 1.76 | 80.37 |
| | DANIEL | 9.53 | 4.12 | 9.08 | 4.71 | **4.95** | 4.73 | 0.69 | 4.77 |
| | **CDQAC (Ours)** | 8.96 | 3.36 | 9.4 | 3.82 | 5.59 | 3.84 | **0.65** | 3.84 |
| 15×10 | FJSP-DRL | 19 | 4.13 | 8.69 | 4.87 | 5.95 | 4.82 | 1.34 | 4.72 |
| | Residual | 19.91 | 66.09 | 11.94 | 50.61 | 8.25 | 61.52 | 1.58 | 77.59 |
| | DANIEL | 8.95 | 4.08 | 8.72 | 4.7 | 5.49 | 4.73 | 0.72 | 4.75 |
| | **CDQAC (Ours)** | **7.94** | 3.22 | 7.77 | 3.66 | 5.08 | 3.68 | 0.69 | 3.72 |
| | MOR-SPT | 25.67 | 0.1 | 17.75 | 0.11 | 14.38 | 0.1 | 6.06 | 0.11 |
| | MOR-EST | 29.59 | 0.1 | 17.59 | 0.11 | 14.3 | 0.11 | 5.59 | 0.11 |
| | GA | 14.29 | 232.95 | 4.55 | 237.06 | 4.43 | 243.91 | 0.67 | 283.97 |
| | 2SGA | 3.17 | 57.7 | - | - | - | - | 0.39 | 51.43 |
| | CP | 1.5 | 1447 | 0 | 900 | 0.11 | 1397 | 0 | 639 |

instances, CP demonstrations for pretraining, and CP-derived reward shaping during online PPO training. Because these benchmark families differ from CDQAC's training distribution, the results also demonstrate robustness to distribution shift. Paired significance tests against DANIEL across the FJSP evaluations remain significant under both single- and multi-seed comparisons (App. J). To our knowledge, this is the first offline RL method trained solely on random-policy data that outperforms online RL baselines across both JSP and FJSP.

On generated instances (Table 5), CDQAC performs comparably to DANIEL at $10 \times 5$ and outperforms it at $15 \times 10$, although DANIEL remains stronger when trained directly on $20 \times 10$. Conversely, when both methods are trained on $10 \times 5$, CDQAC performs better on the substantially larger $30 \times 10$ and $40 \times 10$ instances (Table 6). The on-/off-policy distinction offers a possible explanation: DANIEL's PPO policy is optimized only on rollouts induced by its current policy on the training sizes, which may encourage specialization to training instance distribution. CDQAC is off-policy and reuses all the transition during training. Consequently, online RL is most competitive in-distribution, whereas CDQAC's relative advantage grows on unseen benchmark distributions (Table 4) and larger instances (Table 6), although directly learning on the largest action spaces remains challenging (Sect. 5.5).

Direct training on large FJSP instances such as $20 \times 10$ presents additional challenges due to the size of the action space, which grows from at most 50 machine-operation pairs at $10 \times 5$ to 200 at $20 \times 10$. As the ablation in Sect. 5.5 shows, the delayed policy update and the dueling architecture become increasingly important as this action space grows, since only a single pair's state–action value is updated per transition and any noisy estimate propagates to a larger fraction of subsequent Bellman targets—an effect that, unlike in online RL, cannot be corrected through environment interaction. App. H confirms that this is a training issue rather than a generalization issue, as CDQAC converges stably for both $10 \times 5$ and $15 \times 10$ across all training datasets, but not for $20 \times 10$. Nevertheless, our results in Table 6 demonstrate that training on large instance sizes is not required to achieve state-of-the-art performance. CDQAC learns scheduling policies from small problem sizes ($10 \times 5$) that generalize successfully to larger problems; in fact, it outperforms DANIEL on large unseen instances (e.g., $30 \times 10$ and $40 \times 10$ in Table 6), showing better generalization than online

Table 5: Results generated FJSP evaluation instances. CDQAC trained on Random dataset; training instances size is same as evaluation instance size. **Bold** indicates best performance per evaluation mode.

Table 6: Generalization to large FJSP instances (30×10, and 40×10). CDQAC trained on Random dataset; training size 10×5. **Bold** indicates best performance per evaluation mode.

| Method | 10 × 5 | | 15 × 10 | | 20 × 10 | |
|---|---|---|---|---|---|---|
| | Gap(%) | Time(s) | Gap(%) | Time(s) | Gap(%) | Time(s) |
| *Greedy* FJSP-DRL | 16.03 | 0.45 | 16.33 | 1.43 | 10.15 | 1.91 |
| Residual | 15.23 | 0.27 | 15.93 | 0.85 | 10.01 | 1.28 |
| DANIEL | **10.87** | 0.45 | 12.42 | 1.35 | **1.31** | 1.85 |
| **CDQAC (Ours)** | 11.56 | 0.39 | **11.1** | 1.16 | 4.34 | 1.56 |
| *Sampling* FJSP-DRL | 9.66 | 1.11 | 12.13 | 3.98 | 9.64 | 6.23 |
| Residual | 9.85 | 27.04 | 12.38 | 77.5 | 9.81 | 116.41 |
| DANIEL | **5.57** | 0.74 | 6.79 | 3.89 | **-1.03** | 6.35 |
| **CDQAC (Ours)** | 5.98 | 0.64 | **5.85** | 3.06 | 1.79 | 4.83 |
| MOR-SPT | 19.67 | 0.03 | 17.89 | 0.1 | 11.25 | 0.15 |
| MOR-EST | 19.66 | 0.03 | 19.98 | 0.1 | 12.08 | 0.14 |
| GA | 6.0 | 71.65 | 10.42 | 266.15 | 6.78 | 348.87 |

| Method | 30 × 10 | | 40 × 10 | |
|---|---|---|---|---|
| | Gap(%) | Time(s) | Gap(%) | Time(s) |
| *Greedy 10 × 5* FJSP-DRL | 14.61 | 2.86 | 14.21 | 3.82 |
| Residual | 13.16 | 2.11 | 12.82 | 3.1 |
| DANIEL | 5.1 | 2.78 | 3.65 | 3.77 |
| **CDQAC (Ours)** | 4.43 | 2.32 | **3.17** | 3.19 |
| *Sampling 10 × 5* FJSP-DRL | 12.36 | 12.79 | 12.26 | 24.54 |
| Residual | 12.94 | 213.89 | 12.85 | 319.69 |
| DANIEL | 4.43 | 12.37 | 3.77 | 22.58 |
| **CDQAC (Ours)** | 3.11 | 9.57 | **2.21** | 16.01 |
| MOR-SPT | 14.99 | 0.23 | 14.57 | 0.33 |
| MOR-EST | 15.88 | 0.22 | 15.17 | 0.32 |
| GA | 11.26 | 521.19 | 11.26 | 736.36 |

Table 7: Results JSP benchmarks. Average gap (%) is reported. CDQAC trained on Random dataset for 10 × 5. For DANIEL Wang et al. (2023), only Tailard was reported. **Bold** indicates best result.

| Instance Size | | | | | Greedy | | | | Sampling | | | Exact | |
|---|---|---|---|---|---|---|---|---|---|---|---|---|---|
| | MWR | MOR | L2D | Offline-LD | DANIEL | Residual | **CDQAC (Ours)** | DANIEL | Residual | **CDQAC (Ours)** | MIP | CP |
| *Taillard* 15 × 15 | 18.9 | 21.4 | 28.1 | 25.8 | 19.0 | 17.6 | **15.0** | 13.2 | 13.3 | **10.4** | 0.1 | 0.1 |
| 20 × 15 | 23.0 | 23.6 | 32.7 | 30.2 | 22.1 | 21.2 | **17.7** | 17.4 | 16.1 | **13.2** | 3.2 | 0.2 |
| 20 × 20 | 21.6 | 21.7 | 31.8 | 28.9 | 18.0 | 18.0 | **17.6** | 13.3 | 15.8 | **12.9** | 2.9 | 0.7 |
| 30 × 15 | 24.3 | 23.2 | 30.2 | 29.2 | 21.7 | 20.1 | **19.1** | 17.2 | 18.0 | **14.9** | 10.7 | 2.1 |
| 30 × 20 | 24.8 | 25.0 | 35.2 | 33.1 | 23.2 | 22.3 | **21.2** | 19.0 | 19.7 | **17.9** | 13.2 | 2.8 |
| 50 × 15 | 16.5 | 17.3 | 21.0 | 20.6 | 14.8 | 15.6 | **13.0** | 12.7 | 13.2 | **9.9** | 12.2 | 3.0 |
| 50 × 20 | 18.1 | 17.9 | 26.1 | 24.3 | 16.0 | 14.4 | **12.8** | 13.1 | 14.1 | **11.0** | 13.6 | 2.8 |
| 100 × 20 | 8.3 | 9.1 | 13.3 | 12.7 | 7.3 | 6.5 | **5.3** | 5.9 | 6.5 | **3.6** | 11.0 | 3.9 |
| Mean | 19.4 | 19.9 | 27.3 | 25.6 | 18.2 | 17.0 | **15.2** | 14.4 | 14.6 | **11.7** | 8.4 | 2.0 |
| *Demirkol* 20 × 15 | 27.8 | 30.3 | 36.3 | 35.8 | – | 26.1 | **22.9** | – | 22.6 | **18.4** | 5.3 | 1.8 |
| 20 × 20 | 26.8 | 26.9 | 34.4 | 32.8 | – | 21.5 | **20.3** | – | 18.9 | **16.5** | 4.7 | 1.9 |
| 30 × 15 | 31.9 | 36.4 | 37.8 | 38.8 | – | 27.6 | **27.1** | – | 29.4 | **23.1** | 14.2 | 2.5 |
| 30 × 20 | 31.9 | 33.7 | 38.0 | 36.0 | – | 29.9 | **27.9** | – | 28.3 | **23.4** | 16.7 | 4.4 |
| 40 × 15 | 26.5 | 35.5 | 34.6 | 35.5 | – | 26.2 | **25.5** | – | 28.4 | **20.2** | 16.3 | 4.1 |
| 40 × 20 | 32.0 | 35.9 | 39.2 | 38.5 | – | **27.7** | 27.9 | – | 30.9 | **24.1** | 22.5 | 4.6 |
| 50 × 15 | 27.3 | 34.8 | 33.2 | 34.1 | – | 27.4 | **25.0** | – | 29.5 | **21.7** | 14.9 | 3.8 |
| 50 × 20 | 29.9 | 36.5 | 37.7 | 38.9 | – | 30.0 | **28.6** | – | 32.8 | **25.1** | 22.5 | 4.8 |
| Mean | 29.2 | 33.7 | 36.4 | 36.3 | – | 27.0 | **25.7** | – | 27.6 | **21.6** | 14.6 | 3.5 |

baselines. These results highlight the potential of offline RL, and directly tackling large-scale discrete action spaces–e.g., through factorized action spaces that decouple operation and machine selection–is an interesting future research direction.

## 5.4 Comparison on JSP Instances

Table 7 shows the results for the JSP evaluation, demonstrating that CDQAC surpasses all baselines, including Offline-LD and online methods (Residual, DANIEL, L2D). Notably, CDQAC outperforms Offline-LD using only random data, despite the latter utilizing expert data. On Taillard instances, CDQAC achieves gaps of 15.2% (greedy) and 11.7% (sampling), beating DANIEL (18.2%/14.4%) and Residual (17.0%/14.6%). This advantage extends to the Demirkol benchmark, where CDQAC (25.7%/21.6% sampling) improves over Residual (27.0%/27.6%), confirming its effectiveness for JSP. Table 7 shows that CDQAC demonstrates favorable scaling on large Taillard instances. CDQAC's sampling evaluation outperforms MIP on 50 × 15, and greedy on 50 × 20 and 100 × 20. For CP, CDQAC's sampling evaluation outperforms on 100 × 20. This shows that CDQAC scales to larger JSP problem sizes more effectively than exact solvers, like MIP and CP. More results can be found in App. I.3 and in App. I.5, where we compare CDQAC to JSP-only baselines.

## 5.5 Ablation Study

We ablate each component by removing it and measuring the resulting average gap increase, on the Random dataset and across the three training sizes (Fig. 4; full per-size results in App. I.1). Two components dominate, and—critically—their importance grows sharply with the size of the action space. Removing the Delayed Policy Update increases the gap by 11.7% at $10 \times 5$, 38.5% at $15 \times 10$, and 112.8% at $20 \times 10$ (54.3% on average), and removing our Dueling Critic with separated inputs for $V_\theta$ and $A_\theta$ follows the same trend on a smaller scale ($4.3\% \rightarrow 5.8\% \rightarrow 16.7\%$, 8.9% on average). The Quantile Critic (4.2%) and CQL regularization (2.5%) contribute more modestly and roughly uniformly across sizes. The scale-dependence of the two dominant components is a sharp departure from online continuous control, where

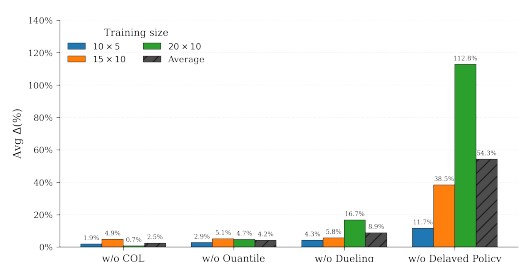

Figure 4: The results of the ablation study on each component of CDQAC: CQL, Quantile Critic, Dueling architecture, and Policy Delay. The results show the average increase in gap $\Delta(\%)$ for each training size. Lower is better.

Fujimoto et al. (2018) report only an 11.4% degradation from removing delayed updates in MuJoCo and find delays beyond $\eta = 2$ harmful; CDQAC instead achieves its best performance at $\eta = 4$.

**Sample Efficiency**    We evaluated CDQAC's sample efficiency by reducing the number of instances in the Random training dataset. Fig. 5 shows that increasing the size of the dataset has only a marginal positive effect on performance. We noticed the greatest performance difference for $10 \times 5$ between 5 instances (greedy 11.8%) and 10 instances (greedy 10.5%), while other results show no significant differences. Due to CDQAC being an off-policy RL method, it is able to reuse examples effectively. Because the bootstrapped target $\mathcal{T}Z(s, a) = r + \gamma Z_{\hat{\theta}}(s', a')$ with $a' \sim \pi_\psi(\cdot \mid s')$ is recomputed against the current policy and target critic, the regression target for a given transition shifts as $\pi_\psi$ and $\hat{\theta}$ evolve, so each time a transition is resampled it carries a fresh learning signal rather than a redundant one. Therefore, CDQAC achieves near-full performance using only 1 to 5% of the original dataset. Extended results are in App. I.4.

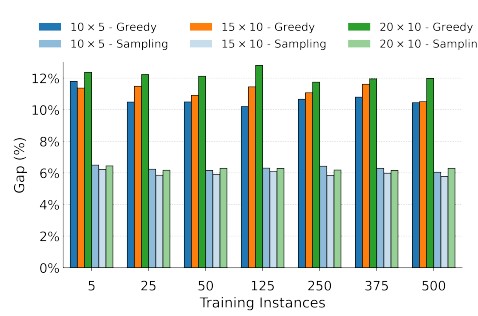

Figure 5: Results of reducing the number of instances in Random dataset, evaluated on FJSP benchmarks Brandimarte and Hurink.

# 6    Conclusion

We proposed CDQAC, a novel offline RL method for JSP and FJSP. CDQAC is a offline RL method for JSP, FJSP that has shown to outperform strong online RL baselines, while training purely on data from a random policy. This is a stronger setting than prior offline RL successes in either JSP or FJSP, which required near-optimal CP solutions. CDQAC achieves this by learning an accurate representation of the returns of a possible scheduling action from a static dataset, enabling CDQAC to "stitch" together high-quality partial solutions into a new policy. A central message of this work is that the conventional offline-RL ordering—random data weakest, noisy-expert strongest—does not apply to scheduling problems, FJSP and JSP. We hypothesized that, due to two structural properties of JSP and FJSP — a dense, objective-aligned reward and equal-length trajectories across heuristics — coverage rather than data quality should be the stronger predictor of offline performance, and therefore that random data should be the strongest source; our experiments support this hypothesis, with coverage the only measured dataset property that correlates significantly with performance. CDQAC also generalizes well from small to larger instance sizes.

Offline RL remains underexplored in scheduling and, more broadly, in COPs. In future work, we plan to extend our approach to other scheduling and combinatorial optimization problems, and to real-world scheduling problems, for which building a simulated environment is infeasible but has suboptimal training data generated by heuristics.

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

---

**Algorithm 1** Training Procedure of CDQAC

---

**Input:** Dataset $D$, batch size $B$, policy update frequency $\eta$, total training steps $T$, CQL coefficient $\alpha_{\text{CQL}}$, entropy coefficient $\lambda$, target update rate $\rho$, learning rates $\ell_\psi, \ell_\theta$

Initialize policy network $\psi$, critic network $\theta$, target network $\hat{\theta} \leftarrow \theta$

**for** $t = 1$ **to** $T$ **do**

    Sample mini-batch $\{(s_i, a_i, r_i, s_i')\}_{i=1}^B \sim D$

    Compute target quantiles: $\mathcal{T}Z_i \leftarrow r_i + \gamma Z_{\hat{\theta}}(s_i', a_i')$ where $a_i' \sim \pi_\psi(\cdot \mid s_i')$

    Compute TD loss: $\mathcal{L}_{\text{TD}}(\theta) \leftarrow \frac{1}{B} \sum_{i=1}^B \sum_{j=1}^N \rho_{\tau_j}^H(\mathcal{T}Z_i - Z_\theta(s_i, a_i))$

    Compute conservative critic loss: $\mathcal{L}_Z(\theta) \leftarrow \frac{1}{B} \sum_{i=1}^B \left[ \log \sum_{a' \in \mathcal{A}(s_i)} \exp(Q_\theta^Z(s_i, a')) - Q_\theta^Z(s_i, a_i) \right] + \mathcal{L}_{\text{TD}}(\theta)$

    Update critic: $\theta \leftarrow \theta + \ell_\theta \nabla_\theta \mathcal{L}_Z(\theta)$

    **if** $t \bmod \eta = 0$ **then**

        Compute policy loss: $\mathcal{L}_\pi(\psi) \leftarrow \frac{1}{B} \sum_{i=1}^B \left[ \sum_{a \in \mathcal{A}(s_i)} -Q_\theta^Z(s_i, a)\pi_\psi(a \mid s_i) + \lambda \mathcal{H}[\pi_\psi(\cdot \mid s_i)] \right]$

        Update policy: $\psi \leftarrow \psi + \ell_\psi \nabla_\psi \mathcal{L}_\pi(\psi)$

    **end if**

    Update target network: $\hat{\theta} \leftarrow (1 - \rho)\hat{\theta} + \rho\theta$

**end for**

---

## A  Pseudocode

Algorithm 1 shows the training process of CDQAC. In it, we train CDQAC using a static dataset $D = (s, a, r, s')$ of scheduling transitions. At each training step, we sample a mini-batch of $B$ transitions from $D$. For each transition, we compute the target $\mathcal{T}Z = r + \gamma Z_{\hat{\theta}}(s', a')$ using the target network $\hat{\theta}$ and next actions $a' \sim \pi_\psi(\cdot \mid s')$ drawn from the current policy. The critic is optimized through a conservative quantile-based objective, combining the temporal difference (TD) loss $\mathcal{L}_{\text{TD}}$ (Eq. 3) with a CQL penalty that discourages overestimation of out-of-distribution actions (Eq. 4). The critic parameters $\theta$ are updated via gradient descent on the combined loss $\mathcal{L}_Z$.

To stabilize training, we employ a delayed policy update strategy: the actor $\pi_\psi$ is updated every $\eta$ steps by minimizing the Q-learning objective (Eq. 5), with the entropy bonus $\mathcal{H}[\pi_\psi(\cdot \mid s)]$. The policy update relies on the scalarized quantile values $Q_\theta^Z(s, a) = \mathbb{E}[Z_\theta(s, a)]$, where $Z_\theta$ is the minimum of two dueling quantile networks. Finally, the target network is updated using Polyak averaging: $\hat{\theta} \leftarrow (1 - \rho)\hat{\theta} + \rho\theta$.

## B  Network Architecture

The dual attention network (Wang et al., 2023) (DAN) is an attention-based network architecture for JSP and FJSP that encodes the operation features $h_{O_{i,j}}^{(L)}$, and machine features $h_{M_k}^{(L)}$, where $L$ presents the current layer input, so $L = 1$ is the input features. DAN is able to learn the complex relation between each operation $O_{i,j}$ and each compatible machine $M_k$, through separate *operation attention blocks* and *machine attention blocks* as seen in Fig. 3 in Sect. 4.1. In this section, we provide an overview of each attention block, and their interaction. Afterwards, we state the features used for the operations, machines and machine-operation pairs.

**Operation Attention Block.**  To capture the sequential nature of operations within jobs, the operation attention blocks attend each operation $O_{i,j}$ in the context of its predecessor $O_{i,j-1}$ and successor $O_{i,j+1}$, if they exist. An attention coefficient is calculated between these operations:

$$a_{i,j,p} = \text{Softmax}\Big( \text{LeakyReLU}\Big( \mathbf{V}^T \Big[ \Big( \mathbf{W}h_{O_{i,j}}^{(L)} \parallel \mathbf{W}h_{O_{i,p}}^{(L)} \Big) \Big] \Big) \Big), \tag{7}$$

where $\mathbf{W}$, and $\mathbf{V}$ are learned projections. The attention coefficient $a_{i,j,p}$, calculated in Eq. 7, is used to calculate the output of the operation attention block as follows:

$$h_{O_{i,j}}^{(L+1)} = \sigma \left( \sum_{p=j-1}^{j+1} a_{i,j,p} \mathbf{W}h_{O_{i,p}}^{(L)} \right), \tag{8}$$

where $\sigma$ is an activation function. The operation blocks in DAN (Wang et al., 2023) function similar to a GNN, in that information, one by one, is propagated through the operations.

**Machine Attention Block.** The machine attention block considers the relationship between two machines $M_y \in \mathcal{M}_t$ and $M_z \in \mathcal{M}_t$ in relation to the set of unscheduled operations $\hat{O}_{y,z}$ that can be processed by either $M_y$ or $M_z$. The embedding of the pooled operation is calculated as $h_{\hat{O}_{y,z}}^{(L)} = \frac{1}{|\hat{O}_{y,z}|} \sum_{O_{i,j} \in \hat{O}_{y,z} \cap \mathcal{O}_c} h_{O_{i,j}}^{(L)}$, where $\mathcal{O}_c$ represents the current operations available to schedule. The attention in this block is calculated through:

$$u_{y,z} = \text{Softmax}\Big(\text{LeakyReLU}\Big(\mathbf{X}\Big[(\mathbf{Y}h_{M_y}^{(L)}) \parallel (\mathbf{Y}h_{M_z}^{(L)}) \parallel (\mathbf{Z}h_{\hat{O}_{y,z}}^{(L)})\Big]\Big)\Big) \tag{9}$$

where $\mathbf{X}$, $\mathbf{Y}$, and $\mathbf{Z}$ are linear projections. Whenever two machines $M_y$ and $M_z$ do not share any operations in the current candidate set $\hat{O}_{y,z} \cap J_c = \emptyset$, we set the attention $u_{y,z}$ to zero. The output of the machine operation block is calculated as:

$$h_{M_k}^{(L+1)} = \sigma\left(\sum_{q \in \mathcal{N}_k} u_{k,q}\mathbf{Y}h_{M_q}^{(L)}\right), \tag{10}$$

where $\mathcal{N}_k$ is the set of machines, for which $M_k$ shares operations, including $M_k$ itself.

Lastly, DAN (Wang et al., 2023) uses a multihead attention approach, whereby each operation attention and machine attention block consist of $H$ heads. The results of the $H$ heads can be concatenated or averaged. Following the prior work of Wang et al. (2023), we concatenate the heads for each layer, except the last layer, which was averaged over the $H$ heads. We use ELU as our activation function for both operation and machine attention blocks.

### B.1 Features

Table 8 shows the features used in our paper, based on the prior work of Wang et al. (2023). Both the machine features $M_k$ and the operation features $O_{i,j}$ are embedded using the DAN network. These embeddings, with the machine-operation pair $(O_{i,j}, M_k)$ features are used as input for the quantile critic and actor networks. In Table 8, we introduce the notation $\mathcal{O}_k$, which represents all operations $O_{i,j} \in \mathcal{O}_k$ that $M_k$ can process.

## C Benchmark Instance Sets

As described in Sect.5, we evaluate our approach on generated instance sets as well as four established benchmark sets. For FJSP, we use the generated evaluation instances, the Brandimarte (mk) benchmark (Brandimarte, 1993) and the Hurink benchmark (Hurink et al., 1994), which includes the edata, rdata, and vdata subsets. For JSP, we evaluate on the Taillard (Taillard, 1993) and Demirkol (Demirkol et al., 1998) benchmarks. For each benchmark, we report the range of processing times, number of jobs, number of machines, and, specifically for FJSP, the number of machines available per operation.

### C.1 FJSP

**Generated Evaluation Instances.** We generated 100 instances for each of the following sizes: $10 \times 5, 15 \times 10, 20 \times 10, 30 \times 10, 40 \times 10$, using the same generation procedure as for the training data (Sect. 5). Each operation is assigned between 1 and $|\mathcal{M}|$ available machines, selected uniformly at random.

**Brandimarte (mk) Benchmark.** The Brandimarte benchmark (Brandimarte, 1993) comprises 10 instances, each with 10 to 20 jobs and 4 to 15 machines. Processing times range from 1 to 19. The average number of machines available per operation ranges from 1.4 to 4.1, depending on the instance.

**Hurink Benchmark.** The Hurink benchmark (Hurink et al., 1994) consists of three subsets, edata, rdata, and vdata, each containing 40 instances. These subsets vary in degree of flexibility, with edata providing the lowest and vdata the highest average number of machines per operation. All instances include between 7 and

Table 8: Features used by CDQAC, separated by operation $O_{i,j}$, machine $M_k$, and machine-operation pair $(O_{i,j}, M_k)$.

| Feature | Description |
|---|---|
| Operation Features $O_{i,j}$ | |
| Min. proc. time | $\min_{M_k \in \mathcal{M}_{i,j}} p_{i,j}^k$ |
| Mean proc. time | $\frac{1}{|\mathcal{M}_{i,j}|} \sum_{M_k \in \mathcal{M}_{i,j}} p_{i,j}^k$ |
| Span proc. time | $\max_{M_k \in \mathcal{M}_{i,j}} p_{i,j}^k - \min_{M_k \in \mathcal{M}_{i,j}} p_{i,j}^k$ |
| Compatibility ratio | $\frac{|\mathcal{M}_{i,j}|}{|\mathcal{M}|}$ |
| Scheduled | 1 if scheduled, 0 otherwise |
| Estimated LB | Estimated lower bound completion time $C(O_{i,j})$ |
| Remaining ops $J_i$ | Number of unscheduled operations in $J_i$ |
| Remaining proc. time $J_i$ | Total proc. time of unscheduled operations in $J_i$ |
| Waiting time | Time since $O_{i,j}$ became available |
| Remaining proc. time | Remaining processing time (0 if not started) |
| Machine Features $M_k$ | |
| Min. proc. time | $\min_{O_{i,j} \in \mathcal{O}_k} p_{i,j}^k$ |
| Mean proc. time | $\frac{1}{|\mathcal{O}_k|} \sum_{O_{i,j} \in \mathcal{O}_k} p_{i,j}^k$ |
| Total unscheduled ops | $|\mathcal{O}_k|$ |
| Schedulable ops at $t$ | # of ops schedulable at timestep $t$ |
| Free time | Time until $M_k$ becomes available |
| Waiting time | 0 if $M_k$ is working |
| Working status | 1 if working, 0 otherwise |
| Remaining proc. time | Time left on current task (0 if idle) |
| Machine-Operation Pair $(O_{i,j}, M_k)$ | |
| Processing time | $p_{i,j}^k$ |
| Ratio to max of $O_{i,j}$ | $\frac{p_{i,j}^k}{\max_{M_k} p_{i,j}^k}$ |
| Ratio to max schedulable on $M_k$ | $\frac{p_{i,j}^k}{\max p_{i,j}^k \in \mathcal{O}_k(t)}$ |
| Ratio to global max | $\frac{p_{i,j}^k}{\max p_{i,j}^k \in \mathcal{O}}$ |
| Ratio to $M_k$'s unscheduled max | $\frac{p_{i,j}^k}{\max p_{i,j}^k \in \mathcal{O}_k}$ |
| Ratio to compatible max | $\frac{p_{i,j}^k}{\max p_{i,j}^k \in \mathcal{M}_{i,j}}$ |
| Ratio to $J_i$ workload | $\frac{p_{i,j}^k}{\sum p_{i,j} \in J_i}$ |
| Joint waiting time | Sum of $O_{i,j}$ and $M_k$ waiting times |

30 jobs and between 4 and 15 machines, with processing times between 5 and 99. The average number of machines available per operation is as follows: **edata:** Between 1.13 and 1.2, **rdata:** Between 1.88 and 2.06, and **vdata:** Between 2.38 and 6.7.

## C.2 JSP

**Taillard Benchmark.** The Taillard benchmark (Taillard, 1993) contains 80 instances, ranging from $15 \times 15$ to $100 \times 20$. Processing times range between 1 and 99. These instances are similar to those used to train CDQAC.

**Demirkol Benchmark.** The Demirkol benchmark (Demirkol et al., 1998) includes 80 instances, with instance sizes ranging from $20 \times 15$ to $80 \times 20$. Processing times range from 1 to 200, twice the maximum value found in Taillard and CDQAC's training data.

## D Details of Dataset Generation Heuristics

Our experimental setup in Sect. 5 stated that we used three types of heuristics to generate our training datasets, namely, priority dispatching rules (PDR), genetic algorithms (GA) and a random policy. We will now give a detailed explanation of each heuristic, and, in the case of GA, the hyperparameters.

### D.1 Priority Dispatching Rules (PDR)

For the priority dispatching rules (PDR), we have separate rules for the selection of *jobs* and *machines* for FJSP. In our setup, first, a job $J_i \in \mathcal{J}$ is selected by the job selection rule. This job selection rule selects a

job based on a specific rule, in which it is checked if there are still operations in $J_i$ to be scheduled. The machine selection rule selects the machine $M_k \in \mathcal{M}_{i,j}$ for operation $O_{i,j} \in J_i$, where $O_{i,j}$ is the current operation in $J_i$ that needs to be scheduled. For JSP, we only considered the job selection rules, since only one machine is ever available per operation. Furthermore, both the job and machine selection rules follow the MDP formulation, stated in Sect. 2, by which operation $O_{i,j}$ can only be scheduled on $M_k$, if it is free at timestep $t$. In the following, we give an overview of the job selection rules and the machine selection rules.

**Job selection rules.** We utilized four different job selection rules, namely, *Most Operations Remaining* (MOR), *Least Operations Remaining* (LOR), *Most Work Remaining* (MWR), and *Least Work Remaining* (LWR). Both MOR and LOR decide on the basis of the number of unscheduled operations in a job $J_i$. MOR selects the job with the most operations and LOR selects the job with the least operations to be scheduled. MWR and LWR focus on the remaining total processing times, a.k.a. the summation of processing times in a $J_i$, whereby we average the processing times of the available machines $M_k \in \mathcal{M}_{i,j}$. MWR selects the job with the highest total remaining processing times, whereas LWR selects the job with the least.

**Machine selection rules.** We considered four different machine selection rules, namely, *Shortest Processing Time* (SPT), *Longest Processing Time* (LPT), *Earliest Start Time* (EST), and *Latest Start Time* (LST). Both SPT and LPT select a machine $M_k \in \mathcal{M}_{i,j}$ for operation $O_{i,j}$ based on the processing time, with SPT selecting the machine with the shortest and LPT with the longest. EST and LST consider how long a machine $M_k$ is already free, with EST selecting the machine that is free the shortest, and LST the longest.

### D.2 Genetic Algorithms (GA)

For our genetic algorithm (GA), we used the implementation of Reijnen et al. (2023), whereby we introduced the constraint that $O_{i,j}$ can only be scheduled if machine $M_k$ is free at that time. This results in a more tight solution, with no gaps. Furthermore, we used a population size of 200, and ran the GA for 100 generations. The crossover probability was set at 0.7, and the mutation probability at 0.2.

### D.3 Random Policy

The random policy adheres to the MDP introduced in Sect. 2. This means that the random policy selects a random machine-operation pair based on those available at the time step $t$. The random policy can only select a machine-operation pair, if it can be scheduled at timestep $t$.

## E Details of Offline Reinforcement Learning Baselines

For our comparison of CDQAC to Offline-LD (Remmerden et al., 2025) in Sect. 5.1, we adapted both versions of it, namely, Offline-LD with a maskable Quantile Regression DQN (mQRDQN) and with a discrete maskable Soft Actor-Critic (d-mSAC), using a dual attention network (Wang et al., 2023), such that both versions of Offline-LD used the same encoding as our introduced CDQAC approach. We provide a brief explanation of our implementations of each method, in which we state the hyperparameters used for each. If a hyperparameter is not stated, it is the same as CDQAC, as stated in App. G.

**Offline-LD (mQRDQN).** The mQRDQN version of Offline-LD is implemented identically as described by Remmerden et al. (2025). The hyperparameters are identical to CDQAC, whereby we set $\ell_\theta = 2 \times 10^{-4}$. In the original implemented of Offline-LD (mQRDQN) was not able to sample actions; therefore, for the sampling evaluation, we use Boltzmann sampling.

**Offline-LD (d-mSAC).** For d-mSAC version of Offline-LD, we implemented both the policy network and the Q network with a separate dual attention network (Wang et al., 2023) for each. We used the hyperparameters as with CDQAC, except for $\alpha_{\mathrm{CQL}}$, which we set to $\alpha_{\mathrm{CQL}} = 0.1$, and the target entropy of d-mSAC, which we set to 0.3. During initial testing, we found that this increased stability and performance with d-mSAC.

**Implicit Q-learning.** The main difference between Implicit Q-learning (IQL) (Kostrikov et al., 2021) and Offline-LD and CDQAC is that IQL constrains training by not using OOD actions, whereas Offline-LD and CDQAC regularize the Q-values of OOD actions during training to prevent overestimation. IQL consists of three networks, a policy, a value, and a Q network. Two hyperparameters of IQL are important to mention, namely $\beta_{\text{IQL}}$ and $\tau_{\text{IQL}}$. Firstly, $\beta_{\text{IQL}}$ controls how much the policy should learn to "exploit" the learned Q-values, or if it should stay close to the behavior found in the dataset, with $\beta_{\text{IQL}} = 0$, being equal to behavioral cloning. We decided, due to the suboptimality of our training datasets, to set $\beta_{\text{IQL}} = 15$. $\tau_{\text{IQL}}$ controls how much IQL should focus on positive examples, whereby $\tau_{\text{IQL}} = 0.5$ is equal to a SARSA update. Kostrikov et al. (2021) reported settings between 0.7 and 0.9 for $\tau_{\text{IQL}}$. We therefore tested 0.7, 0.8 and 0.9 to identify the ideal value and found that $\tau_{\text{IQL}} = 0.7$ result in the most stable updates. We set all learning rates at $2 \times 10^{-4}$, by which we also tested $2 \times 10^{-5}$; however, we found that this did not produce good results.

**Behavioral Cloning** Behavioral Cloning (BC) learns to imitate the behavioral policy $\pi_\beta$, which generated the training dataset. The BC loss is the cross-entropy loss between the predicted action for each state and the action found in the dataset. BC only trains a policy network and does not use a critic. All hyperparameters are the same as CDQAC (Table 11).

**TD3-AWR.** TD3-AWR (Jackson et al., 2025) is a model-free offline RL algorithm from the *Unifloral* framework that combines the ReBRAC critic (Tarasov et al., 2023)—the offline, behavior-regularized variant of TD3 (Fujimoto & Gu, 2021), using twin critics, a delayed target-smoothed policy, and a behavior-cloning penalty on the critic target—with an advantage-weighted regression (AWR) actor (Peng et al., 2019; Kostrikov et al., 2021) that weights each cloned action by its estimated advantage. As with our other baselines, we use the dual attention network (Wang et al., 2023) as the shared encoder, so that only the offline-RL objective differs from CDQAC; TD3-AWR replaces CDQAC's dueling quantile critic with two scalar critics and adds a value network $V_\phi$ for the AWR weight. Because TD3-style target smoothing perturbs a continuous action, which is undefined for the discrete, variable-size action set $\mathcal{A}(s_t)$ of FJSP, we adapt it with the unified action rule of Fujimoto et al. (2025): the policy scores each available machine–operation pair, we add clipped Gaussian noise to this score vector, and select the target action by an argmax over $\mathcal{A}(s')$, so feasibility is preserved by construction without a separate action mask. The actor uses the Gumbel-softmax relaxation of Fujimoto et al. (2025) to keep its deterministic-policy-gradient term differentiable, greedy deployment reduces to the argmax over $\mathcal{A}(s_t)$, and the behavior-cloning terms use the cross-entropy to the dataset action. TD3-AWR uses no CQL penalty, controlling out-of-distribution actions through its critic- and actor-side behavior-cloning terms instead. We set the AWR inverse temperature $\beta_{\text{AWR}} = 15$, the value expectile $\tau_{\text{AWR}} = 0.7$, the policy noise $\sigma = 0.2$ with clip $c = 0.5$, and the actor and critic behavior-cloning coefficients to 0.1 and 0.01; all remaining hyperparameters are the same as CDQAC.

## F Expanded Explanation of Results

Sect. 5.2 argued that two structural properties of scheduling—a dense reward aligned exactly with the makespan objective and equal-length trajectories across all constructive heuristics—place offline RL in a regime where state–action coverage, rather than trajectory quality, governs performance, and supported this with size-averaged dataset metrics and their Spearman correlations against CDQAC's performance (Table 2). This appendix expands that analysis in three directions. App. F.1 details how the metrics are computed and reports them at each training size. App. F.2 develops the mechanism that connects coverage to performance through pessimism, intrinsic uncertainty, and trajectory stitching. App. F.3 presents a controlled intervention that separates the effect of coverage from a possible intrinsic affinity of CDQAC for uniformly random data.

### F.1 State–Action Coverage and Local Support Density

We characterize each dataset using the three measures introduced in Sect. 5. State–Action Coverage (SACo) measures the breadth of the observed state–action set. Let $u_{s,a}(\mathcal{D})$ denote the number of unique state–action pairs in dataset $\mathcal{D}$. We compute

$$\text{SACo}(\mathcal{D}) = \frac{u_{s,a}(\mathcal{D})}{u_{s,a}(\mathcal{D}_{\text{PDR}})}, \tag{11}$$

Table 9: State–action coverage, trajectory quality, and local support density proxy of the FJSP training datasets at each training-instance size. SACo is normalized by the corresponding PDR dataset, with higher values indicating broader coverage (↑). Trajectory Quality (TQ) normalizes each trajectory's return between the worst (0) and best (1) observed return per instance, with higher values indicating higher-quality solutions (↑). The reference-policy-anchored $k$-NN distance uses $k = 10$, with lower values indicating denser support around high-quality reference trajectories (↓). Size-averaged values and their Spearman correlations with CDQAC's performance are reported in Table 2 in the main text. **Bold** indicates the best result for each metric and instance size.

| Instance size | Metric | PDR | GA | PDR-GA | Random |
|---|---|---|---|---|---|
| | SACo (↑) | $1.00 \pm 0.00$ | $3.13 \pm 0.38$ | $4.13 \pm 0.38$ | $\mathbf{8.46 \pm 0.71}$ |
| $10 \times 5$ | TQ (↑) | $0.563 \pm 0.065$ | $\mathbf{0.898 \pm 0.029}$ | $0.811 \pm 0.036$ | $0.591 \pm 0.059$ |
| | $k$-NN distance (↓) | $0.0459 \pm 0.0045$ | $0.0406 \pm 0.0035$ | $\mathbf{0.0391 \pm 0.0033}$ | $0.0405 \pm 0.0036$ |
| | SACo (↑) | $1.00 \pm 0.00$ | $2.59 \pm 0.46$ | $3.59 \pm 0.46$ | $\mathbf{6.93 \pm 0.29}$ |
| $15 \times 10$ | TQ (↑) | $0.589 \pm 0.040$ | $\mathbf{0.883 \pm 0.021}$ | $0.796 \pm 0.026$ | $0.604 \pm 0.059$ |
| | $k$-NN distance (↓) | $0.0279 \pm 0.0015$ | $0.0257 \pm 0.0016$ | $\mathbf{0.0247 \pm 0.0014}$ | $0.0259 \pm 0.0017$ |
| | SACo (↑) | $1.00 \pm 0.00$ | $3.16 \pm 0.40$ | $4.16 \pm 0.40$ | $\mathbf{7.70 \pm 0.18}$ |
| $20 \times 10$ | TQ (↑) | $0.603 \pm 0.050$ | $\mathbf{0.896 \pm 0.024}$ | $0.816 \pm 0.026$ | $0.557 \pm 0.061$ |
| | $k$-NN distance (↓) | $0.0247 \pm 0.0013$ | $0.0243 \pm 0.0015$ | $\mathbf{0.0226 \pm 0.0012}$ | $0.0254 \pm 0.0016$ |

where PDR is the reference dataset at the corresponding instance size, so that $\mathrm{SACo}(\mathcal{D}_{\mathrm{PDR}}) = 1$ and larger values indicate broader empirical coverage. Trajectory Quality (TQ) normalizes the return of each trajectory between the worst (0) and best (1) return observed on its instance, as defined in Sect. 5.

Coverage and quality alone do not establish whether a dataset provides sufficient local support around the states used by a high-quality policy. We therefore complement them with a representation-space support-density measure. For each of 100 analyzed instances at every training size, we obtain a high-quality reference trajectory by retaining the best solution from 1,000 stochastic CDQAC rollouts. Each transition is represented as

$$e_t = [h_G(s_t) \,\|\, h_G(s_{t+1})], \tag{12}$$

where $h_G$ is the global state representation produced by a frozen DAN encoder and $\|$ denotes concatenation. The same encoder is used for all four dataset sources at a given instance size, and the embeddings are normalized before computing distances. For every reference transition, we calculate its mean Euclidean distance to the $k = 10$ nearest transitions from each dataset source. The resulting distances are averaged over the reference trajectory and subsequently over instances. A smaller distance indicates denser local support around the high-quality reference behavior.

Table 9 shows that the ordering reported in Sect. 5 holds at every individual training size, not merely on average. PDR has the narrowest support because it contains trajectories from a small collection of deterministic dispatching rules. GA explores a broader region but concentrates its transitions around the high-quality solutions discovered by its evolutionary search, which is also reflected in its uniformly highest TQ (0.878–0.898 across sizes). PDR-GA combines these two sources and consequently covers more state–action pairs than either source alone. Random attains the highest SACo at every size, between 6.93 and 8.46 times the coverage of PDR, while its TQ is on par with PDR (0.570–0.599). For local support, PDR-GA has the smallest $k$-NN distance at every size, which is expected because it contains the union of the PDR and GA transitions. Random nevertheless remains close despite its substantially broader coverage: averaged across sizes, the $k$-NN distances are 0.0288 for PDR-GA, 0.0302 for GA, 0.0306 for Random, and 0.0328 for PDR.

These per-size results distinguish coverage from local support density. Random is only the third densest dataset, but its breadth does not come at the cost of a severe broad-but-sparse failure mode around the high-quality reference trajectories. Conversely, PDR-GA provides the greatest local density but covers only approximately half as many unique state–action pairs as Random. This separation is consistent with the correlation analysis in Table 2, where SACo is the only metric significantly correlated with CDQAC's performance while the density proxy is uncorrelated and TQ is negatively but not significantly correlated. Broad coverage therefore appears to be the operative dataset property, with adequate local support acting as a complementary precondition rather than as the driver.

**Connection to pessimism.** A conservative offline learner assigns low values to actions whose outcomes are weakly supported by the dataset. Narrow datasets may therefore leave many potentially useful actions pessimistically valued because the learner has insufficient observations with which to distinguish genuinely poor actions from merely unobserved ones. Broader datasets provide observed counterfactual transitions that allow these estimates to be grounded in data (Jin et al., 2021; Kumar et al., 2022). Random and PDR-GA contain more such counterfactual actions than PDR or GA alone, while the density analysis shows that this broader coverage does not leave the high-quality regions severely unsupported.

This interpretation also depends on the structure of the scheduling reward. Let

$$M(s_t) = \max_{O_{i,j} \in \mathcal{O}} C(O_{i,j}, s_t) \tag{13}$$

denote the partial makespan. The reward is

$$r_t = M(s_t) - M(s_{t+1}) \le 0. \tag{14}$$

Consequently,

$$\sum_{t=0}^{H-1} r_t = M(s_0) - M(s_H), \qquad C_{\max} = M(s_H) = M(s_0) - \sum_{t=0}^{H-1} r_t. \tag{15}$$

Maximizing the undiscounted return is therefore equivalent to minimizing the final makespan: every increase in the partial makespan produces an objective-aligned learning signal against which counterfactual actions can be valued. The second precondition, equal trajectory length, does not imply that different heuristics visit the same states; it ensures that both weak and strong heuristics generate complete trajectories containing transitions from every decision depth. The ablations in Table 3, discussed in Sect. 5, test both preconditions directly: when either is broken, the advantage of Random shrinks or reverses, showing that coverage alone is insufficient when complete trajectories or an objective-aligned learning signal are unavailable.

### F.2 Intrinsic Uncertainty and Trajectory Stitching

The broader support of Random and PDR-GA can reduce intrinsic uncertainty: uncertainty arising because transitions needed by a high-quality policy are absent from the offline dataset (Jin et al., 2021). No practically sized dataset is likely to contain a complete optimal trajectory. Under a simplified uniform behavior policy, the probability of reproducing a particular trajectory is

$$P(\tau^\star) = \prod_{t=0}^{H-1} \frac{1}{|\mathcal{A}(s_t)|}, \tag{16}$$

which decreases exponentially with the horizon. Conditional on reaching a particular state $s_t$, however, observing a particular feasible action has probability $1/|\mathcal{A}(s_t)|$. It is therefore considerably more plausible for a dataset to contain useful transitions from many different trajectories than to contain one complete optimal trajectory.

Value-based offline RL can combine these transitions through trajectory stitching. Consider two suboptimal trajectories that reach the same state, or states mapped to similar representations by the learned encoder:

$$\tau_1 = s_{\text{good start}} \to \cdots \to s_{\text{intersect}} \to \cdots \to s_{\text{poor end}},$$
$$\tau_2 = s_{\text{poor start}} \to \cdots \to s_{\text{intersect}} \to \cdots \to s_{\text{good end}}.$$

Although neither trajectory is globally strong, the prefix of $\tau_1$ and suffix of $\tau_2$ may both be valuable. Bellman updates propagate the value of the strong suffix back to the shared region:

$$Q(s_t, a_t) \leftarrow r_t + \gamma \mathbb{E}_{a' \sim \pi_\psi(\cdot | s_{t+1})} \left[ Q(s_{t+1}, a') \right]. \tag{17}$$

The learned policy can consequently select a prefix represented in one trajectory and a continuation represented in another, producing a schedule that does not occur as a complete trajectory in the dataset.

Table 10: Supporting coverage intervention using $\varepsilon$-PDR datasets. CDQAC is trained at all three instance sizes, $10 \times 5$, $15 \times 10$, and $20 \times 10$, and the results are averaged across these sizes. SACo is measured relative to PDR ($\uparrow$), Trajectory Quality (TQ, $\uparrow$) and the embedding-space $k$-NN density proxy ($k$=10, $\downarrow$) are averaged over the three training sizes, and gaps are reported as the mean $\pm$ standard deviation over four seeds ($\downarrow$). The $p$-values are obtained from Wilcoxon signed-rank comparisons with Random on per-instance gaps. **Bold** indicates the best gap, coverage, TQ, and density, or $p < 0.05$ in the final column.

| Dataset | Greedy gap (%) $\downarrow$ | Sampling gap (%) $\downarrow$ | SACo $\uparrow$ | TQ $\uparrow$ | $k$-NN dist. $\downarrow$ | $p$ vs. Random |
|---|---|---|---|---|---|---|
| Random | $10.68 \pm 0.51$ | $5.87 \pm 0.30$ | $7.70 \pm 0.77$ | $0.58 \pm 0.02$ | $0.031 \pm 0.009$ | — |
| $\varepsilon$-PDR ($\varepsilon = 0.1$) | $10.97 \pm 0.42$ | $5.98 \pm 0.23$ | $6.82 \pm 0.56$ | $0.64 \pm 0.03$ | $0.028 \pm 0.009$ | **0.006** |
| $\varepsilon$-PDR-GA ($\varepsilon = 0.1$) | $10.76 \pm 0.47$ | $5.86 \pm 0.32$ | $7.83 \pm 0.68$ | $\mathbf{0.68 \pm 0.02}$ | $\mathbf{0.027 \pm 0.009}$ | $0.364$ |
| $\varepsilon$-PDR ($\varepsilon = 0.2$) | $10.87 \pm 0.53$ | $5.93 \pm 0.62$ | $7.37 \pm 0.61$ | $0.63 \pm 0.01$ | $0.028 \pm 0.009$ | $0.054$ |
| $\varepsilon$-PDR-GA ($\varepsilon = 0.2$) | $\mathbf{10.45 \pm 0.36}$ | $\mathbf{5.81 \pm 0.32}$ | $\mathbf{8.38 \pm 0.72}$ | $0.66 \pm 0.01$ | $\mathbf{0.027 \pm 0.008}$ | $0.209$ |

The behavior-policy results in Table 1 provide direct evidence that CDQAC improves beyond individual logged trajectories. For the Random dataset, the best solution produced by the data-generating policy has an average gap of 28.16%, whereas a single greedy rollout of CDQAC achieves 10.68%. Under sampling, CDQAC also matches or improves upon the best data-generating policy for all four dataset sources. These results are consistent with trajectory stitching: the learned policy is not limited to reproducing the quality of any complete trajectory generated by the behavior policy.

The GA dataset contains high-quality schedules but concentrates its transitions around a comparatively narrow search region. If a useful alternative action is absent around a decision state, conservative offline learning has no evidence with which to assign that action a high value. The broader Random and PDR-GA datasets provide more alternative continuations, increasing the number of transition segments that can be combined. The density results in App. F.1 qualify this explanation: Random is successful not merely because it covers more transitions, but because this increased breadth is accompanied by local support close to the reference trajectories.

## F.3 Controlled Supporting Evidence

The preceding analyses establish a correlation between broad coverage and stronger offline performance, but they do not by themselves rule out an algorithmic affinity between CDQAC and uniformly random data—for instance, that its conservative regularization or quantile critic cooperates well with uniform action sampling specifically. To examine this alternative, we construct $\varepsilon$-PDR datasets in which the behavior policy chooses a uniformly random feasible action with probability $\varepsilon$ and follows a PDR action with probability $1 - \varepsilon$.

We evaluate $\varepsilon \in \{0.1, 0.2\}$ under two configurations. The $\varepsilon$-PDR dataset contains 100 trajectories per instance and samples a new PDR rule at the beginning of every trajectory. The $\varepsilon$-PDR-GA dataset retains the 200 GA trajectories per instance and adds 50 $\varepsilon$-PDR trajectories. Unlike the purely Random dataset, both constructions preserve substantial heuristic structure within each trajectory.

The results in Table 10 are more consistent with a coverage explanation than with an intrinsic preference for purely random trajectories. The configuration with the largest coverage deficit relative to Random, $\varepsilon$-PDR with $\varepsilon = 0.1$, is also the only configuration that significantly underperforms Random ($p = 0.006$). Increasing the exploratory component reduces both the coverage deficit and the performance difference ($p = 0.054$ at $\varepsilon = 0.2$). Moreover, $\varepsilon$-PDR-GA with $\varepsilon = 0.2$ attains the highest SACo and the lowest mean gaps despite retaining the complete GA component. If CDQAC required random data per se, the retained heuristic structure should have hurt performance; instead, performance tracks coverage regardless of how that coverage is obtained. The intervention does not hold every dataset property fixed—in particular, the $\varepsilon$-PDR-GA datasets contain more trajectories than Random—so it should be interpreted as controlled supporting evidence rather than as identifying coverage as the sole causal factor. Together with the coverage and support-density analyses in App. F.1 and the correlations in Table 2, these results indicate that broad coverage is particularly useful when accompanied by adequate local support and complete, objective-aligned trajectories.

## G  Hyperparameters

In Table 11, we state the hyperparameters used in all our experiments. Furthermore, we used two layers of the DAN network, whereby we concatenated the output of each head for the first layer and averaged the heads for the second layer. Both the value stream $V_\theta$ and the advantage stream $A_\theta$, consist of three layers, each having 64 neurons. For each seed, we train for $200,000$ steps, with a batch size of 256. We normalize all features in the training dataset. We used ADAM (Kingma, 2014) optimizer.

Table 11: Hyperparameter settings CDQAC.

| Hyperparameter | Value |
| --- | --- |
| Policy Frequency Update $\eta$ | 4 |
| CQL Strength $\alpha_{\mathrm{CQL}}$ | 0.05 |
| Number of quantile fractions $N$ | 64 |
| Learning rate quantile critic $\ell_\theta$ | $2 \times 10^{-4}$ |
| Learning rate policy $\ell_\psi$ | $2 \times 10^{-5}$ |
| Target Update Frequency $\rho$ | 0.005 |
| Entropy Coefficient $\lambda$ | 0.005 |
| Batch Size | 256 |
| Training Steps | 200,000 |
| Network Parameters | |
| Layers DAN network | 2 |
| Output Dimension DAN | (32, 8) |
| Number of Heads $H$ | 4 |
| Hidden Dimension Quantile Critic $Z_\theta$ | 64 |
| Hidden Layers Quantile Critic $Z_\theta$ | 2 |
| Hidden Dimension Policy $\pi_\psi$ | 64 |
| Hidden Layers Policy $\pi_\psi$ | 2 |

## H  Training Plots

Fig. 6, Fig. 7, and Fig 8 show the training plots for all the methods used in our FJSP evaluations (Tables 1, 14, 15, and 16). In the figure, we note that CDQAC converges in significantly fewer steps than the 200,000 training steps used. For example, for $10 \times 5$ CDQAC requires around 10,000 steps according to Fig. 6, and around 25,000 training steps for $15 \times 10$ as seen in Fig. 7.

Based on the training plots, we can determine that CDQAC achieves the most stable training with the highest average Makespan in each evaluation step. The only exception is with the Random dataset for $20 \times 10$ (Fig. 8d), where both Offline-LD (d-mSAC) and IQL are more stable and have a higher evaluation at the last training step. However, CDQAC for all other datasets and training datasets. For example, Offline-LD (d-mSAC) cannot learn a policy with the PDR-GA dataset for $20 \times 10$ (Fig. 8c), and IQL with the PDR dataset for all training sizes. (Figs 6a, 7a and 8a). Lastly, we can notice for all training plots that CDQAC converges significantly faster than the other offline RL methods.

## I  Additional Results

### I.1  Ablation Study

Sect. 5.5 reports the ablation of CDQAC's four components — *CQL regularization*, *Quantile Critic*, *Dueling Architecture*, and *Delayed Policy Update* — averaged across instance sizes and evaluation modes. Table 12 expands the bars of Fig. 4 along two additional dimensions: the three training sizes ($10 \times 5$, $15 \times 10$, $20 \times 10$), and a separation between *in-distribution* performance (generated instances matching the training distribution) and *out-of-distribution* performance (the Brandimarte (Brandimarte, 1993) and Hurink (Hurink et al., 1994) benchmarks).

Two patterns emerge. First, removing the Quantile Critic or the CQL regularization increases the gap by only 1–5%, with little variation across instance sizes, consistent with the averages reported in Section 5.5. Second, the contributions of the Delayed Policy Update and the Dueling Architecture grow sharply with instance size: removing the Delayed Policy Update increases the gap by 11.7% at $10 \times 5$, 38.5% at $15 \times 10$, and 112.8% at

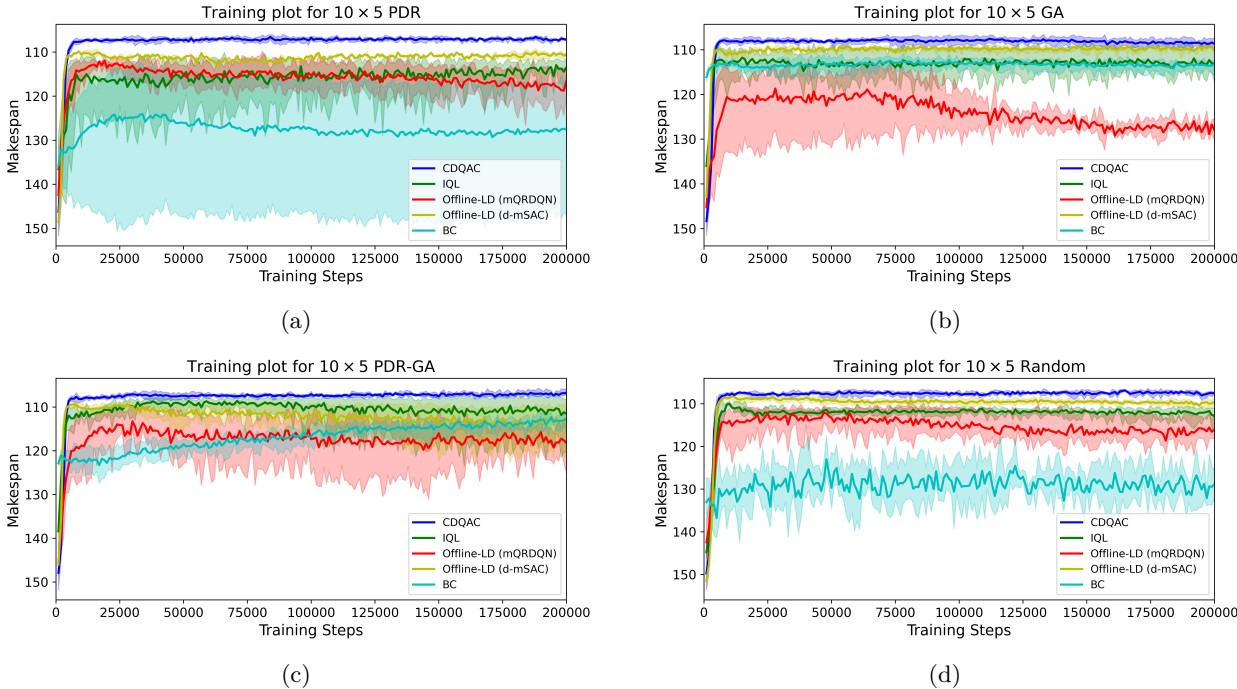

Figure 6: The training plots when trained of FJSP instances of size $10 \times 5$ for BC, CDQAC, IQL and Offline-LD, both mQRDQN and d-mSAC. Fig. 6a shows the training plots when trained on the PDR dataset, Fig. 6b with the GA dataset, Fig. 6c with the PDR-GA dataset, and Fig. 6d the Random dataset. The line is average makespan over four different seeds and the shaded area is minimal and maximal makespan of these seeds. We evaluate each method at every 1,000 steps of offline training.

Table 12: Ablation study of CDQAC components, trained on the Random dataset across instance sizes. We report the mean and standard deviation of the gap (%) across four seeds on In-Distribution instances (generated and similar to training set) and Out-of-Distribution FJSP benchmarks (Brandimarte and Hurink). Avg $\Delta(\%)$ is the mean percentage difference in gap relative to Standard (full CDQAC). **Bold** indicates best result per column within each instance size.

| | Component | In-Distribution (Gap %) | | Out-of-Distribution (Gap %) | | |
| | | Greedy | Sampling | Greedy | Sampling | Avg $\Delta(\%)$ |
|---|---|---|---|---|---|---|
| $10 \times 5$ | Standard | $\mathbf{11.19 \pm 0.35}$ | $\mathbf{5.87 \pm 0.14}$ | $\mathbf{10.45 \pm 0.39}$ | $\mathbf{6.05 \pm 0.10}$ | 0.00% |
| | w/o CQL | $11.49 \pm 0.26$ | $5.98 \pm 0.17$ | $10.59 \pm 0.27$ | $6.16 \pm 0.14$ | +1.93% |
| | w/o Quantile | $11.72 \pm 0.53$ | $6.05 \pm 0.11$ | $10.50 \pm 0.21$ | $6.24 \pm 0.14$ | +2.86% |
| | w/o Dueling | $11.59 \pm 0.53$ | $5.99 \pm 0.27$ | $10.97 \pm 0.43$ | $6.45 \pm 0.30$ | +4.30% |
| | w/o Delayed Policy | $12.27 \pm 0.49$ | $6.30 \pm 0.14$ | $12.46 \pm 1.12$ | $6.69 \pm 0.27$ | +11.70% |
| $15 \times 10$ | Standard | $\mathbf{12.04 \pm 0.59}$ | $6.70 \pm 0.62$ | $\mathbf{9.81 \pm 0.36}$ | $\mathbf{5.10 \pm 0.26}$ | 0.00% |
| | w/o CQL | $12.42 \pm 0.36$ | $7.03 \pm 0.76$ | $10.56 \pm 0.81$ | $5.25 \pm 0.26$ | +4.88% |
| | w/o Quantile | $12.35 \pm 0.28$ | $\mathbf{6.57 \pm 0.07}$ | $10.81 \pm 0.26$ | $5.47 \pm 0.08$ | +5.13% |
| | w/o Dueling | $12.42 \pm 0.31$ | $6.65 \pm 0.28$ | $10.94 \pm 0.47$ | $5.41 \pm 0.19$ | +5.77% |
| | w/o Delayed Policy | $13.13 \pm 0.43$ | $7.03 \pm 0.30$ | $17.24 \pm 1.96$ | $7.87 \pm 1.04$ | +38.46% |
| $20 \times 10$ | Standard | $\mathbf{5.20 \pm 0.66}$ | $2.87 \pm 0.73$ | $\mathbf{10.90 \pm 0.73}$ | $5.50 \pm 0.22$ | 0.00% |
| | w/o CQL | $5.32 \pm 0.70$ | $2.78 \pm 0.72$ | $11.18 \pm 1.00$ | $\mathbf{5.39 \pm 0.26}$ | +0.74% |
| | w/o Quantile | $5.44 \pm 0.30$ | $3.11 \pm 0.22$ | $11.28 \pm 0.54$ | $5.83 \pm 0.19$ | +4.68% |
| | w/o Dueling | $6.60 \pm 1.65$ | $\mathbf{2.66 \pm 0.34}$ | $13.30 \pm 1.99$ | $6.02 \pm 0.54$ | +16.73% |
| | w/o Delayed Policy | $15.37 \pm 5.51$ | $6.02 \pm 2.00$ | $21.31 \pm 4.61$ | $10.41 \pm 2.47$ | +112.75% |

$20 \times 10$, with the Dueling Architecture following the same trend on a smaller scale ($4.3\% \rightarrow 5.8\% \rightarrow 16.7\%$). The in-distribution / out-of-distribution split further shows that this scaling does not reflect overfitting: at $20 \times 10$, removing the Delayed Policy Update degrades both the in-distribution gap (from 5.20% to 15.37%) and the out-of-distribution gap (from 10.90% to 21.31%), indicating that without delayed updates CDQAC fails to train rather than fails to generalize. These findings reinforce the action-space argument in Section 5.5:

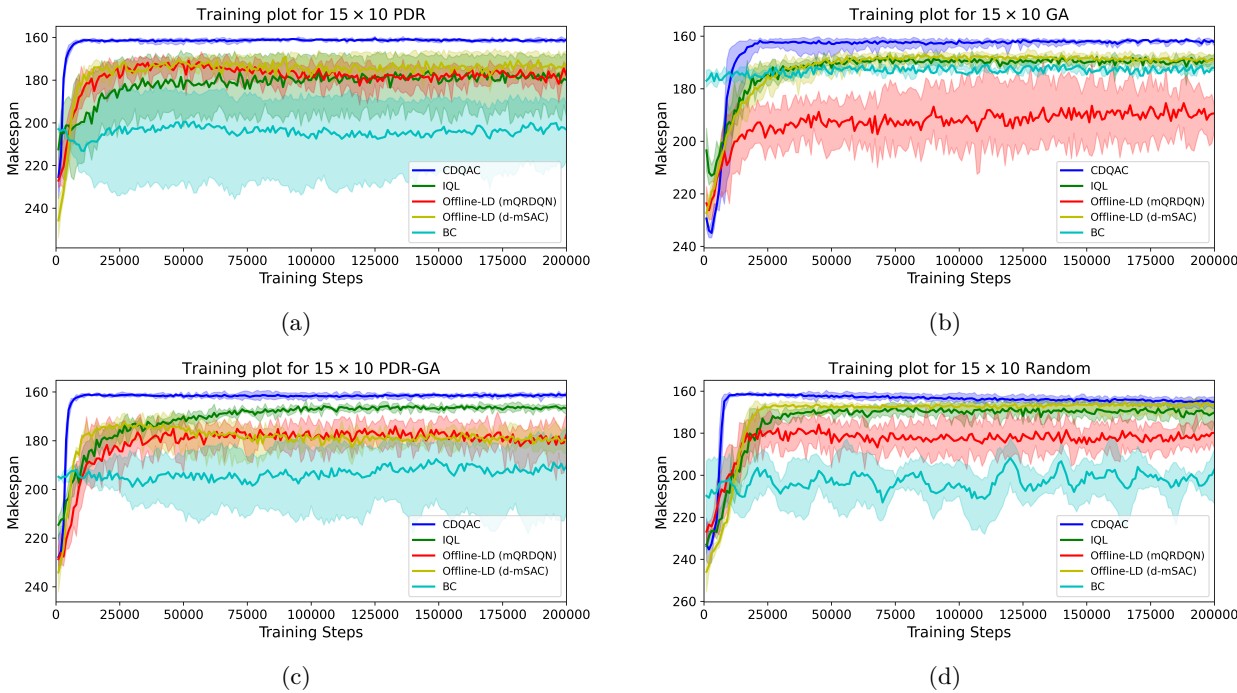

Figure 7: The training plots when trained of FJSP instances of size $15 \times 10$ for BC, CDQAC, IQL and Offline-LD, both mQRDQN and d-mSAC. Fig. 7a shows the training plots when trained on the PDR dataset, Fig. 7b with the GA dataset, Fig. 7c with the PDR-GA dataset, and Fig. 7d the Random dataset. The line is average makespan over four different seeds and the shaded area is minimal and maximal makespan of these seeds. We evaluate each method at every 1,000 steps of offline training.

Table 13: Sensitivity of CDQAC to the policy update frequency $\eta$ and the CQL strength $\alpha_{\text{CQL}}$, trained on the Random dataset ($10 \times 5$). We report the mean gap (%) on In-Distribution instances (generated and similar to training set) and Out-of-Distribution FJSP benchmarks (Brandimarte and Hurink). Avg $\Delta(\%)$ is the mean percentage difference in gap relative to the default configuration ($\eta = 4$, $\alpha_{\text{CQL}} = 0.05$). Results are averaged over four seeds. **Bold** indicates best result per column within each sweep.

| | | In-Distribution (Gap %) | | Out-of-Distribution (Gap %) | | |
|---|---|---|---|---|---|---|
| | Value | Greedy | Sampling | Greedy | Sampling | Avg $\Delta(\%)$ |
| Policy Delay $\eta$ | $\eta = 1$ | $12.27 \pm 0.56$ | $6.30 \pm 0.16$ | $12.46 \pm 1.29$ | $6.69 \pm 0.31$ | $+11.72\%$ |
| | $\eta = 2$ | $12.27 \pm 0.70$ | $6.30 \pm 0.40$ | $11.10 \pm 0.60$ | $6.39 \pm 0.27$ | $+7.19\%$ |
| | $\eta = 3$ | $11.67 \pm 0.45$ | $6.05 \pm 0.36$ | $10.69 \pm 0.28$ | $6.39 \pm 0.15$ | $+3.78\%$ |
| | $\eta = 4$ (default) | $\mathbf{11.19 \pm 0.41}$ | $\mathbf{5.87 \pm 0.16}$ | $\mathbf{10.45 \pm 0.46}$ | $\mathbf{6.05 \pm 0.12}$ | $0.00\%$ |
| | $\eta = 6$ | $11.78 \pm 0.72$ | $6.36 \pm 0.36$ | $10.49 \pm 0.59$ | $6.53 \pm 0.25$ | $+5.45\%$ |
| | $\eta = 8$ | $12.26 \pm 0.33$ | $6.36 \pm 0.23$ | $10.64 \pm 0.09$ | $6.53 \pm 0.27$ | $+6.87\%$ |
| $\alpha_{\text{CQL}}$ | $\alpha_{\text{CQL}} = 0$ | $11.49 \pm 0.30$ | $5.98 \pm 0.19$ | $10.59 \pm 0.31$ | $6.16 \pm 0.16$ | $+1.93\%$ |
| | $\alpha_{\text{CQL}} = 0.05$ (default) | $\mathbf{11.19 \pm 0.41}$ | $\mathbf{5.87 \pm 0.16}$ | $\mathbf{10.45 \pm 0.46}$ | $\mathbf{6.05 \pm 0.12}$ | $0.00\%$ |
| | $\alpha_{\text{CQL}} = 0.1$ | $11.36 \pm 0.59$ | $6.18 \pm 0.49$ | $10.81 \pm 0.24$ | $6.21 \pm 0.35$ | $+3.19\%$ |

the Delayed Policy Update and the Dueling Architecture are the two components responsible for keeping CDQAC stable as the combinatorial action space grows.

**Hyperparameter Sensitivity**   Table 13 reports a sensitivity analysis of the two hyperparameters most central to CDQAC's design: the policy update frequency $\eta$ and the CQL strength $\alpha_{\text{CQL}}$, trained on the Random dataset at $10 \times 5$. The sweep over $\eta$ exhibits a clear U-shape around the default $\eta = 4$. Updating the policy at every critic step ($\eta = 1$), which corresponds by construction to the *w/o Delayed Policy* ablation in Table 12, is the worst configuration ($+11.72\%$), with the degradation most pronounced in the out-of-distribution greedy evaluation ($12.46\% \pm 1.29$ vs. $10.45\% \pm 0.46$), where it is also accompanied by the largest seed variance in the sweep—both consistent with unstable premature policy improvement against a noisy

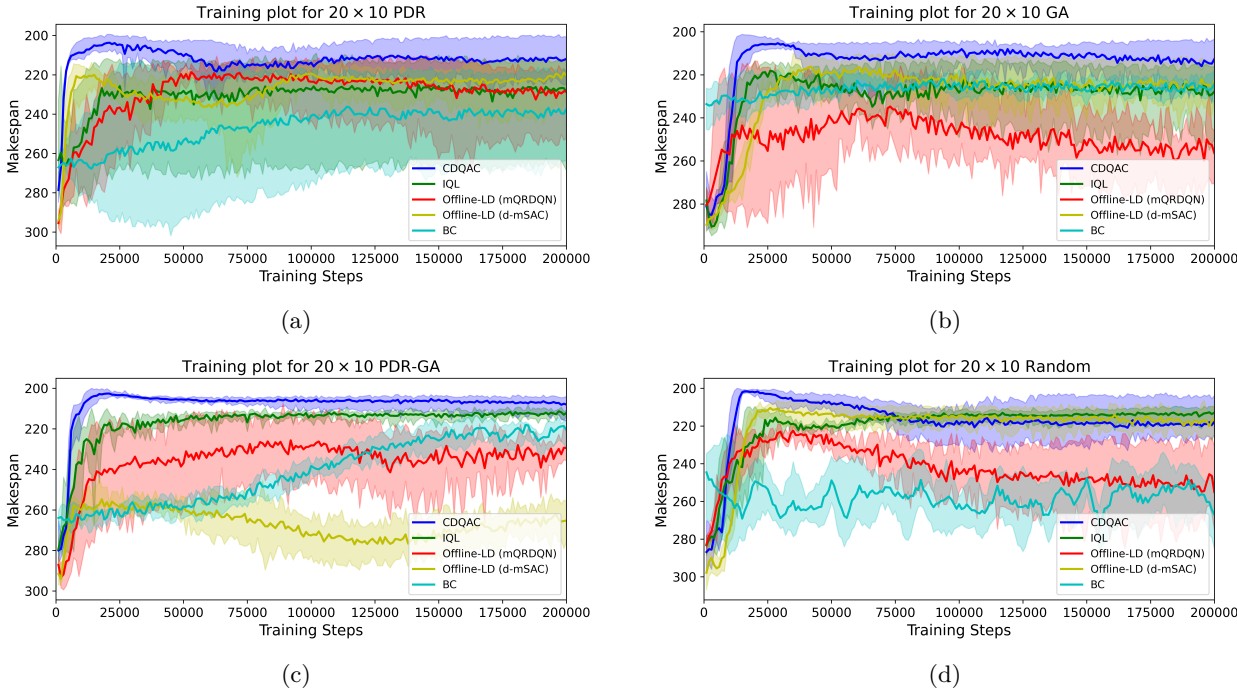

Figure 8: The training plots when trained of FJSP instances of size $20 \times 10$ for BC, CDQAC, IQL and Offline-LD, both mQRDQN and d-mSAC. Fig. 8a shows the training plots when trained on the PDR dataset, Fig. 8b with the GA dataset, Fig. 8c with the PDR-GA dataset, and Fig. 8d the Random dataset. The line is average makespan over four different seeds and the shaded area is minimal and maximal makespan of these seeds. We evaluate each method at every 1,000 steps of offline training.

critic. Increasing the delay beyond the default also degrades performance (+5.45% at $\eta = 6$, +6.87% at $\eta = 8$), with the loss concentrated in the in-distribution and sampling evaluations (e.g., $6.36\% \pm 0.23$ vs. $5.87\% \pm 0.16$ sampling at $\eta = 8$) rather than in out-of-distribution greedy performance. We attribute this to the actor–critic coupling in Eq. 2: because the Bellman target samples successor actions from the current policy, an overly stale actor in turn slows the critic's own convergence. Notably, even the largest tested delay remains clearly preferable to no delay at all, reinforcing the contrast with online continuous action spaces, drawn in Sect. 5.5, where delays beyond $\eta = 2$ are reported to be harmful (Fujimoto et al., 2019). The sweep over $\alpha_{\mathrm{CQL}}$ shows that CDQAC is comparatively insensitive to the conservatism strength: all tested values lie within 3.2% of the default, and the per-column differences fall largely within one standard deviation of one another, with removing the penalty entirely ($\alpha_{\mathrm{CQL}} = 0$, +1.93%) proving slightly less harmful than doubling it ($\alpha_{\mathrm{CQL}} = 0.1$, +3.19%). This mirrors the modest contribution of CQL observed in Table 12 at this instance size and suggests that on broad-coverage random data, where most of the relevant action space is in-support, only mild conservatism is needed, while over-penalization begins to restrict the very stitching of high-value segments that drives CDQAC's performance.

## I.2 Results Offline RL

In this section, we provide a comprehensive overview of the results discussed in Sect.5.1 and Table1, where we compare our proposed method, CDQAC, to Offline-LD (Remmerden et al., 2025). Table 1 presents the average performance across all evaluation instance sets—both generated and benchmark—for each training size ($10 \times 5$, $15 \times 10$, and $20 \times 10$). The detailed results for each evaluation set are reported in Table 14 (training size $10 \times 5$), Table 15 ($15 \times 10$), and Table 16 ($20 \times 10$).

As shown in Tables 14, 15, and 16, CDQAC consistently outperforms both versions of Offline-LD in nearly all evaluations. There are only a few exceptions: in Table 14, Offline-LD (d-mSAC) marginally exceeds

Table 14: Results of FJSP offline RL comparison $10 \times 5$, for all training datasets (PDR, GA, PDR-GA, and Random). The columns show the evaluation benchmarks sets and the rows the methods. The mean and standard deviation of the gap (%) are reported from four different seeds. **Bold** indicates best result (lowest gap) for either the Greedy and Sampling (100 solutions) evaluation, for a given training dataset.

| | Generated $10 \times 5$ | | Brandimarte (mk) | | Hurink edata | | Hurink rdata | | Hurink vdata | |
|---|---|---|---|---|---|---|---|---|---|---|
| | Greedy | Sampling | Greedy | Sampling | Greedy | Sampling | Greedy | Sampling | Greedy | Sampling |
| PDR | | | | | | | | | | |
| BC | 31.79±1.96 | 10.45±0.84 | 72.5±5.36 | 33.7±1.42 | 31.03±2.02 | 13.93±0.89 | 30.04±3.18 | 12.58±1.12 | 14.97±2.2 | 4.16±0.84 |
| Offline-LD (mQRDQN) | 15.4±1.2 | 14.39±0.12 | 22.81±3.76 | 25.07±0.27 | 25.54±2.4 | 12.38±0.06 | 18.74±2.55 | 10.24±0.09 | 11.77±1.11 | 3.37±0.05 |
| Offline-LD (d-mSAC) | 15.26±0.85 | 8.16±0.11 | 43.74±5.43 | 23.18±3.39 | 22.17±2.1 | 10.18±0.8 | 21.93±3.5 | 9.34±2.36 | 7.55±0.76 | 1.3±0.2 |
| IQL | 15.58±0.47 | 8.13±0.17 | 41.75±3.87 | 21.89±1.15 | 22.87±1.84 | 11.25±0.98 | 21.36±3.56 | 8.12±1.0 | 7.32±0.89 | 1.43±0.4 |
| TD3-AWR | 14.02±0.63 | 7.24±0.13 | 23.72±1.09 | 13.13±0.28 | 20.32±0.56 | **9.32±0.40** | 16.78±1.45 | 6.95±0.37 | 6.51±0.88 | 0.98±0.05 |
| **CDQAC** | **11.49±0.38** | **5.64±0.08** | **12.43±1.45** | **8.3±0.14** | **15.11±1.06** | 9.68±0.57 | **10.81±0.22** | **5.54±0.12** | **3.69±0.25** | **0.78±0.02** |
| GA | | | | | | | | | | |
| BC | 14.63±0.7 | 8.91±0.3 | 16.03±1.81 | 15.03±0.46 | 15.27±0.58 | 8.79±0.31 | 11.36±0.59 | 6.69±0.17 | 4.48±0.23 | 1.43±0.07 |
| Offline-LD (mQRDQN) | 17.28±3.88 | 14.52±0.08 | 33.45±8.26 | 26.62±0.62 | 29.64±3.0 | 12.55±0.07 | 22.84±1.78 | 10.47±0.2 | 14.13±1.99 | 3.51±0.06 |
| Offline-LD (d-mSAC) | **11.38±0.64** | **5.29±0.1** | 23.47±3.33 | 12.05±1.37 | 21.55±3.24 | **9.23±1.23** | 16.32±2.16 | 5.99±0.47 | 11.37±1.92 | 2.89±1.02 |
| IQL | 13.02±0.86 | 7.32±0.18 | 26.71±1.01 | 14.57±0.71 | 25.67±2.1 | 10.72±0.58 | 17.37±1.81 | 6.75±0.31 | 10.69±1.76 | 2.14±0.62 |
| TD3-AWR | 11.57±0.79 | 6.37±0.42 | 24.98±1.31 | 12.95±1.55 | 22.27±1.75 | 10.23±0.57 | 18.18±1.56 | 6.69±0.11 | 11.59±1.48 | 3.07±0.24 |
| **CDQAC** | 11.62±0.35 | 6.09±0.22 | **15.51±1.0** | **9.58±0.76** | **14.87±0.25** | 9.45±0.54 | **10.44±0.4** | **5.39±0.2** | **3.24±0.3** | **0.65±0.01** |
| PDR-GA | | | | | | | | | | |
| BC | 16.79±1.13 | 8.86±0.08 | 57.26±4.68 | 27.0±1.77 | 24.37±1.15 | 11.17±0.3 | 28.38±1.03 | 10.99±1.29 | 15.72±1.5 | 3.18±1.17 |
| Offline-LD (mQRDQN) | 14.7±0.99 | 14.33±0.04 | 21.77±1.22 | 25.27±0.45 | 25.53±2.79 | 12.25±0.11 | 19.34±2.61 | 10.33±0.06 | 11.94±2.17 | 3.45±0.05 |
| Offline-LD (d-mSAC) | 12.1±0.65 | 5.9±0.48 | 19.49±2.67 | 11.17±0.68 | 19.04±1.61 | **8.82±0.61** | 13.27±0.84 | 5.58±0.31 | 7.59±1.86 | 1.32±0.32 |
| IQL | 12.4±0.24 | 7.22±0.09 | 33.13±4.61 | 19.43±2.27 | 26.44±3.42 | 11.69±1.26 | 21.42±3.37 | 7.83±0.67 | 11.24±1.83 | 2.06±0.3 |
| TD3-AWR | 11.42±0.59 | 6.34±0.26 | 22.11±1.63 | 11.63±0.14 | 20.82±3.50 | 8.89±1.03 | 15.68±1.68 | 6.02±0.24 | 6.38±2.13 | 1.05±0.16 |
| **CDQAC** | **11.16±0.43** | **5.88±0.37** | **14.24±1.23** | **8.79±0.74** | **15.3±0.57** | 9.84±0.38 | **10.96±0.56** | **5.51±0.16** | **3.59±0.31** | **0.72±0.03** |
| Random | | | | | | | | | | |
| BC | 25.59±2.86 | 14.91±0.05 | 31.74±2.78 | 26.95±0.2 | 22.12±1.46 | 12.26±0.06 | 17.16±2.35 | 10.48±0.17 | 7.98±2.22 | 3.46±0.08 |
| Offline-LD (mQRDQN) | 14.41±0.87 | 14.17±0.14 | 21.42±1.44 | 25.0±1.03 | 19.05±1.5 | 11.93±0.11 | 14.85±1.64 | 9.98±0.15 | 7.91±1.68 | 3.22±0.15 |
| Offline-LD (d-mSAC) | 13.29±0.45 | 6.26±0.27 | 16.62±0.6 | 9.49±0.37 | 16.12±1.43 | 8.24±0.32 | 12.13±0.99 | 5.67±0.23 | 4.14±0.74 | 0.87±0.08 |
| IQL | 15.64±1.2 | 8.98±0.15 | 33.11±5.9 | 18.73±1.42 | 26.91±4.2 | 11.5±1.29 | 17.65±3.0 | 7.5±0.87 | 12.85±5.68 | 2.75±1.45 |
| TD3-AWR | 12.48±0.16 | 7.59±0.47 | 16.28±1.05 | 11.55±0.47 | 14.77±0.56 | **8.00±0.24** | 11.60±1.15 | 5.88±0.25 | 3.85±1.04 | 0.98±0.09 |
| **CDQAC** | **11.19±0.35** | **5.87±0.14** | **13.78±0.78** | **8.67±0.21** | **14.53±0.41** | 9.54±0.39 | **10.4±0.36** | **5.3±0.22** | **3.1±0.22** | **0.68±0.03** |

CDQAC in the generated instances and Hurink edata using the sampling evaluation when trained on the GA dataset, as well as on Hurink edata with the sampling evaluation when both methods are trained on the Random dataset. Nevertheless, CDQAC shows better performance on the remaining evaluation sets for both the GA and Random training sets. Furthermore, with larger training sizes, $15 \times 10$ (Table 15) and $20 \times 10$ (Table 16), CDQAC consistently outperforms Offline-LD, and the performance margins widen as the instance size increases. These findings indicate that CDQAC scales more efficiently to larger instance sizes, and is generally an improvement over the offline RL baseline, Offline-LD.

Analyzing CDQAC's performance across different instance sizes and training datasets, we observe that for both $10 \times 5$ (Table 14) and $15 \times 10$ (Table 15), CDQAC achieves the worst performance when trained on the GA dataset across all evaluation sets. In contrast, for $20 \times 10$, CDQAC trained on the GA dataset achieves the best performance on generated instances (Greedy: 5.01%±0.28%), while training on PDR yields the worst results (Greedy: 9.38%±6.1%), accompanied by a high standard deviation. This higher standard deviation with PDR suggests instability during training, as one of the four runs did not train effectively. Additionally, we find that, when trained on GA, CDQAC struggles to generalize to unseen evaluation instances compared to when trained on more diverse datasets, such as Random and PDR-GA. This further supports the conclusion that training on a diverse set of examples is critical for strong generalization performance in offline RL for FJSP.

## I.3   Additional Results JSP

In Sect. 5.4, we compared CDQAC on the Taillard and Demirkol instances. The results in Table 7 included only CDQAC trained on the Random dataset for $10 \times 5$ instances. In this section, we show the results for the other training sets for both $10 \times 5$ (Table 17) and $15 \times 10$ (Table 18) instances.

Table 15: Results of FJSP offline RL comparison $15 \times 10$, for all training datasets (PDR, GA, PDR-GA, and Random). The columns show the evaluation benchmarks sets and the rows the methods. The mean and standard deviation of the gap (%) are reported from four different seeds. **Bold** indicates best result (lowest gap) for either the Greedy and Sampling (100 solutions) evaluation, for a given training dataset.

| | Generated $15 \times 10$ | | Brandimarte (mk) | | Hurink edata | | Hurink rdata | | Hurink vdata | |
|---|---|---|---|---|---|---|---|---|---|---|
| | Greedy | Sampling | Greedy | Sampling | Greedy | Sampling | Greedy | Sampling | Greedy | Sampling |
| PDR | | | | | | | | | | |
| BC | 36.31±3.88 | 13.58±0.88 | 58.23±13.17 | 30.47±3.21 | 28.66±6.05 | 12.13±1.48 | 23.93±2.1 | 8.57±1.13 | 10.38±1.4 | 2.04±0.49 |
| Offline-LD (mQRDQN) | 17.36±1.17 | 20.28±0.09 | 22.89±1.89 | 24.88±0.22 | 30.32±1.54 | 12.51±0.17 | 19.93±1.61 | 10.2±0.15 | 10.01±2.47 | 3.33±0.07 |
| Offline-LD (d-mSAC) | 16.37±0.5 | 10.54±0.16 | 39.55±6.46 | 23.6±2.54 | 23.63±6.52 | 11.85±3.21 | 14.93±2.03 | 6.43±0.16 | 5.82±0.95 | 1.26±0.22 |
| IQL | 16.35±0.53 | 10.51±0.22 | 30.95±2.93 | 19.75±0.77 | 20.5±0.29 | **9.98±0.19** | 14.08±1.57 | 6.38±0.08 | 5.54±0.34 | 1.04±0.06 |
| TD3-AWR | 14.82±0.48 | 9.84±0.18 | 29.47±8.34 | 16.05±1.28 | 24.58±2.28 | 10.38±0.39 | 13.43±1.58 | 6.30±0.10 | 6.28±1.50 | 0.95±0.04 |
| **CDQAC** | **12.21±0.37** | **6.48±0.15** | **14.6±0.78** | **9.6±0.1** | **17.67±1.49** | 10.77±0.35 | **11.67±0.6** | **5.76±0.08** | **3.94±0.43** | **0.87±0.16** |
| GA | | | | | | | | | | |
| BC | 17.21±0.31 | 13.41±0.09 | 28.88±1.67 | 18.13±0.21 | 17.73±0.78 | 9.22±0.12 | 14.43±0.44 | 6.97±0.05 | 11.03±0.61 | 1.8±0.07 |
| Offline-LD (mQRDQN) | 24.67±2.98 | 20.47±0.07 | 45.24±4.87 | 27.03±0.38 | 34.83±1.61 | 12.9±0.11 | 28.1±1.6 | 10.72±0.07 | 19.63±1.82 | 3.78±0.06 |
| Offline-LD (d-mSAC) | 16.11±0.71 | 8.74±0.1 | 29.23±1.9 | 14.89±0.51 | 31.93±2.12 | 13.69±0.86 | 22.88±1.09 | 8.39±0.13 | 16.12±2.14 | 4.71±0.56 |
| IQL | 15.54±0.84 | 11.08±0.2 | 26.69±3.37 | 16.28±0.79 | 26.84±3.04 | **11.78±0.84** | 20.41±2.1 | 7.72±0.47 | 14.15±2.5 | 3.26±1.04 |
| TD3-AWR | 14.52±0.37 | 10.10±0.03 | 25.63±2.77 | 15.15±0.80 | 29.98±2.28 | 12.53±1.14 | 22.35±0.62 | 7.87±0.28 | 14.99±1.10 | 4.01±0.73 |
| **CDQAC** | **12.3±0.45** | **6.19±0.24** | **19.6±4.61** | **10.22±1.76** | **23.53±6.23** | 11.8±2.82 | **14.37±3.46** | **6.13±0.83** | **7.46±3.69** | **1.63±0.96** |
| PDR-GA | | | | | | | | | | |
| BC | 23.94±4.08 | 13.35±0.76 | 57.21±3.94 | 26.53±0.61 | 27.61±1.06 | 11.73±0.68 | 22.35±2.5 | 8.37±0.79 | 12.78±1.97 | 2.67±0.31 |
| Offline-LD (mQRDQN) | 18.15±1.12 | 20.34±0.04 | 23.98±3.91 | 25.53±0.44 | 27.62±2.08 | 12.52±0.23 | 21.92±1.47 | 10.42±0.14 | 12.19±2.4 | 3.5±0.1 |
| Offline-LD (d-mSAC) | 17.42±0.65 | 9.36±0.36 | 35.9±4.16 | 17.54±1.75 | 34.09±3.15 | 14.81±1.36 | 21.91±1.3 | 8.75±0.29 | 14.99±1.35 | 4.62±0.22 |
| IQL | 15.33±0.52 | 10.5±0.13 | 28.15±1.59 | 19.1±0.66 | 25.06±2.39 | 11.43±0.43 | 16.4±2.51 | 6.69±0.24 | 6.56±2.62 | 1.22±0.26 |
| TD3-AWR | 13.66±0.85 | 9.76±0.18 | 19.79±1.12 | 13.10±0.41 | 26.36±1.39 | 10.86±0.30 | 14.50±0.73 | 6.45±0.03 | 6.55±0.64 | 1.41±0.15 |
| **CDQAC** | **12.28±0.26** | **6.15±0.47** | **14.75±1.53** | **8.72±0.59** | **18.02±4.44** | **9.55±1.42** | **11.44±0.88** | **5.44±0.28** | **3.51±0.91** | **0.78±0.15** |
| Random | | | | | | | | | | |
| BC | 30.41±3.73 | 20.87±0.09 | 36.61±4.36 | 26.66±0.33 | 25.66±2.26 | 12.33±0.09 | 22.56±3.89 | 10.5±0.12 | 10.92±3.12 | 3.58±0.03 |
| Offline-LD (mQRDQN) | 16.95±0.54 | 20.21±0.07 | 29.14±4.62 | 25.6±0.39 | 29.07±3.02 | 12.58±0.24 | 20.17±2.17 | 10.24±0.12 | 12.83±1.86 | 3.41±0.07 |
| Offline-LD (d-mSAC) | 15.02±0.43 | 8.17±0.31 | 20.44±1.58 | 11.27±0.49 | 30.92±3.15 | 14.52±1.5 | 18.06±1.22 | 7.46±0.33 | 9.97±1.41 | 2.32±0.45 |
| IQL | 15.58±1.64 | 13.75±0.28 | 24.5±2.93 | 18.12±0.42 | 24.63±4.43 | 11.54±0.78 | 19.69±2.85 | 8.81±0.53 | 12.43±3.42 | 3.75±0.86 |
| TD3-AWR | 14.28±0.56 | 13.18±0.44 | 21.59±0.72 | 16.53±0.86 | 26.65±4.27 | 11.85±1.20 | 17.03±2.25 | 8.01±0.54 | 8.37±2.91 | 2.08±0.64 |
| **CDQAC** | **12.04±0.59** | **6.7±0.62** | **13.58±0.66** | **8.73±0.73** | **14.56±0.55** | **8.51±0.52** | **10.77±0.36** | **5.22±0.12** | **3.16±0.1** | **0.67±0.02** |

Tables 17 and 18 show only minor performance differences between the training datasets. Table 18 contains the largest difference between the mean Greedy results of Demirkol between PDR (28.87%±0.99%) and PDR-GA (26.42%±2.49%). We also notice that PDR and Random perform better with the Taillard instances compared to GA, but GA performs better on the Demirkol instances. We hypothesize that this difference comes from the differing distributions of processing times: Demirkol instances have processing times ranging from 1 to 200 and those of Taillard only from 1 to 100, whereby CDQAC was trained on instances similar to Taillard instances. These results contrast with those of FJSP in App. I.2, where GA was unable to generalize well to benchmark instances that have a different distribution to the training instances. These results suggest that the choice of training data has a fundamentally different impact in JSP compared to FJSP.

## I.4 Additional Results Dataset Size

In Sect. 5.5, we demonstrated that reducing the number of training in the Random training dataset had little impact on overall performance on the FJSP benchmark sets, Brandimarte and Hurink. In this section, we provide a more comprehensive analysis by including results on generated evaluation instances. Additionally, we introduce a second evaluation for the reduction of the dataset, in which we decrease the number of solutions generated per instance by the random policy. For both evaluations, we considered subsets containing 1%, 5%, 10%, 25%, 50%, 75%, and 100% of the original dataset size. Specifically, when reducing the number of instances, we used either 5, 25, 50, 125, 250, 375, or 500 instances, each with 100 random solutions. When reducing the number of random solutions per instance, we used 500 instances, each with either 1, 5, 10, 25, 50, 75, or 100 random solutions.

As shown in Fig. 9, decreasing the dataset, either by limiting the number of instances or by reducing the number of random solutions per instance, does not lead to a significant loss in performance. The results remain relatively stable, with the standard deviation mostly below 1.5%. The sole exception occurs for $15 \times 10$ on the benchmark instances at 25%, when reducing the number of random solutions, where the greedy

Table 16: Results of FJSP offline RL comparison $20 \times 10$, for all training datasets (PDR, GA, PDR-GA, and Random). The columns show the evaluation benchmarks sets and the rows the methods. The mean and standard deviation of the gap (%) are reported from four different seeds. **Bold** indicates best result (lowest gap) for either the Greedy and Sampling (100 solutions) evaluation, for a given training dataset.

| | Generated $20 \times 10$ | | Brandimarte (mk) | | Hurink edata | | Hurink rdata | | Hurink vdata | |
|---|---|---|---|---|---|---|---|---|---|---|
| | Greedy | Sampling | Greedy | Sampling | Greedy | Sampling | Greedy | Sampling | Greedy | Sampling |
| **PDR** | | | | | | | | | | |
| BC | 33.37±2.71 | 9.29±0.78 | 65.13±6.3 | 34.94±2.12 | 27.47±3.56 | 12.91±2.18 | 24.42±4.57 | 9.11±0.96 | 9.03±2.55 | 1.27±0.04 |
| Offline-LD (mQRDQN) | 27.6±5.91 | 14.82±0.12 | 33.83±2.4 | 26.54±1.25 | 31.03±2.1 | 12.61±0.23 | 28.02±3.74 | 10.55±0.11 | 18.73±2.71 | 3.56±0.09 |
| Offline-LD (d-mSAC) | 15.43±3.82 | 8.38±1.08 | 55.97±4.05 | 33.3±1.67 | 33.17±4.2 | 15.66±2.26 | 23.86±1.87 | 8.91±0.87 | 9.9±2.93 | 2.11±0.9 |
| IQL | 10.43±1.11 | 6.77±0.33 | 45.31±3.96 | 24.95±1.83 | 25.31±4.42 | 11.8±1.26 | 16.59±1.26 | 7.19±0.26 | **5.06±0.41** | **1.06±0.06** |
| TD3-AWR | 10.55±1.12 | 6.43±0.23 | 37.48±9.02 | 21.36±3.32 | 30.29±7.93 | 13.42±3.81 | 22.58±6.62 | 8.85±2.11 | 8.62±5.59 | 1.75±0.95 |
| **CDQAC** | **9.38±6.1** | **4.38±3.47** | **16.65±0.5** | **9.7±0.7** | **21.5±5.18** | **11.23±1.97** | **15.53±3.05** | **6.98±1.29** | 8.47±4.0 | 2.94±2.31 |
| **GA** | | | | | | | | | | |
| BC | 11.73±0.59 | 8.4±0.09 | 24.69±1.69 | 18.36±0.44 | 17.76±0.06 | 9.69±0.16 | 13.51±0.4 | 7.12±0.11 | 8.08±1.44 | 1.85±0.04 |
| Offline-LD (mQRDQN) | 41.47±6.36 | 15.55±0.46 | 59.54±3.52 | 27.8±0.81 | 35.95±2.51 | 13.18±0.42 | 32.75±4.96 | 10.9±0.3 | 23.3±4.49 | 3.93±0.2 |
| Offline-LD (d-mSAC) | 20.78±5.21 | 6.75±1.63 | 28.37±0.97 | 14.84±0.77 | 29.33±1.33 | 12.93±0.52 | 21.76±2.85 | 7.76±0.64 | 14.73±2.45 | 4.35±0.52 |
| IQL | 21.12±4.65 | 7.59±0.57 | 28.71±2.85 | 16.24±0.43 | 26.89±2.31 | 11.63±0.56 | 22.18±2.55 | 7.64±0.4 | 13.96±1.01 | 3.14±0.82 |
| TD3-AWR | 14.45±7.59 | 6.49±0.58 | 27.64±6.37 | 16.22±1.78 | 26.62±4.73 | 11.46±1.09 | 20.68±2.06 | 7.17±0.58 | 13.18±1.26 | 2.71±1.16 |
| **CDQAC** | **5.22±0.63** | **2.19±0.62** | **16.76±2.09** | **9.3±0.36** | **22.62±6.05** | **11.05±3.2** | **13.48±0.97** | **5.92±0.37** | **4.91±1.07** | **0.97±0.2** |
| **PDR-GA** | | | | | | | | | | |
| BC | 26.02±2.15 | 8.21±0.17 | 53.55±6.26 | 28.02±2.45 | 23.61±1.62 | 10.91±0.87 | 16.26±3.61 | 7.15±0.4 | 5.89±2.1 | 1.08±0.09 |
| Offline-LD (mQRDQN) | 27.62±9.83 | 15.0±0.21 | 29.47±8.62 | 25.59±0.29 | 30.82±5.5 | 12.62±0.29 | 24.68±4.86 | 10.51±0.12 | 17.36±5.16 | 3.6±0.14 |
| Offline-LD (d-mSAC) | 43.5±3.7 | 21.72±5.92 | 55.46±4.38 | 24.48±1.64 | 38.71±1.75 | 19.46±1.27 | 32.65±3.36 | 13.12±1.35 | 22.98±2.97 | 8.78±2.03 |
| IQL | 11.42±2.36 | 6.89±0.25 | 33.97±4.27 | 19.32±1.4 | 24.65±2.62 | 10.79±0.56 | 16.03±1.22 | 6.91±0.25 | 6.44±1.63 | 1.19±0.17 |
| TD3-AWR | 7.49±0.65 | 6.64±0.39 | 23.76±4.58 | 16.82±1.14 | 21.94±5.97 | 10.49±1.41 | 13.46±1.07 | 7.05±0.33 | 5.46±0.81 | 1.33±0.03 |
| **CDQAC** | **5.01±0.28** | **2.31±0.36** | **15.34±1.11** | **8.9±0.59** | **17.79±5.04** | **9.17±1.49** | **12.3±1.61** | **5.57±0.43** | **4.07±0.91** | **0.83±0.24** |
| **Random** | | | | | | | | | | |
| BC | 19.65±1.43 | 15.33±0.08 | 35.81±9.72 | 27.02±0.37 | 23.84±2.46 | 12.29±0.1 | 20.23±0.37 | 10.47±0.1 | 11.38±0.95 | 3.51±0.12 |
| Offline-LD (mQRDQN) | 21.73±9.18 | 14.9±0.19 | 40.78±4.11 | 26.36±0.46 | 33.87±1.93 | 12.81±0.25 | 24.68±1.19 | 10.52±0.1 | 15.62±3.59 | 3.59±0.08 |
| Offline-LD (d-mSAC) | 11.59±3.69 | 4.79±1.46 | 22.09±3.25 | 11.7±1.28 | 28.51±2.45 | 13.37±1.85 | 21.7±3.27 | 9.18±1.85 | 13.07±3.76 | 3.7±2.14 |
| IQL | 14.0±4.05 | 10.08±0.93 | 32.8±4.58 | 20.08±0.72 | 31.66±2.55 | 13.17±0.61 | 24.87±3.32 | 9.63±0.39 | 13.81±3.39 | 3.53±0.5 |
| TD3-AWR | 8.49±1.18 | 8.80±0.57 | 19.87±1.64 | 18.10±0.18 | 23.72±1.59 | 11.17±0.29 | 16.02±0.53 | 8.32±0.28 | 5.76±0.36 | 1.97±0.25 |
| **CDQAC** | **5.2±0.66** | **2.87±0.73** | **16.52±0.3** | **9.73±0.43** | **16.53±1.59** | **9.02±0.28** | **11.63±0.52** | **5.66±0.17** | **3.25±0.2** | **0.76±0.05** |

Table 17: Results on JSP benchmarks for CDQAC $10 \times 5$, for all training datasets (PDR, GA, PDR-GA and Random). The mean and standard deviation of the gap (%) are reported from four different seeds. **Bold** indicates best result (lowest gap) for either the Greedy and Sampling (100 solutions) evaluation.

| | Instance Size | Greedy | | | | Sampling | | | |
|---|---|---|---|---|---|---|---|---|---|
| | | PDR | GA | PDR-GA | Random | PDR | GA | PDR-GA | Random |
| Taillard | 15 × 15 | 16.26 ± 0.67 | 16.12 ± 0.69 | 16.33 ± 0.95 | **15.9 ± 0.7** | 11.5 ± 0.51 | 11.27 ± 0.86 | 11.23 ± 0.48 | **10.8 ± 0.55** |
| | 20 × 15 | 20.55 ± 0.95 | 19.7 ± 1.05 | **19.6 ± 1.91** | 19.98 ± 1.91 | 14.8 ± 0.37 | 14.23 ± 0.75 | 14.64 ± 0.58 | **14.12 ± 0.77** |
| | 20 × 20 | 18.65 ± 0.73 | 18.89 ± 1.27 | 17.45 ± 0.7 | **17.19 ± 1.38** | **13.29 ± 0.68** | 14.1 ± 0.72 | 13.88 ± 0.35 | 13.39 ± 0.84 |
| | 30 × 15 | 20.4 ± 0.65 | 21.32 ± 2.78 | 20.44 ± 1.13 | **19.56 ± 0.49** | 15.83 ± 0.34 | 16.04 ± 0.91 | 16.0 ± 0.31 | **15.3 ± 1.13** |
| | 30 × 20 | 22.05 ± 1.64 | 22.58 ± 2.72 | **21.6 ± 2.04** | 22.28 ± 1.01 | **17.89 ± 0.92** | 18.6 ± 1.3 | 18.6 ± 0.4 | 18.27 ± 0.82 |
| | 50 × 15 | 14.26 ± 1.1 | 14.48 ± 1.63 | 13.53 ± 1.41 | **13.06 ± 1.47** | 10.86 ± 0.66 | **10.21 ± 0.75** | 10.47 ± 1.18 | 10.46 ± 1.22 |
| | 50 × 20 | 14.46 ± 0.95 | 15.21 ± 3.36 | **13.83 ± 1.18** | 13.9 ± 1.3 | 11.6 ± 0.35 | 12.07 ± 1.21 | 11.62 ± 0.47 | **11.37 ± 0.52** |
| | 100 × 20 | 6.43 ± 0.12 | 8.1 ± 4.73 | 6.18 ± 0.82 | **5.53 ± 1.12** | 4.66 ± 0.14 | 4.46 ± 1.33 | 4.56 ± 0.49 | **4.25 ± 0.59** |
| | Mean | 16.63 ± 0.85 | 17.05 ± 2.28 | 16.12 ± 1.27 | **15.93 ± 1.17** | 12.55 ± 0.5 | 12.62 ± 0.98 | 12.62 ± 0.53 | **12.24 ± 0.8** |
| Demirkol | 20 × 15 | 24.87 ± 1.51 | **24.03 ± 0.94** | 24.47 ± 2.11 | 24.49 ± 1.83 | 19.4 ± 0.63 | 19.29 ± 0.94 | 19.63 ± 0.81 | **18.82 ± 0.86** |
| | 20 × 20 | 23.3 ± 0.36 | **21.29 ± 1.19** | 22.01 ± 1.12 | 21.71 ± 1.47 | 17.66 ± 0.45 | 17.62 ± 1.15 | 18.03 ± 0.54 | **17.13 ± 0.71** |
| | 30 × 15 | 29.63 ± 0.69 | **28.22 ± 1.8** | 28.71 ± 2.63 | 28.76 ± 1.72 | 24.21 ± 0.61 | **23.22 ± 1.1** | 24.2 ± 1.21 | 23.67 ± 1.7 |
| | 30 × 20 | 28.72 ± 1.13 | **28.33 ± 1.0** | 28.53 ± 2.57 | 28.6 ± 2.39 | 23.72 ± 0.61 | 23.71 ± 0.5 | 24.15 ± 1.55 | **23.56 ± 1.29** |
| | 40 × 15 | 26.98 ± 1.0 | **25.1 ± 1.35** | 25.76 ± 2.78 | 25.51 ± 2.85 | 22.62 ± 0.98 | **20.31 ± 0.84** | 21.73 ± 1.63 | 21.15 ± 1.66 |
| | 40 × 20 | 29.42 ± 1.18 | **27.49 ± 1.45** | 28.5 ± 2.67 | 28.77 ± 1.74 | 24.88 ± 0.18 | **24.06 ± 1.03** | 25.1 ± 1.7 | 24.58 ± 1.49 |
| | 50 × 15 | 27.82 ± 0.94 | **25.03 ± 2.61** | 26.49 ± 3.84 | 25.06 ± 5.42 | 23.8 ± 0.85 | **20.83 ± 0.97** | 22.53 ± 2.5 | 22.5 ± 2.74 |
| | 50 × 20 | 30.43 ± 0.96 | **27.5 ± 1.63** | 28.71 ± 2.98 | 28.65 ± 2.58 | 26.35 ± 0.69 | **24.65 ± 1.28** | 26.1 ± 1.52 | 25.67 ± 1.06 |
| | Mean | 27.65 ± 0.97 | **25.87 ± 1.5** | 26.65 ± 2.59 | 26.44 ± 2.5 | 22.83 ± 0.62 | **21.71 ± 0.98** | 22.68 ± 1.43 | 22.13 ± 1.44 |

evaluation shows a standard deviation of 2.63%. Notably, this increased standard deviation is only observed for benchmark instances and not for generated instances at 25% random solutions, as evidenced in Fig. 9. This suggests that larger datasets may improve generalization to previously unseen instances. Another benefit is training stability, with larger dataset producing a smaller standard deviation. In general, these findings

Table 18: Results on JSP benchmarks for CDQAC $15 \times 10$, for all training datasets (PDR, GA, PDR-GA and Random). The mean and standard deviation of the gap (%) are reported from four different seeds. **Bold** indicates best result (lowest gap) for either the Greedy and Sampling (100 solutions) evaluation.

| | | Greedy | | | | Sampling | | | |
|---|---|---|---|---|---|---|---|---|---|
| | Instance Size | PDR | GA | PDR-GA | Random | PDR | GA | PDR-GA | Random |
| Taillard | $15 \times 15$ | $16.73 \pm 0.6$ | $17.23 \pm 1.28$ | $17.35 \pm 1.89$ | $\mathbf{16.7 \pm 0.97}$ | $11.6 \pm 0.48$ | $11.26 \pm 0.84$ | $11.39 \pm 0.86$ | $\mathbf{11.24 \pm 0.83}$ |
| | $20 \times 15$ | $21.63 \pm 0.33$ | $21.4 \pm 1.92$ | $21.57 \pm 2.04$ | $\mathbf{20.69 \pm 1.09}$ | $15.22 \pm 0.23$ | $14.77 \pm 1.13$ | $14.87 \pm 0.92$ | $\mathbf{14.71 \pm 0.28}$ |
| | $20 \times 20$ | $18.73 \pm 0.71$ | $18.51 \pm 1.41$ | $19.0 \pm 1.11$ | $\mathbf{18.14 \pm 0.81}$ | $13.61 \pm 0.59$ | $\mathbf{13.49 \pm 0.57}$ | $13.76 \pm 0.52$ | $13.53 \pm 0.5$ |
| | $30 \times 15$ | $\mathbf{20.6 \pm 0.65}$ | $21.27 \pm 1.27$ | $21.33 \pm 1.4$ | $20.86 \pm 0.88$ | $16.01 \pm 0.14$ | $16.21 \pm 0.88$ | $16.07 \pm 0.51$ | $\mathbf{15.92 \pm 0.3}$ |
| | $30 \times 20$ | $23.52 \pm 1.14$ | $\mathbf{23.17 \pm 0.34}$ | $23.94 \pm 0.69$ | $23.55 \pm 1.14$ | $18.43 \pm 0.6$ | $\mathbf{18.15 \pm 0.47}$ | $18.43 \pm 0.62$ | $18.29 \pm 0.5$ |
| | $50 \times 15$ | $14.9 \pm 0.28$ | $14.6 \pm 1.09$ | $\mathbf{14.05 \pm 1.31}$ | $15.47 \pm 2.47$ | $11.45 \pm 0.54$ | $10.71 \pm 1.06$ | $10.8 \pm 1.4$ | $\mathbf{10.48 \pm 0.43}$ |
| | $50 \times 20$ | $\mathbf{14.82 \pm 0.77}$ | $16.46 \pm 0.99$ | $15.41 \pm 1.03$ | $16.47 \pm 4.19$ | $12.05 \pm 0.54$ | $11.93 \pm 0.82$ | $12.17 \pm 1.13$ | $\mathbf{11.57 \pm 0.24}$ |
| | $100 \times 20$ | $6.44 \pm 0.34$ | $8.24 \pm 2.43$ | $\mathbf{6.01 \pm 0.97}$ | $8.0 \pm 3.19$ | $4.88 \pm 0.25$ | $4.73 \pm 0.34$ | $\mathbf{4.52 \pm 0.49}$ | $4.96 \pm 0.75$ |
| | Mean | $\mathbf{17.17 \pm 0.6}$ | $17.61 \pm 1.34$ | $17.33 \pm 1.31$ | $17.48 \pm 1.84$ | $12.91 \pm 0.42$ | $12.66 \pm 0.77$ | $12.75 \pm 0.81$ | $\mathbf{12.59 \pm 0.48}$ |
| Demirkol | $20 \times 15$ | $27.13 \pm 0.74$ | $26.03 \pm 1.0$ | $\mathbf{24.94 \pm 1.91}$ | $26.05 \pm 1.37$ | $20.23 \pm 0.8$ | $\mathbf{19.4 \pm 1.05}$ | $19.5 \pm 1.3$ | $19.59 \pm 0.72$ |
| | $20 \times 20$ | $24.01 \pm 0.5$ | $22.86 \pm 1.19$ | $22.73 \pm 2.13$ | $\mathbf{22.67 \pm 1.4}$ | $17.59 \pm 0.62$ | $17.4 \pm 0.62$ | $17.6 \pm 0.66$ | $\mathbf{17.23 \pm 0.68}$ |
| | $30 \times 15$ | $30.3 \pm 1.13$ | $29.66 \pm 1.54$ | $29.19 \pm 1.49$ | $\mathbf{29.15 \pm 1.19}$ | $25.93 \pm 1.37$ | $\mathbf{24.04 \pm 1.21}$ | $24.05 \pm 1.9$ | $24.25 \pm 0.87$ |
| | $30 \times 20$ | $30.43 \pm 1.04$ | $28.65 \pm 1.32$ | $28.5 \pm 2.35$ | $\mathbf{28.24 \pm 1.57}$ | $24.92 \pm 0.64$ | $\mathbf{23.0 \pm 0.83}$ | $23.72 \pm 1.55$ | $23.46 \pm 1.07$ |
| | $40 \times 15$ | $27.81 \pm 0.97$ | $25.68 \pm 0.97$ | $\mathbf{24.77 \pm 2.6}$ | $25.61 \pm 1.71$ | $23.51 \pm 1.04$ | $\mathbf{21.03 \pm 1.25}$ | $21.08 \pm 2.15$ | $21.38 \pm 1.49$ |
| | $40 \times 20$ | $30.54 \pm 1.26$ | $\mathbf{27.64 \pm 2.05}$ | $28.47 \pm 2.71$ | $28.99 \pm 0.81$ | $25.86 \pm 1.0$ | $\mathbf{23.63 \pm 0.98}$ | $24.48 \pm 1.73$ | $24.21 \pm 1.6$ |
| | $50 \times 15$ | $29.14 \pm 0.94$ | $\mathbf{23.84 \pm 4.59}$ | $24.28 \pm 5.27$ | $26.16 \pm 2.21$ | $25.09 \pm 1.04$ | $20.78 \pm 2.54$ | $21.87 \pm 3.35$ | $\mathbf{20.65 \pm 3.01}$ |
| | $50 \times 20$ | $31.56 \pm 1.3$ | $28.85 \pm 1.36$ | $\mathbf{28.43 \pm 1.44}$ | $30.53 \pm 1.94$ | $27.19 \pm 1.34$ | $\mathbf{24.4 \pm 0.79}$ | $24.97 \pm 0.87$ | $25.47 \pm 1.48$ |
| | Mean | $28.87 \pm 0.99$ | $26.65 \pm 1.75$ | $\mathbf{26.42 \pm 2.49}$ | $27.18 \pm 1.52$ | $23.79 \pm 0.98$ | $\mathbf{21.71 \pm 1.16}$ | $22.16 \pm 1.69$ | $22.03 \pm 1.36$ |

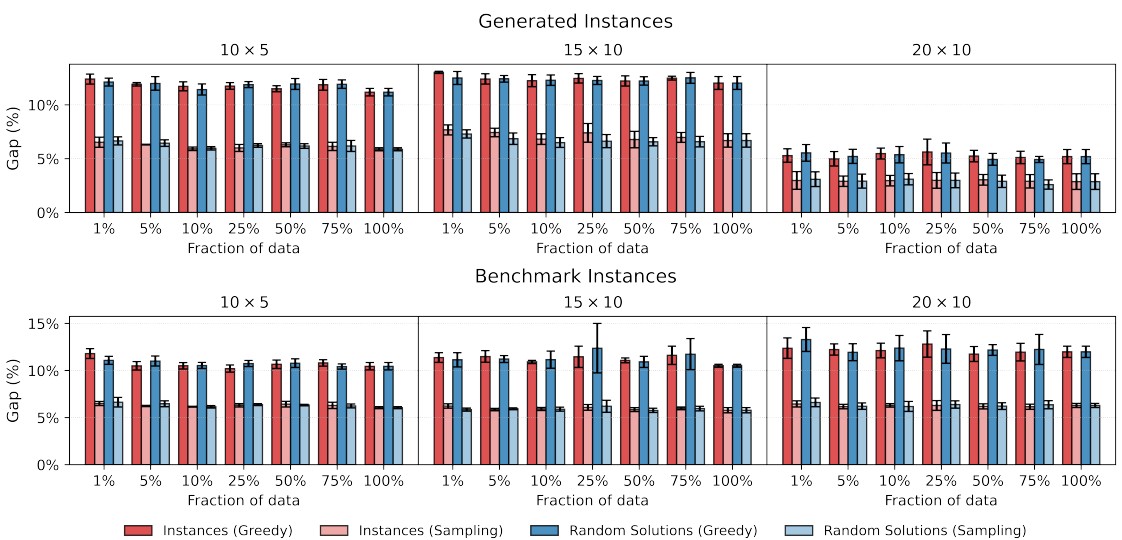

Figure 9: Effect of different dataset sizes. We evaluate the sample efficiency of CDQAC by reducing the Random training dataset in two ways. **Red**: the number of *instances* (1%: 5 instances, 5%: 25 instances, 10%: 50 instances, 25%: 125 instances, 50%: 250 instances, 75%: 375 instances, 100%: 500 instances, with each instance having 100 random solutions). **Blue**: the number of *random solutions* per instance (1%: 1 solution, 5%: 5 solutions, 10%: 10 solutions, 25%: 25 solutions, 50%: 50 solutions, 75%: 75 solutions, 100%: 100 solutions, for each instance, with 500 instances in total). Performance is reported as the mean gap across four seeds, with error bars indicating standard deviation.

reinforce our conclusion from Sect. 5.5: CDQAC maintains competitive performance even when trained on substantially reduced datasets, underscoring its sample efficiency.

Table 19: Results JSP benchmarks. Average gap (%) is reported. In this additional comparison, we compare CDQAC to constructive learning-based approaches that only function for JSP and to approaches that function for both JSP and FJSP. The approaches that only function for JSP are: L2D (Zhang et al., 2020), CL (Iklassov et al., 2023), Sched (Park et al., 2021), SL (Corsini et al., 2024), GD (Pirnay & Grimm, 2024), and OD (Remmerden et al., 2025). Approaches that can do both JSP and FJSP are: DAN (Wang et al., 2023), Res (Ho et al., 2024), and CDQAC (ours). We additionally include the JSP-only improvement heuristics L2S (Zhang et al., 2024a) and TBGAT (Zhang et al., 2024b) evaluated for 500 and 1000 improvement steps. We note the best performing overall approach with *, and the best approach that can handle both JSP and FJSP in **bold**.

| | | Greedy | | | | | | | | | Sampling | | | | | | Improvement | | | |
| | | | | | | | | | | | | | | | | | L2S | | TBGAT | |
| | Instance Size | L2D | CL | Sched | SL | GD | OD | DAN | Res | **CDQAC** | CL[a] | SL[a] | GD[b] | DAN[c] | Res[c] | **CDQAC[c]** | 500 | 1000 | 500 | 1000 |
|---|---|---|---|---|---|---|---|---|---|---|---|---|---|---|---|---|---|---|---|---|
| Taillard | $15 \times 15$ | 28.1 | 14.3 | 15.3 | 13.8 | 9.6 | 25.8 | 19.0 | 17.6 | **15.0** | 9.0 | 7.2* | 10.1 | 13.2 | 13.3 | **10.4** | 9.3 | 8.6 | 8.0 | 6.1 |
| | $20 \times 15$ | 32.7 | 16.5 | 19.4 | 15.0 | 9.9* | 30.2 | 22.1 | 21.2 | **17.7** | 10.6 | 9.3* | 9.8 | 17.4 | 16.1 | **13.2** | 11.6 | 10.4 | 9.9 | 8.7 |
| | $20 \times 20$ | 31.8 | 17.3 | 17.2 | 15.2 | 11.1* | 28.9 | 18.0 | 18.0 | **17.6** | 10.9 | 10.0* | 10.4 | 13.3 | 15.8 | **12.9** | 12.4 | 11.4 | 10.0 | 9.0 |
| | $30 \times 15$ | 30.2 | 18.5 | 18.0 | 17.1 | 9.5* | 29.2 | 21.7 | 20.1 | **19.1** | 14.0 | 11.0 | 8.5* | 17.2 | 18.0 | **14.9** | 14.7 | 12.9 | 13.3 | 10.9 |
| | $30 \times 20$ | 35.2 | 21.5 | 18.7 | 18.5 | 13.8* | 33.1 | 23.2 | 22.3 | **21.2** | 16.1 | 13.4 | 12.3* | 19.0 | 19.7 | **17.9** | 17.5 | 15.7 | 16.4 | 14.0 |
| | $50 \times 15$ | 21.0 | 12.2 | 13.8 | 10.1 | 2.7* | 20.6 | 14.8 | 15.6 | **13.0** | 9.3 | 5.5 | 2.6* | 12.7 | 13.2 | **9.9** | 11.0 | 9.0 | 9.6 | 7.5 |
| | $50 \times 20$ | 26.1 | 13.2 | 13.5 | 11.6 | 6.7* | 24.3 | 16.0 | 14.4 | **12.8** | 9.9 | 8.4 | 7.7* | 13.1 | 14.1 | **11.0** | 13.0 | 11.4 | 11.9 | 9.4 |
| | $100 \times 20$ | 13.3 | 5.9 | 6.6 | 5.8 | 1.7* | 12.7 | 7.3 | 6.5 | **5.3** | 4.0 | 2.3 | 1.3* | 5.9 | 6.5 | **3.6** | 7.9 | 6.6 | 6.4 | 4.9 |
| | Mean | 27.3 | 14.9 | 15.4 | 13.4 | 8.1* | 25.6 | 18.2 | 17.0 | **15.2** | 10.5 | 8.4 | 7.8* | 14.4 | 14.6 | **11.7** | 12.2 | 10.8 | 10.7 | 8.8 |
| Demirkol | $20 \times 15$ | 36.3 | – | – | 18.0* | – | 35.8 | – | 26.1 | **22.9** | – | 12.0* | – | – | 22.6 | **18.4** | – | – | – | – |
| | $20 \times 20$ | 34.4 | – | – | 19.4* | – | 32.8 | – | 21.5 | **20.3** | – | 13.5* | – | – | 18.9 | **16.5** | – | – | – | – |
| | $30 \times 15$ | 37.8 | – | – | 21.8* | – | 38.8 | – | 27.6 | **27.1** | – | 14.4* | – | – | 29.4 | **23.1** | – | – | – | – |
| | $30 \times 20$ | 38.0 | – | – | 25.7* | – | 36.0 | – | 29.9 | **27.9** | – | 17.1* | – | – | 28.3 | **23.4** | – | – | – | – |
| | $40 \times 15$ | 34.6 | – | – | 17.5* | – | 35.5 | – | 26.2 | **25.5** | – | 11.7* | – | – | 28.4 | **20.2** | – | – | – | – |
| | $40 \times 20$ | 39.2 | – | – | 22.2* | – | 38.5 | – | 27.7 | **27.9** | – | 16.0* | – | – | 30.9 | **24.1** | – | – | – | – |
| | $50 \times 15$ | 33.2 | – | – | 15.7* | – | 34.1 | – | 27.4 | **25.0** | – | 11.2* | – | – | 29.5 | **21.7** | – | – | – | – |
| | $50 \times 20$ | 37.7 | – | – | 22.4* | – | 38.9 | – | 30.0 | **28.6** | – | 15.8* | – | – | 32.8 | **25.1** | – | – | – | – |
| | Mean | 36.4 | – | – | 20.3* | – | 36.3 | – | 27.0 | **25.7** | – | 14.0* | – | – | 27.6 | **21.6** | – | – | – | – |

[a] Used 128 samples for each instance during the sampling evaluation.
[b] Used beam search with a width of 16.
[c] Used 100 samples for each instance during the sampling evaluation.

## I.5 Additional JSP baselines

Table 19 provides an extended JSP comparison that includes specialized JSP-only baselines excluded from Table 7, where the requirement was that methods must operate on both JSP and FJSP. Among constructive RL approaches, to which CDQAC also belongs, CDQAC is competitive with CL (Iklassov et al., 2023) and Sched (Park et al., 2021), with CL slightly ahead overall. The self-labeling methods SL (Corsini et al., 2024) and GD (Pirnay & Grimm, 2024) achieve lower gaps, and the improvement-based methods L2S (Zhang et al., 2024a) and TBGAT (Zhang et al., 2024b) outperform CDQAC on smaller Taillard instances. CDQAC scales more favorably, however: on the largest Taillard instances ($100 \times 20$), its sampling evaluation (3.6%) outperforms both L2S-1000 (6.6%) and TBGAT-1000 (4.9%).

These comparisons should be read in context. All additional baselines are JSP-specific and require a training environment, so none extend to FJSP, where CDQAC remains state-of-the-art. The lower gaps reported by SL and GD also come at substantial training cost: both methods require up to seven days of GPU training, while improvement methods incur additional inference cost proportional to the number of improvement steps. CDQAC, by contrast, trains in one to two hours from a static random dataset, without simulator access or expert demonstrations. We therefore position CDQAC not as a replacement for specialized JSP methods but as a complementary offline approach with broader applicability and substantially lower training cost.

Table 20: Multi-seed comparison on FJSP benchmark sets (Brandimarte and Hurink). Both methods trained on the Random dataset at the indicated training size. Single-seed DANIEL values are taken from Table 4 of the main paper for reference. Mean and standard deviation of the gap (%) reported over four seeds. Bold indicates the lower mean gap among the multi-seed entries within each (training size, evaluation mode) row group.

| | Train size | Method | Brandimarte (mk) | Hurink edata | Hurink rdata | Hurink vdata |
|---|---|---|---|---|---|---|
| Greedy | $10 \times 5$ | DANIEL (single seed)[†] | 13.58 | 16.33 | 11.42 | 3.28 |
| | | DANIEL (multi-seed) | $14.04 \pm 1.44$ | $15.72 \pm 0.47$ | $11.24 \pm 0.60$ | $\mathbf{2.94 \pm 0.12}$ |
| | | CDQAC (Ours) | $\mathbf{13.78 \pm 0.78}$ | $\mathbf{14.53 \pm 0.41}$ | $\mathbf{10.40 \pm 0.36}$ | $3.10 \pm 0.22$ |
| | $15 \times 10$ | DANIEL (single seed)[†] | 12.97 | 14.41 | 12.07 | 3.75 |
| | | DANIEL (multi-seed) | $\mathbf{13.58 \pm 0.24}$ | $14.97 \pm 0.14$ | $11.15 \pm 0.37$ | $\mathbf{3.06 \pm 0.07}$ |
| | | CDQAC (Ours) | $\mathbf{13.58 \pm 0.66}$ | $\mathbf{14.56 \pm 0.55}$ | $\mathbf{10.77 \pm 0.36}$ | $3.16 \pm 0.10$ |
| Sampling | $10 \times 5$ | DANIEL (single seed)[†] | 9.53 | 9.08 | 4.95 | 0.69 |
| | | DANIEL (multi-seed) | $9.66 \pm 0.63$ | $\mathbf{8.90 \pm 0.55}$ | $5.34 \pm 0.13$ | $\mathbf{0.66 \pm 0.06}$ |
| | | CDQAC (Ours) | $\mathbf{8.67 \pm 0.21}$ | $9.54 \pm 0.39$ | $\mathbf{5.30 \pm 0.22}$ | $0.68 \pm 0.03$ |
| | $15 \times 10$ | DANIEL (single seed)[†] | 8.95 | 8.72 | 5.49 | 0.72 |
| | | DANIEL (multi-seed) | $8.89 \pm 0.46$ | $9.47 \pm 0.47$ | $5.44 \pm 0.24$ | $\mathbf{0.64 \pm 0.05}$ |
| | | CDQAC (Ours) | $\mathbf{8.73 \pm 0.73}$ | $\mathbf{8.51 \pm 0.52}$ | $\mathbf{5.22 \pm 0.12}$ | $0.67 \pm 0.02$ |

[†] As reported in Table 4 of the main paper.

## J   Significance Test

Our comparison for FJSP (Sect. 5.3) and JSP (Sect. 5.4) showed that CDQAC outperformed DANIEL (Wang et al., 2023) in most evaluations. To assess whether these results are significant, we conducted a one-sided Wilcoxon signed-rank test for both JSP and FJSP.

### J.1   FJSP.

Although CDQAC consistently outperformed DANIEL in most FJSP evaluations, the margins were smaller than in other results. To this end, we paired all results from Tables 4, 5, and 6, in both greedy and sampling evaluations. Furthermore, we paired the results of both $10 \times 5$ and $15 \times 10$ in Table 4, resulting in a sample size of 26 pairs. The statistical test yielded a $p \approx 0.018$ rejecting the null hypothesis of $p > 0.05$, indicating that CDQAC, trained solely on random data, significantly outperforms the online RL baseline DANIEL (Wang et al., 2023) in our FJSP evaluation.

**Multi-seed Comparison with DANIEL**   The Wilcoxon test above relies on the single-seed DANIEL results in Tables 4, 5, and 6, following the convention of prior online RL work for FJSP, which typically reports single-seed results (see Sect. 5.3). To verify that the observed significance is not an artifact of an unfavorable seed choice for DANIEL, we additionally retrain DANIEL (Wang et al., 2023) with the same four random seeds as CDQAC (1, 2, 3, 4) Tables 20, 21, and 22 report the multi-seed results, per benchmark set, for CDQAC and DANIEL Wang et al. (2023) and the single-seed results of DANIEL, as reference, originally given in Tables 4, 5, and 6, respectively.

The single-seed DANIEL values fall within approximately one to two standard deviations of the multi-seed results, indicating that the originally reported values are representative of DANIEL performance. Across the 16 benchmark results in Table 20, CDQAC achieves the lower mean gap on 10 results, ties on 1, and trails on 5, while on the four large-instance generalization sets in Table 22 CDQAC outperforms DANIEL on all four by 1.3–2.8 percentage points; DANIEL retains its edge on the largest in-distribution training size ($20 \times 10$ in Table 21), consistent with the observation in Sect. 5.3 that online RL is most competitive within its training distribution. Repeating the one-sided Wilcoxon signed-rank test on the same 26 pairs using the multi-seed DANIEL means yields $p \approx 0.03$, which still rejects the null hypothesis at the 0.05 level. The conclusion that CDQAC significantly outperforms DANIEL on FJSP therefore holds under multi-seed evaluation.

Table 21: Multi-seed comparison on generated FJSP evaluation instances. Both methods are trained and evaluated at the same instance size, on the Random dataset. Single-seed DANIEL values are taken from Table 5 of the main paper for reference. Mean and standard deviation of the gap (%) reported over four seeds. Bold indicates the lower mean gap among the multi-seed entries.

| | Method | $10 \times 5$ | $15 \times 10$ | $20 \times 10$ |
|---|---|---|---|---|
| | DANIEL (single seed)[†] | 10.87 | 12.42 | 1.31 |
| Greedy | DANIEL (multi-seed) | $12.14 \pm 0.61$ | $12.22 \pm 0.48$ | $\mathbf{1.98 \pm 0.62}$ |
| | CDQAC (Ours) | $\mathbf{11.19 \pm 0.35}$ | $\mathbf{12.04 \pm 0.59}$ | $5.20 \pm 0.66$ |
| | DANIEL (single seed)[†] | 5.57 | 6.79 | -1.03 |
| Sampling | DANIEL (multi-seed) | $6.18 \pm 0.36$ | $\mathbf{6.46 \pm 0.55}$ | $\mathbf{-0.18 \pm 0.53}$ |
| | CDQAC (Ours) | $\mathbf{5.87 \pm 0.14}$ | $6.70 \pm 0.62$ | $2.87 \pm 0.73$ |

[†] As reported in Table 5 of the main paper.

Table 22: Multi-seed comparison on large generalization FJSP instances. Both methods trained on $10 \times 5$ instances with the Random dataset, evaluated on larger unseen sizes. Single-seed DANIEL values are taken from Table 6 of the main paper for reference. Mean and standard deviation of the gap (%) reported over four seeds. Bold indicates the lower mean gap among the multi-seed entries.

| | Method | $30 \times 10$ | $40 \times 10$ |
|---|---|---|---|
| | DANIEL (single seed)[†] | 5.10 | 3.65 |
| Greedy | DANIEL (multi-seed) | $6.63 \pm 0.88$ | $5.19 \pm 0.91$ |
| | CDQAC (Ours) | $\mathbf{5.37 \pm 0.68}$ | $\mathbf{3.89 \pm 0.68}$ |
| | DANIEL (single seed)[†] | 4.43 | 3.77 |
| Sampling | DANIEL (multi-seed) | $6.30 \pm 0.90$ | $5.70 \pm 0.92$ |
| | CDQAC (Ours) | $\mathbf{3.79 \pm 0.61}$ | $\mathbf{2.86 \pm 0.56}$ |

[†] As reported in Table 6 of the main paper.

### J.2 JSP.

To evaluate the significance of the JSP results, we again paired the results of CDQAC and DANIEL in Table 7, whereby we paired each Taillard result, both for greedy and sampling. This results in a sample size of 16 pairs. The Wilcoxon test resulted in $p \approx 0.00022$, indicating that CDQAC also significantly outperforms DANIEL on JSP.

## K   Quantile Critic Analysis

This appendix analyzes two properties of the quantile critic that are relevant to the design of CDQAC. First, we examine whether applying the conservative penalty only to the expected return leaves optimistic upper-tail estimates uncontrolled. Second, we quantify the extent and practical consequences of quantile crossing after the dueling recombination in Eq. 6. We additionally compare the mean-value CQL penalty used by CDQAC with the full-distribution conservative penalty proposed by CODAC (Ma et al., 2021).

### K.1   Mean-based conservatism and upper-tail estimates

CDQAC applies its conservative penalty to

$$Q_\theta^Z(s, a) = \mathbb{E}[Z_\theta(s, a)] = \frac{1}{N} \sum_{n=1}^{N} \theta^n(s, a),$$

rather than regularizing every predicted quantile separately. This choice is consistent with the risk-neutral objective considered in this work: both the policy objective in Eq. 5 and the CQL penalty in Eq. 4 depend only on the expected return. Consequently, an optimistic upper-tail estimate can affect policy improvement only through its contribution to the mean, which is directly subject to the conservative penalty.

The clipped double-critic target provides an additional safeguard. For each state–action pair, CDQAC compares the expected values predicted by its two critic heads and uses the complete quantile vector from

Return distributions learned by the distributional (QR-DQN) critic for FJSP state-action pairs

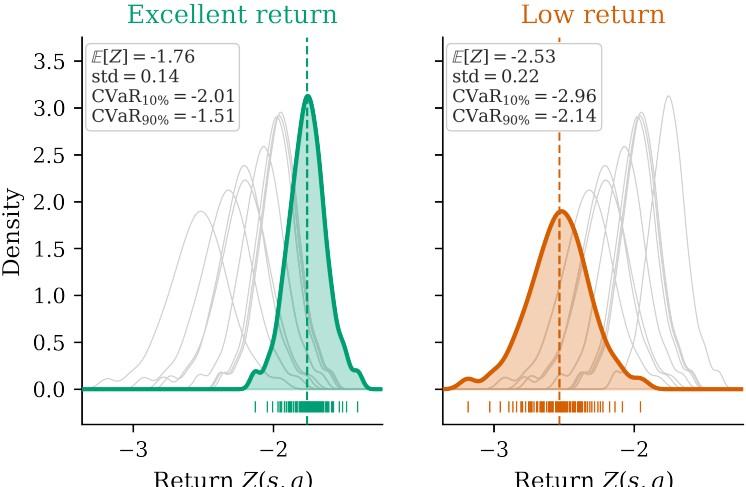

Figure 10: The return distribution of a single state. The green return represent an excellent return, whereas the orange represents a low return

the head with the lower expected value:

$$i^\star(s,a) = \arg\min_{i \in \{1,2\}} \frac{1}{N} \sum_{n=1}^{N} \theta_i^n(s,a), \qquad Z_\theta(s,a) = Z_{\theta_{i^\star}}(s,a).$$

Thus, the minimum is taken over the expected values of the two heads; individual quantiles are not minimized independently. This preserves the internal dependence structure of the selected return distribution while preventing an optimistic critic head from being propagated through the Bellman target.

We evaluate the trained critic on more than 270,000 reachable state–action pairs obtained by replaying trajectories from the training datasets. These states contain more than 1.3 million within-state ordered action comparisons. The expected return and the 0.95-quantile are almost perfectly rank-aligned across the available actions, with a Spearman rank correlation of $\rho = 0.997$.

A lower-mean action has a higher 0.95-quantile in only 7.2% of the within-state action comparisons. Moreover, the action with the highest expected return is also the action with the highest 0.95-quantile in 89.7% of the evaluated states. These results indicate that the potentially problematic configuration of a low-mean action with a highly optimistic upper tail is uncommon in the learned critic.

Figure 10 provides a representative visualization at a single decision state. It shows the predicted return distributions of all feasible machine–operation pairs and highlights the actions with the highest and lowest expected returns. The two highlighted distributions are clearly separated, and the higher-value action stochastically dominates the lower-value action over the plotted quantile range. The distributions are unimodal; the relevant observation is therefore not multimodality, but that the quantile critic represents the complete ordering and separation of the action-conditioned return distributions rather than only their scalar expectations.

## K.2   Quantile crossing after dueling recombination

The dueling recombination in Eq. 6 is applied independently at every quantile fraction. More explicitly, for quantile fraction $\tau_n$,

$$\theta^n(s,a) = V^n(h_G) + A^n(s,a) - \frac{1}{|\mathcal{A}(s)|} \sum_{a' \in \mathcal{A}(s)} A^n(s,a'). \tag{18}$$

Table 23: Comparison between the mean-value CQL penalty used by CDQAC and CODAC's distributional conservative penalty, trained on the Random dataset ($10 \times 5$). We report the mean gap (%) on In-Distribution instances and Out-of-Distribution FJSP benchmarks (Brandimarte and Hurink). Avg $\Delta(\%)$ is the mean percentage difference across the four evaluation columns relative to standard CDQAC with $\alpha_{\mathrm{CQL}} = 0.05$. Results are averaged over four seeds. Lower values are better. **Bold** indicates the best result per column.

| Mechanism | Value | In-Distribution (Gap %) | | Out-of-Distribution (Gap %) | | Avg $\Delta(\%)$ |
| | | Greedy | Sampling | Greedy | Sampling | |
| --- | --- | --- | --- | --- | --- | --- |
| Mean CQL | $\alpha_{\mathrm{CQL}} = 0.05$ | **$11.19 \pm 0.41$** | **$5.87 \pm 0.16$** | **$10.45 \pm 0.46$** | **$6.05 \pm 0.12$** | **0.00%** |
| CODAC | $\alpha_{\mathrm{CODAC}} = 0.025$ | $11.97 \pm 0.44$ | $6.21 \pm 0.18$ | $10.96 \pm 0.61$ | $6.26 \pm 0.21$ | +5.30% |
| | $\alpha_{\mathrm{CODAC}} = 0.05$ | $11.35 \pm 0.56$ | $5.95 \pm 0.10$ | $10.52 \pm 0.42$ | $6.23 \pm 0.22$ | +1.59% |
| | $\alpha_{\mathrm{CODAC}} = 0.1$ | $11.43 \pm 0.65$ | $5.97 \pm 0.63$ | $10.54 \pm 0.50$ | $6.25 \pm 0.35$ | +2.00% |
| | $\alpha_{\mathrm{CODAC}} = 0.25$ | $11.76 \pm 0.20$ | $6.18 \pm 0.42$ | $10.66 \pm 0.42$ | $6.23 \pm 0.29$ | +3.83% |
| | $\alpha_{\mathrm{CODAC}} = 0.5$ | $11.43 \pm 0.24$ | $6.04 \pm 0.36$ | $10.72 \pm 0.38$ | $6.26 \pm 0.25$ | +2.76% |
| | $\alpha_{\mathrm{CODAC}} = 1.0$ | $11.59 \pm 0.35$ | $5.92 \pm 0.24$ | $10.81 \pm 0.21$ | $6.26 \pm 0.36$ | +2.80% |
| | $\alpha_{\mathrm{CODAC}} = 5.0$ | $13.40 \pm 1.31$ | $6.40 \pm 0.29$ | $15.59 \pm 3.07$ | $7.04 \pm 0.65$ | +23.59% |

The mean-advantage term is therefore computed separately for every quantile fraction and is shared across the available actions at that fraction. As in QR-DQN (Dabney et al., 2018), the network does not explicitly enforce the ordering

$$\theta^1(s,a) \le \theta^2(s,a) \le \cdots \le \theta^N(s,a). \tag{19}$$

The quantile Huber loss provides an ordering pressure during training, but occasional crossings can occur both with and without the dueling decomposition.

Across the evaluated state–action pairs, 74% contain at least one crossing between adjacent quantile fractions. However, these crossings are generally small. For each predicted distribution, we measure the largest adjacent crossing relative to its total predicted return spread. The average maximum crossing is 0.8% of the distribution's spread, while its 99th percentile is 5.0%. Thus, crossings are frequent in a binary sense but typically negligible in magnitude.

Quantile crossing does not affect the expected value used by the actor or the CQL penalty. The critic represents a uniform mixture of Dirac atoms, and both its empirical distribution and its mean are invariant to permutations of these atoms. We verify this through a monotone-rearrangement experiment in which the predicted atoms are sorted before action selection. Across 5,000 evaluated greedy decisions, rearrangement changes only one selected action. This change results from a floating-point tie between actions whose expected values differ by approximately $6 \times 10^{-8}$, and it does not change the resulting makespan. A monotonic projection layer is therefore unnecessary for the risk-neutral objective considered here. Such a layer may nevertheless be useful in future work if individual quantiles are used directly for risk-sensitive policy optimization.

### K.3 Comparison with distributional conservatism

CODAC (Ma et al., 2021) regularizes the complete predicted return distribution to support risk-sensitive offline control. Directly applying CODAC as published is not possible in our setting because it is formulated for fixed-dimensional continuous action spaces. We therefore isolate its conservative mechanism by replacing the mean-value CQL penalty in CDQAC with CODAC's distributional conservative regression term while retaining the same encoder, actor, quantile critic, dataset, and training procedure. This experiment should consequently be interpreted as a mechanism-level comparison rather than as a full reimplementation of CODAC.

Table 23 shows that regularizing the complete return distribution does not improve performance under the risk-neutral makespan objective. The strongest CODAC configuration, $\alpha_{\mathrm{CODAC}} = 0.05$, increases the average gap by 1.59% relative to standard CDQAC. The remaining moderate penalty strengths produce increases between 2.00% and 5.30%, while the strongest penalty, $\alpha_{\mathrm{CODAC}} = 5.0$, substantially degrades both performance and training stability, particularly on the out-of-distribution benchmarks.

These findings support applying conservatism to the expected return in CDQAC. The actor selects actions according to this expectation, and the empirical diagnostics show that mean and upper-tail rankings are

already strongly aligned. Penalizing the complete distribution introduces an additional constraint that is useful when optimizing tail-sensitive objectives, but in the present risk-neutral setting it can unnecessarily restrict the propagation and stitching of high-value transitions. Nevertheless, this conclusion is specific to mean-return optimization; risk-sensitive scheduling objectives may benefit from full-distribution conservatism.

## L Practical Deployment and Limitations

**Intended deployment setting.** CDQAC is intended for repeated static JSP and FJSP settings in which job routes, machine eligibility, and processing times are available before schedule construction. In such a setting, training and deployment can be separated: a transition dataset is collected once from historical schedules or generated using an available scheduling model, CDQAC is trained offline, and the resulting policy is subsequently used as a constructive dispatching rule. Importantly, we do not advocate executing a uniformly random policy in a live production system. In our experiments, random trajectories are generated in an offline environment. In practice, broad-coverage data could instead be obtained from heterogeneous historical policies, existing dispatching rules, or controlled mixtures of heuristic and exploratory decisions in a validated digital environment.

Offline learning removes the need for environment interaction during policy optimization, but it does not remove all deployment engineering. The logged data must contain sufficient information to reconstruct the state, feasible machine–operation pairs, transition, and objective-aligned reward. When such logs are unavailable, a scheduling model is still required to generate the training dataset. The principal practical advantage is therefore that this model is not queried repeatedly by an evolving on-policy learner and that existing static operational data can be reused.

**Data-generation and training cost.** Generating the broad-coverage dataset is inexpensive relative to generating high-quality heuristic demonstrations. In our implementation, generating 100 random trajectories requires approximately 0.05 seconds per instance, whereas running the GA requires approximately 233–284 seconds per benchmark instance (Table 4). Moreover, the sample-efficiency results in Fig. 5 and App. I.4 show that competitive performance is already obtained from approximately 10–25 training instances in most settings, rather than the full set of 500 instances. Reducing either the number of instances or the number of trajectories per instance generally has only a limited effect on performance, although larger datasets can reduce variance and improve out-of-distribution generalization.

All reported models were trained on a single NVIDIA A100 GPU with an Intel Xeon CPU. A complete run of 200,000 gradient steps takes approximately one to two hours. The training curves in App. H indicate that CDQAC typically converges after approximately 10,000 steps for $10 \times 5$ instances and 25,000 steps for $15 \times 10$ instances; we retain 200,000 steps in the experiments to use a common conservative training budget across methods and datasets. Thus, early stopping using a held-out validation set can substantially reduce the practical training cost. This cost is also considerably lower than that of several specialized self-supervised JSP approaches (Corsini et al., 2024; Pirnay & Grimm, 2024), which report up to seven GPU-days of training, although those approaches target a different performance–cost trade-off.

**Inference cost and solution quality.** CDQAC supports two deployment modes. Greedy decoding produces one schedule and is appropriate when latency is the primary concern. On the standard FJSP benchmark sets, greedy inference takes approximately 1.1–1.2 seconds per instance, and between 0.39 and 1.56 seconds on the generated $10 \times 5$–$20 \times 10$ instances (Table 5). Sampling evaluates 100 independently generated schedules and retains the best one, improving solution quality at an increased cost of approximately 3–5 seconds on these instance sizes. On the larger unseen $30 \times 10$ and $40 \times 10$ instances, greedy inference takes 2.32 and 3.19 seconds, while sampling takes 9.57 and 16.01 seconds, respectively (Table 6). The independent sampling rollouts may also be batched or parallelized when computational resources are available.

These runtimes are substantially below those of the evaluated GA and exact solvers. Nevertheless, CDQAC does not provide an optimality certificate and should not be interpreted as a replacement for exact optimization in every regime. CP remains preferable for small instances when optimality is required and sufficient solving time is available. CDQAC is most relevant for repeated or larger scheduling problems in which a good

solution must be produced under a limited per-instance computational budget. A practical system could also use CDQAC to provide an initial solution or warm start for a subsequent exact or improvement-based solver.

**Scope of the random-data finding.** The conclusion that broad random data can be preferable to higher-quality heuristic data is conditional rather than universal. It relies on two properties of the studied constructive scheduling formulation: (i) the dense reward is exactly aligned with the makespan objective, and (ii) all behavioral policies generate complete trajectories of equal length. Table 3 directly tests these conditions. Early termination breaks the equal-length property, while proxy and sparse rewards weaken objective alignment; in each case, the advantage of the Random dataset shrinks or reverses, with Random often degrading most strongly. Thus, the result should be understood as identifying a particular offline-RL regime in which coverage becomes the binding constraint, rather than as a general recommendation to prefer random data in arbitrary domains.

Several important industrial scheduling settings violate these assumptions. Stochastic processing times, dynamic job arrivals, machine breakdowns, cancellations, and partial observability may produce variable-length trajectories and transitions whose consequences cannot be inferred from the logged state. Similarly, objectives involving tardiness, energy consumption, setup costs, service levels, or multiple competing criteria may not admit the dense makespan-aligned reward used here. In these settings, broad random trajectories may contain insufficient information about desirable behavior, and the relative value of expert or targeted exploratory data may increase. Extending CDQAC to these setups requires additional future research and explicit evaluation in dynamic or stochastic MDPs or POMDPs and, potentially, uncertainty-aware or risk-sensitive policy objectives.

**Modeling and generalization limitations.** The current formulation assumes that all hard scheduling constraints are represented by the feasibility mask. The learned policy therefore preserves feasibility with respect to the encoded JSP/FJSP model, but it cannot account for omitted operational constraints such as sequence-dependent setup times, planned machine maintenance, material availability, batching, workforce constraints, or intentional machine idling. Errors in the feasibility mask or state reconstruction can likewise lead to schedules that are valid in the model but invalid in the physical system.

The quantile critic models variation in predicted returns, but the deployed actor optimizes the mean return and is therefore risk-neutral. The method does not automatically provide robustness to rare breakdowns or guarantees on worst-case makespan. Risk-sensitive deployment would require optimizing an appropriate tail functional or robust objective rather than merely retaining the current quantile representation.

Finally, although CDQAC generalizes across benchmark families, processing-time distributions, and instance sizes, the experiments do not constitute validation on a live industrial system. Direct training also becomes less stable as the variable action space grows; the results show that training on smaller instances and generalizing to larger ones can be more reliable than training directly at the largest tested size. Performance beyond the tested distributions and sizes is therefore not guaranteed. A conservative deployment should evaluate the policy on held-out data from the target system, measure dataset support around the deployed policy, compare against existing dispatching and optimization baselines, retain a feasible fallback policy, and monitor changes in job mix, processing times, and machine availability that may require retraining.

