# OpenReview forum: "Generalizing Beyond Suboptimality: Offline Reinforcement Learning Learns Effective Scheduling through Random Solutions"
_TMLR — Under review for TMLR_

### Review · Reviewer_G6b5 · 2026-06-22

**Summary Of Contributions:**

## Summary and Strengths

The paper introduces Conservative Discrete Quantile Actor-Critic (CDQAC), an offline reinforcement learning framework explicitly tailored to solve combinatorial optimization problems. The primary core contribution is demonstrating that offline RL can successfully bypass the massive sample-efficiency bottleneck of online RL approaches by learning from static, entirely suboptimal datasets. The paper identifies two fundamental structural properties unique to scheduling domains that break the traditional offline RL data-quality paradigm: dense, objective-aligned rewards (derived from partial makespan increases) and equal-length constructive trajectories regardless of the generating behavior policy. Leveraging these properties, the authors establish that state-action coverage rather than trajectory quality is the critical constraint for offline scheduling performance. Consequently, they show that a simple random policy serves as the ideal data generator by maximizing state-action coverage. Mechanistically, CDQAC stabilizes offline learning over discrete, variable action spaces by combining a two-head quantile critic with a mean-regularized Conservative Q-Learning (CQL) penalty, an innovative separated-input dueling network architecture, and delayed policy updates.


## Weaknesses

- The paper applies the Conservative Q-Learning (CQL) penalty strictly to the scalar expectation (mean) of the estimated quantile distribution rather than regularizing the full distribution. While the authors justify this choice through their risk-neutral makespan objective, this creates a structural vulnerability during bootstrapping. In complex scheduling landscapes, out-of-distribution (OOD) or sub-optimal actions can display highly skewed or multi-modal return distributions; an action might have an artificially depressed mean value yet still possess highly optimistic upper-tail quantiles. Because the target generation relies on distributional Bellman backups, these unpenalized tail overestimations can propagate back into the network, leading to policy instability in large combinatorial spaces.
- The empirical evaluation positions CDQAC against standard heuristic rules (PDRs, basic GA) and online deep RL baselines (such as DANIEL and FJSP-DRL). However, the practical viability of any machine learning approach to JSSP/FJSP must be contextualized against production-grade Operations Research (OR) methods. The paper lacks comparison against exact modern solvers (e.g., Google OR-Tools CP-SAT) or highly optimized meta-heuristics (such as advanced Tabu Search or hybrid Genetic Algorithms). Without these baselines, it is difficult to determine whether the policy's final makespan minimization is truly competitive for industrial deployment or merely superior to sub-optimal baseline models.
- The foundational claim that random datasets yield optimal training properties depends heavily on the assumption of equal-length trajectories and highly dense, clean reward metrics. While this paradigm holds true for static, deterministic JSSP/FJSP instances where every operation must be scheduled exactly once, it is poorly matched to realistic industrial settings. In dynamic environments featuring stochastic machine breakdowns, variable job arrival times, or flexible routing alternatives, trajectories vary in length, and rewards often include sparse tardiness penalties. Under such conditions, a random policy would generate chaotic, failure-prone behaviors, invalidating the coverage-centric assumption.
- This paper lacks some additional baselines on offline RL, e.g., traditional offline RL algorithms (besides BC and IQL) like MCQ [1], offline RL with distributional critics, CODAC [2]

## Questions

- Equation 6 details subtracting the mean advantage across available actions to ensure identifiability. Given that you are predicting a mixture of $N$ Dirac deltas for the quantile distribution, does subtracting a scalar mean advantage preserve valid quantile properties (e.g., non-decreasing quantile values across fractions $\tau_n$) across the final $Z_\theta$ output?
- For exceptionally large scheduling instances (e.g., $40 \times 10$ or larger), does the global graph embedding $h_G$ suffer from over-smoothing? Have you evaluated whether performance degrades when $V_\theta$ lacks access to explicitly aggregated bottleneck indicators, such as the variance of machine workloads?
- Given that random datasets contain a massive volume of highly sub-optimal transitions, how sensitive is the policy's convergence to the hyperparameter $\alpha_{CQL}$? Does a single choice of $\alpha_{CQL}$ generalize across both JSP and FJSP benchmarks, or does it require instance-specific calibration?

## References

[1] Mildly conservative q-learning for offline reinforcement learning. NeurIPS

[2] Conservative offline distributional reinforcement learning. NeurIPS

**Audience:**

Yes

**Audience Explanation:**

The advances in solving combinatorial optimization problems are important and match the scope of this venue

**Claims And Evidence:**

Yes

**Claims Explanation:**

The claims are mostly supported by the extensive experimental results

**Requested Changes:**

Please see the weaknesses above and also see the requests below

- Add an overall figure: Add an overview figure of the entire algorithm/system in the introduction or the method part to better illustrate the entire contents of the proposed method
- Incorporate Additional Baselines: Add a comparative column in your benchmark evaluation tables (Tables 5, 12, 13) showing the performance of a standard commercial or open-source constraint programming solver given a reasonable timeout to demonstrate the method's competitive standing against non-RL optimization paradigms. Also, including TD-based offline RL baselines like MCQ, and offline RL algorithms with distributional critics (e.g., CODAC) as baselines would strengthen this paper.
- Hyperparameter Sensitivity Ablations: Provide an explicit hyperparameter sensitivity analysis for the CQL coefficient $\alpha_{CQL}$ and the delayed policy update frequency $\eta$. Show how variations in these parameters impact the final makespan gap across a subset of Brandimarte or Hurink benchmarks.
- Mathematical Clarification on Distributional Dueling: Clarify the text in Section 4 to explicitly explain how the mean-subtraction operation over the advantage stream affects individual quantile distributions, and verify whether a monotonic projection layer is required to prevent quantile crossing after the dueling recombination.
- Discussion on Dynamic Environments: Expand the conclusion or limitations section to include a formal discussion regarding how performance and data requirements are expected to change when moving from static, constructive scheduling settings to dynamic, stochastic environments where trajectory lengths vary.
- Visualization of Quantile Outputs: Provide a visualization plot of the predicted return distributions ($Z_\theta$) for both a highly optimal machine-operation choice and a clearly sub-optimal choice to empirically demonstrate that the quantile critic successfully captures the multi-modality of the scheduling process.

---

> ### Author Response · Authors · 2026-07-17
> **Response to reviewer G6b5 (Part 1/2)**
>
> We thank the reviewer for the positive overall assessment and detailed technical questions. We added the requested end-to-end overview as Fig. 1 and the requested return-distribution visualization as Fig. 10.
>
> ### W1. Mean-CQL and optimistic upper-tail quantiles
>
> Applying CQL to the mean is deliberate because both the actor and the conservative objective optimize $Q_\theta^Z(s,a)=\mathbb E[Z_\theta(s,a)]$. Any net optimism that raises the score used for policy improvement therefore also raises the penalized mean. In the Bellman target, CDQAC compares the means of its two critic heads and retains the complete quantile vector from the lower-mean head, rather than minimizing individual quantiles. This does not theoretically preclude a low-mean distribution with an optimistic upper tail, so we evaluated the concern directly.
>
> Across more than 270,000 reachable state-action pairs and 1.3 million within-state action comparisons (Appendix K.1), the mean and 0.95-quantile rankings have Spearman $\rho=0.997$. A lower-mean action has a higher 0.95-quantile in only 7.2% of comparisons, and the maximum-mean and maximum-upper-tail actions coincide in 89.7% of states. The problematic pattern is therefore uncommon in the trained critic.
>
> We also replaced mean-CQL with CODAC's full-distribution penalty while holding the rest of CDQAC fixed (Appendix K.3/Table 23). All seven tested strengths $\alpha_{\mathrm{CODAC}}\in[0.025,5.0]$ underperform mean-CQL. The best, $\alpha_{\mathrm{CODAC}}=0.05$, increases the average gap by 1.59%; $\alpha_{\mathrm{CODAC}}=5.0$ increases it by 23.59% and raises variance, especially OOD. This is a mechanism-level comparison, not a full reimplementation of continuous-action CODAC. It supports mean conservatism for the present risk-neutral objective, while leaving open that risk-sensitive scheduling may benefit from full-distribution regularization.
>
> **CDQAC with CODAC's distributional penalty in place of mean-CQL.** Average gap increase across greedy/sampling and in-/out-of-distribution evaluations, relative to standard CDQAC with $\alpha_{\mathrm{CQL}}=0.05$:
>
> | | $\alpha_{\mathrm{CODAC}}=0.025$ | $0.05$ | $0.1$ | $0.25$ | $0.5$ | $1.0$ | $5.0$ |
> |:--|--:|--:|--:|--:|--:|--:|--:|
> | Average $\Delta$ gap (%) | +5.30 | +1.59 | +2.00 | +3.83 | +2.76 | +2.80 | +23.59 |
>
> ### W2. Exact solvers and optimized metaheuristics
>
> These comparisons were already present but are now signposted more clearly. OR-Tools CP-SAT with a 30-minute limit is the CP row in Table 4 and Table 7; MIP is also in Table 7; and the hybrid two-stage GA 2SGA is in Table 4. CP-SAT additionally supplies the reference solutions for generated instances, so a separate CP entry for the generated-instance columns in Tables 5 and 14-16 would be approximately zero by construction. Benchmark gaps use literature best-known solutions, which incorporate strong historical metaheuristics.
>
> We do not claim that CDQAC replaces exact optimization on small instances. The relevant result is favorable scaling under a fixed budget: on Taillard $100\times20$, CDQAC sampling reaches a 3.6% gap, versus 3.9% for CP and 11.0% for MIP (Table 7). Appendix L now states this deployment trade-off explicitly.
>
> ### W3. Dynamic and stochastic environments
>
> We agree that the random-data conclusion should not be extrapolated to these settings. Table 3 shows that breaking equal-length trajectories through early termination or weakening reward alignment removes much of the Random advantage. These variants constitute a different problem class from the static, deterministic JSP/FJSP formulation addressed here — the setting targeted by every learning-based baseline and benchmark suite in our comparison — and are therefore outside the scope of this paper. Appendix L discusses dynamic arrivals, stochastic durations, breakdowns, partial observability, variable horizons, and alternative objectives, and identifies them explicitly as future work requiring separate empirical validation.

---

> ### Author Response · Authors · 2026-07-17
> **Response to reviewer G6b5 (Part 2/2)**
>
> ### W4. MCQ, CODAC, and other offline RL baselines
>
> MCQ (Lyu et al., 2022) and CODAC (Ma et al., 2021) are formulated and evaluated for fixed-dimensional continuous actions and are not directly applicable as published to variable-size categorical scheduling actions. IQL is compatible after adapting the policy/action representation and is included with a shared encoder, alongside BC and two Offline-LD variants. To address the distributional-conservatism question without conflating it with a new discrete CODAC implementation, Appendix K.3 isolates CODAC's penalty within CDQAC; the results are summarized under W1.
>
> MCQ's defining mechanism does not admit the same isolation. MCQ constructs pseudo Bellman targets for OOD actions from the maximum critic value over actions sampled from a learned behavior model — in practice a CVAE that reconstructs continuous action vectors (Lyu et al., 2022). Scheduling actions are categorical selections from an instance- and state-dependent feasible set, so there is no fixed-dimensional action vector to reconstruct. The faithful discrete substitute for the CVAE is a categorical model over feasible machine–operation pairs, i.e., exactly our BC baseline, and the remaining design choices (sample count, loss weighting) would be ours rather than MCQ's — a negative result would therefore reflect our adaptation, not the method. If another recent offline-RL method can be transferred to this variable-size discrete setting without changing its defining mechanism, we welcome a specific suggestion.
>
>
> ### Q1. Dueling recombination and quantile crossing
>
> Equation 6 is applied separately at each quantile fraction:
>
> $$\theta^n(s,a)=V^n(h_G)+A^n(s,a)-\frac{1}{|A(s)|}\sum_{a'\in A(s)}A^n(s,a').$$
>
> As in QR-DQN (Dabney et al., 2018), monotonicity across fractions is not enforced; the quantile Huber loss supplies ordering pressure, and crossing can occur with or without dueling recombination. Appendix K.2 quantifies it: 74% of evaluated pairs contain at least one adjacent crossing, but the average largest crossing is only 0.8% of the distribution's spread (99th percentile: 5.0%). Sorting the atoms before greedy action selection changes only one of 5,000 decisions, caused by a $6\times10^{-8}$ mean-value tie, and changes no makespan. Because the actor and CQL use the permutation-invariant mean, a monotonic projection is unnecessary for the present risk-neutral objective; it may be useful for future policies that optimize individual quantiles.
>
> ### Q2. Over-smoothing of $h_G$ and bottleneck indicators
>
> DAN has only two attention layers, limiting depth-related over-smoothing. More importantly, $h_G$ feeds only the value stream: operation-, machine-, and pair-specific embeddings reach the advantage stream directly, so action-specific bottleneck information is not compressed solely through global mean pooling. Machine features already include unscheduled-operation counts, free/waiting times, and remaining processing time (Table 8).
>
> The large-instance results do not show a degradation consistent with over-smoothing. When trained at $10\times5$, CDQAC outperforms the learned baselines at $30\times10$ and $40\times10$ in both the single-seed comparison (Table 6) and the four-seed comparison (Table 22).
>
> ### Q3. Sensitivity to $\alpha_{\mathrm{CQL}}$ and $\eta$
>
> All JSP and FJSP results use one configuration, $\alpha_{\mathrm{CQL}}=0.05$, $\eta=4$, and $N=64$, without benchmark-specific calibration (Table 11). Appendix I.1/Table 13 evaluates both in- and out-of-distribution performance. The $\eta$ sweep is U-shaped around the default: $\eta=1$ is worst (+11.72%), while $\eta=2,3,6,8$ yield +7.19%, +3.78%, +5.45%, and +6.87%, respectively. CDQAC is less sensitive to the tested CQL strengths: $\alpha_{\mathrm{CQL}}=0$ gives +1.93% and $0.1$ gives +3.19% relative to the default. This agrees with the ablation showing that delayed policy updates become increasingly important as the action set grows.
>
> ---
> #### References
> - Lyu, J., Ma, X., Li, X., & Lu, Z. (2022). Mildly conservative q-learning for offline reinforcement learning. Advances in Neural Information Processing Systems, 35, 1711-1724.
>
> - Ma, Y., Jayaraman, D., & Bastani, O. (2021). Conservative offline distributional reinforcement learning. Advances in neural information processing systems, 34, 19235-19247.
>
> - Dabney, W., Rowland, M., Bellemare, M., & Munos, R. (2018). Distributional reinforcement learning with quantile regression. In Proceedings of the AAAI conference on artificial intelligence (Vol. 32, No. 1).

---

> > ### Comment · Reviewer_G6b5 · 2026-07-21
> >
> > Thank you for the rebuttal and the revision, which address most of my concerns and questions. Here is some follow-up:
> > - the overview figure of the entire algorithm/system (Figure 1) can be improved. Figure 1 is a bit simple. Perhaps the authors can take some CV papers as examples for the overall figure
> > - concerning on the baseline methods: It should not be difficult to transform the continuous RL algorithm to fit the discrete domain (e.g., Eq 18 in [1]). What I want to say here is that the baselines are insufficient and the authors should include more offline RL baselines. It would be great if the authors can add those results
> >
> > [1] Towards General-Purpose Model-Free Reinforcement Learning. ICLR

---

> ### Author Response · Authors · 2026-07-22
>
> Dear Reviewer G6b5,
>
> Thank you for the follow-up. We are pleased to read that we addressed most concerns. Below, we address the two remaining concerns.
>
> ## 1. Overview figure (Figure 1)
>
>  We would be happy to further improve Figure 1, but we would like to ensure that the revision addresses your intended concern. Figure 1 currently provides a high-level end-to-end overview, offline data generation, CDQAC training, and schedule construction, while Figure 2 represents overestimation of OOD scheduling actions and Figure 3 presents the detailed network architecture. When describing Figure 1 as “a bit simple,” do you primarily mean that (i) its visual presentation should be more illustrative and polished or (ii) it should contain more algorithmic detail? For the latter, would you prefer us to show the internal CDQAC training mechanisms or provide a more detailed illustration of the scheduling problem and constructed solution? If possible, a representative figure or CV paper would also help us understand the level of detail and presentation style you have in mind.
>
> ## 2. Additional offline-RL baselines
>
> Thank you for the suggestion, and for clarifying the point about Eq. 18. Following it, we adapted **TD3-AWR** (Jackson et al., 2025) to our discrete, variable-size action space. In the evaluation of Jackson et al., TD3-AWR is the strongest offline-RL algorithm: it outperforms both of its source methods, ReBRAC and IQL, themselves the two strongest prior baselines in that study.
>
> The results (Table 1, average gap over the three training sizes; lower is better) show that **CDQAC outperforms TD3-AWR on every dataset and in both evaluation modes**, frequently by a large margin. TD3-AWR is nonetheless a strong baseline. It is the best-performing baseline in the majority of the results, which makes this a meaningful comparison: CDQAC improves over a recent, competitive offline-RL method.
>
> **Table 1.** Average gap (%) over all training sizes (10×5, 15×10, 20×10); lower is better. Top block: greedy evaluation; bottom block: sampling evaluation. **Bold** = best per column.
>
> | Evaluation | Method | PDR | GA | PDR-GA | Random |
> |---|---|---:|---:|---:|---:|
> | Greedy | BC | 29.13 ± 3.20 | 13.91 ± 0.60 | 22.37 ± 2.24 | 21.85 ± 2.51 |
> |  | Offline-LD (mQRDQN) | 22.26 ± 2.43 | 30.85 ± 3.57 | 21.80 ± 3.64 | 21.49 ± 2.62 |
> |  | Offline-LD (d-mSAC) | 23.28 ± 3.06 | 21.02 ± 2.13 | 25.94 ± 2.29 | 16.91 ± 1.89 |
> |  | IQL | 19.93 ± 1.83 | 20.66 ± 2.18 | 19.24 ± 2.34 | 21.34 ± 3.54 |
> |  | TD3-AWR | 18.63 ± 2.58 | 19.91 ± 1.08 | 15.29 ± 1.44 | 14.72 ± 0.50 |
> |  | **CDQAC (Ours)** | **12.34 ± 1.72** | **13.06 ± 2.10** | **11.31 ± 1.33** | **10.68 ± 0.51** |
> | Sampling | BC | 10.71 ± 0.99 | 8.30 ± 0.15 | 9.49 ± 0.56 | 13.15 ± 0.09 |
> |  | Offline-LD (mQRDQN) | 13.64 ± 0.20 | 14.26 ± 0.26 | 13.68 ± 0.17 | 13.63 ± 0.23 |
> |  | Offline-LD (d-mSAC) | 11.61 ± 1.32 | 8.83 ± 0.69 | 11.69 ± 1.23 | 7.79 ± 0.86 |
> |  | IQL | 10.01 ± 0.58 | 9.19 ± 0.56 | 9.48 ± 0.59 | 10.79 ± 0.74 |
> |  | TD3-AWR | 8.86 ± 0.52 | 8.87 ± 0.34 | 7.86 ± 0.27 | 8.93 ± 0.16 |
> |  | **CDQAC (Ours)** | **6.57 ± 0.76** | **6.43 ± 0.87** | **5.87 ± 0.51** | **5.86 ± 0.30** |
>
>
> We added TD3-AWR to Table 1 and its implementation details to Appendix E.
>
> On the two further offline-RL methods raised in your original review (MCQ and CODAC): the distributional-critic mechanism of **CODAC** is already compared at the mechanism level in Appendix K.3, where we transplant its full-distribution conservative penalty into CDQAC with the encoder, actor, critic, data, and training protocol held fixed. **MCQ** is a different case. Its defining mechanism is a CVAE that generates candidate actions as continuous vectors, which MCQ uses to build a max-value bootstrap target for out-of-distribution actions (Lyu et al., 2022). Our actions are categorical selections from an instance- and state-dependent feasible set, so there is no fixed-dimensional action vector for a CVAE to reconstruct; a faithful transfer would require replacing the CVAE with a categorical generative model of our own design, and any negative result would then reflect our adaptation rather than MCQ itself. We therefore prioritized the comparisons we can make faithfully: TD3-AWR and CODAC's penalty in Appendix K.3. With TD3-AWR added, our offline-RL comparison now includes the best-performing model-free method in the current standardized benchmark, alongside IQL, BC, and both Offline-LD variants; we are not aware of a stronger general offline-RL method that transfers to our variable-size discrete action space without changing its defining mechanism.
>
> ### References
>
> - Fujimoto et al. (2025). Towards General-Purpose Model-Free Reinforcement Learning. ICLR
> - Jackson et al. (2025). A Clean Slate for Offline Reinforcement Learning. NeurIPS
> - Lyu et al . (2022). Mildly Conservative Q-Learning for Offline Reinforcement Learning. NeurIPS

---

> > ### Comment · Reviewer_G6b5 · 2026-07-22
> >
> > Thank you for the additional results. The results of TD3-AWR is a nice addition to the paper. Regarding Figure 1, it should contain more technical details (e.g., training mechanisms). As some random examples, the authors can take the illustration figures in the following papers as references:
> >
> > [1] Mst++: Multi-stage spectral-wise transformer for efficient spectral reconstruction
> >
> > [2] On the integration of self-attention and convolution
> >
> > [3] Learning continual compatible representation for re-indexing free lifelong person re-identification
> >
> > [4] UniRain: Unified Image Deraining with RAG-based Dataset Distillation and Multi-objective Reweighted Optimization

---

### Review · Reviewer_FhPp · 2026-06-26

**Summary Of Contributions:**

The paper studies offline RL for Job Shop Scheduling (JSP) and Flexible Job Shop Scheduling (FJSP). The authors propose Conservative Discrete Quantile Actor-Critic (CDQAC), which combines a quantile-based critic, conservative Q-learning regularization, a dueling critic architecture, a dual-attention encoder, and delayed policy updates to learn scheduling policies from static datasets.

The main strength of the paper is that it targets an important and practical setting: learning scheduling policies without online simulator interaction or expert demonstrations. The empirical results are extensive across JSP/FJSP benchmarks, and the paper raises an interesting domain-specific observation about random data and coverage in offline RL.

However, I have several concerns.
* The proposed method appears to be an integration of several existing techniques rather than a fundamentally new algorithmic idea, which raises questions about the degree of novelty and the practical cost of deployment.
* The paper does not clearly characterize the actual state space size or the support structure of the offline dataset, which is important for understanding the claim that random data is beneficial.
* The justification for the quantile critic is somewhat underdeveloped; the paper motivates it intuitively, but does not provide a strong analysis showing why distributional value estimation is necessary in this setting. Typically, quantile-based Q is used for complex state space, like Atari.
* Some statements in the abstract and contribution section seem overstated. For example, since transferring online scheduling RL ideas to offline RL and combining known components is not by itself a new conceptual framework.
* The baseline set appears somewhat limited and potentially outdated for supporting strong SOTA claims.
* Finally, the conclusion that random data is the best source for offline RL in scheduling should be presented more carefully. This finding conflicts with much of the standard offline RL literature, where random data is usually weak. Although random data may provide broader coverage, it may also lack sufficient support density for reliable critic updates.

**Audience:**

Yes

**Audience Explanation:**

The paper addresses a topic that is likely to interest at least some members of the TMLR audience, particularly researchers working on offline reinforcement learning, combinatorial optimization, and learning-based scheduling. The empirical observation that random or highly diverse suboptimal data may outperform stronger heuristic data in JSP/FJSP is potentially interesting, even if I have concerns about whether the claim is sufficiently justified and how broadly it should be interpreted.

**Claims And Evidence:**

No

**Claims Explanation:**

While the paper provides extensive empirical results, I do not think all major claims are supported by sufficiently accurate, convincing, and clear evidence. In particular, the claim that random data is the best source for offline RL in scheduling requires a more careful analysis of the actual state-action support structure. Broad coverage does not necessarily imply sufficient support density for reliable critic learning. The paper also does not clearly characterize the effective state space size, which is important for evaluating this claim.

In addition, the justification for the quantile critic is mostly intuitive and supported only by relatively modest ablation results. The method combines several existing techniques, so stronger evidence is needed to support the more ambitious claims of novelty and SOTA performance. Some baseline choices also appear limited or potentially outdated, which weakens the strength of the empirical claims.

**Requested Changes:**

1. Critical: Clarify and quantify the effective state space and dataset support.

2. Critical: Distinguish coverage from support density. The paper currently argues that random data is effective because it provides broader state-action coverage. However, broad coverage alone does not imply reliable critic learning if each state-action region is visited too sparsely.

3. Critical: Provide a stronger justification for the quantile critic. The paper does not sufficiently demonstrate why distributional value estimation is necessary here.

4. Critical: Reframe the random-data conclusion more carefully. The claim that random data is the best source for offline RL in scheduling is potentially interesting, but it should be presented as a domain-specific finding rather than a broad offline RL conclusion.

5. Critical: Strengthen the baseline comparison.

6. Critical: Moderate overstated claims in the abstract and contribution section.

7. Critical: Discuss practical deployment cost.

8. Would strengthen: Provide clearer evidence for trajectory stitching.

9. Would strengthen: Discuss limitations more explicitly. The method appears to rely on structural properties of JSP/FJSP: dense rewards, fixed-length constructive trajectories, and feasible-action masking. The authors should discuss whether the approach would still work for scheduling variants with stochastic processing times, dynamic job arrivals, partial observability, machine failures, multi-objective rewards, or early termination.

---

> ### Author Response · Authors · 2026-07-17
> **Response to reviewer FhPp (Part 1/2)**
>
> We thank the reviewer for the detailed questions concerning support, novelty, quantile estimation, deployment, and scope.
>
> ### C1. Effective state space and dataset support
>
> The feasible action set is instance- and state-dependent and contains at most 50, 150, and 200 machine-operation pairs at the three FJSP training sizes. The horizon equals the number of operations. For a fixed instance with $|O|=\sum_j n_j$ operations and eligible-machine sets $\mathcal M_o$, an assignment-sequence representation admits up to
>
> $$
> \frac{|O|!}{\prod_j n_j!}\prod_{o=1}^{|O|}|\mathcal M_o|
> $$
>
> complete candidates before accounting for feasibility. A $10\times5$ JSP instance already has approximately $4.9\times10^{43}$ precedence-respecting operation sequences; if every operation has five eligible machines, the corresponding FJSP count is approximately $4.36\times10^{78}$. The state domain is larger because it also contains reachable partial schedules, and training spans many generated instances rather than one fixed state graph. Tabular support is therefore unattainable; the revision characterizes empirical support through relative combinatorial coverage and local density in a common learned representation (Table 2; Appendix F.1/Table 9). In addition, all the features used can be found in Appendix B.1/Table 8.
>
> ### C2. Coverage versus support density
>
> We now measure these separately. For each instance per training size, we retain the best of 1,000 stochastic CDQAC rollouts as a high-quality reference, embed reference and dataset transitions as $e_t=[h_G(s_t)\Vert h_G(s_{t+1})]$ with the same frozen DAN encoder, normalize the embeddings, and compute the mean distance to the $k=10$ nearest dataset transitions. We treat this as a model-relevant proxy, not a complete characterization of support in the original state space.
>
> PDR-GA is locally densest ($0.029\pm0.009$); Random is only third ($0.031\pm0.009$) but has almost twice PDR-GA's SACo (7.70 versus 3.96). Across the four primary datasets and the $\epsilon$-PDR mixtures, density is not significantly correlated with the gap ($\rho_s=0.578$, $p_s=0.140$), whereas SACo is ($\rho_s=-0.946$, $p_s=0.0006$). Per-size results in Table 9 show the same separation. The $\epsilon$-mixture intervention — already part of the original submission, extended in the revision with TQ and density columns in Table 10 — sharpens it: $\epsilon$-PDR-GA ($\epsilon=0.2$) retains the full GA component and the densest local support in Table 10, yet obtains the best mean gaps at the highest coverage (SACo 8.38), while the configuration with the largest coverage deficit ($\epsilon$-PDR, $\epsilon=0.1$) is the only one that significantly underperforms Random ($p=0.006$).
>
> Appendix I.4/Fig. 9 also reduces Random from 100 to one trajectory per instance, and separately from 500 to five instances. Performance remains broadly stable, with standard deviations mostly below 1.5%. We treat this only as robustness to dataset thinning and effective off-policy reuse, not as a direct density test. The direct density analysis and $\epsilon$-mixture intervention provide the relevant evidence.
>
> ### C3. Justification for the quantile critic
>
> We do not claim that a quantile critic is strictly necessary. Although the environment is deterministic, $Z^\pi(s,a)$ is non-degenerate under a stochastic continuation policy because one machine-operation choice changes the feasible downstream decisions. Fig. 10 shows that the trained critic captures action-dependent distributional separation and unequal variance at one state: the highlighted high- and low-value actions have means $-1.76$ and $-2.53$ and standard deviations 0.14 and 0.22, respectively. Appendix K evaluates more than 270,000 reachable state-action pairs and shows that the learned distributions are unimodal; the motivation is therefore return variation and distributional estimation, not an unsupported multimodality claim.
>
> The ablation positions this component proportionately: removing the quantile critic raises the average gap by 4.2% (Fig. 4; Appendix I.1/Table 12). It is a consistent secondary contributor, whereas the delayed policy update (+54.3% on average) and separated-input dueling critic (+8.9%) are the dominant stabilizers. Sect. 4 has been revised accordingly.
>
> ### C4. Domain-specific framing of the random-data result
>
> We agree. The abstract now says performance “depends more on” coverage, and Contribution 3 reports a significant association rather than a universal or causal rule. Table 3 shows that the Random advantage shrinks or reverses when equal-length trajectories or objective-aligned dense rewards are removed, and Appendix L explicitly limits the finding to the static constructive JSP/FJSP formulation. To make the paper fully consistent with the evidence, we also replaced the remaining “governs,” “confirms the mechanism,” and “binding constraint” wording in Sect. 5.2, the Conclusion, and Appendix L with this association-based formulation.

---

> > ### Author Response · Authors · 2026-07-17
> > **Response to reviewer FhPp (Part 2/2)**
> >
> > ### C5. Baseline comparison
> >
> > The original submission already included CP-SAT, MIP, 2SGA, GA, PDRs, online FJSP/JSP methods, BC, IQL, and both Offline-LD variants. Appendix I.5/Table 19 additionally covers specialized JSP-only constructive, self-supervised, and improvement methods. The revision adds DOAGNN and PRISMA; their aggregate results are given in the general response. Appendix J retrains DANIEL over the same four seeds as CDQAC and retains statistical significance ($p\approx0.03$). Appendix K.3 adds a controlled comparison with CODAC's conservative mechanism. General offline methods based on TD3/SAC continuous actions are not directly applicable as published. Of the strongest methods in the standardized comparison of Jackson et al. (2025), IQL is compatible with our setting and is included.
> >
> > If the reviewer has a specific additional offline-RL method in mind that is compatible with instance-dependent discrete action spaces, we would be grateful for the suggestion and will evaluate it during the discussion period.
> >
> > ### C6. Claims and novelty
> >
> > We moderated the performance and coverage claims as detailed in the general response. We also distinguish the method's specific contributions from its use of established ingredients:
> >
> > - The generic dueling principle is not new. CDQAC's architectural contribution is the **separated-input distributional dueling critic** for variable-size feasible-action sets: $V_\theta$ receives only the global instance representation $h_G$, whereas $A_\theta$ also receives operation-, machine-, and pair-specific representations. Standard dueling networks feed both streams the same state representation.
> > - Delayed actor updates originate in continuous-action online actor-critic learning. To our knowledge, CDQAC is the first use of this mechanism in offline actor-critic learning with instance-dependent discrete action sets. Its role is empirically scale-dependent: removing the delay increases the gap by 11.7%, 38.5%, and 112.8% at $10\times5$, $15\times10$, and $20\times10$; removing the separated-input dueling critic yields 4.3%, 5.8%, and 16.7%.
> >
> > We therefore do not claim novelty from merely combining CQL, quantile regression, dueling networks, and delayed updates. The novelty claim concerns these two adaptations and their demonstrated role in offline variable-action scheduling.
> >
> > ### C7. Practical deployment cost
> >
> > New Appendix L provides the full accounting. Generating 100 Random trajectories takes about 0.05 s per instance, compared with approximately 233-284 s for GA on the reported benchmarks. A 200,000-step run takes about 1-2 h on one A100, while the training curves indicate convergence at roughly 10,000 steps for $10\times5$ and 25,000 for $15\times10$. On standard FJSP benchmarks, greedy inference takes approximately 1.1-1.2 s and 100-sample inference approximately 3-5 s; at $40\times10$, these become 3.19 s and 16.01 s.
> >
> > Offline RL removes the need for environment interaction during training, but not all modeling requirements: usable logs must allow reconstruction of states, feasible actions, transitions, and rewards, and a scheduling model is still required when such logs are unavailable.
> >
> > ### WS8. Evidence for trajectory stitching
> >
> > Table 1 and Appendix F.2 make the behavioral evidence explicit. The $\pi_\beta$ row reports the best performance of the data-generating policy, which is 28.16% for Random, whereas one greedy CDQAC rollout reaches 10.68%. Under sampling, CDQAC also matches or improves on $\pi_\beta$ for every source dataset. This demonstrates improvement beyond complete logged trajectories and is consistent with stitching, although it does not uniquely identify stitching as the mechanism.
> >
> > ### WS9. Limitations
> >
> > Appendix L now discusses the dependence on a dense objective-aligned reward, complete equal-length trajectories, and feasibility masking. The variants the reviewer lists — stochastic processing times, dynamic job arrivals, machine failures, partial observability, multi-objective rewards, and early termination — constitute a different problem class from the static, deterministic JSP/FJSP formulation studied here and by every method in our comparison; Appendix L now places them explicitly outside the scope of this work and identifies them as future research directions. Table 3 provides an empirical preview of why the transfer is nontrivial: early termination and proxy/sparse rewards reduce or reverse the Random advantage.
> >
> > ---
> > #### References
> > - Jackson, M., Berdica, U., Liesen, J., Whiteson, S., & Foerster, J. (2026). A clean slate for offline reinforcement learning. Advances in Neural Information Processing Systems, 38, 161052-161083.

---

> > > ### Comment · Reviewer_FhPp · 2026-07-20
> > >
> > > Thank you for the authors’ detailed response. Most of my previous concerns have been adequately addressed. However, I still have some reservations regarding the random-data result. It is possible that random data is particularly effective for the scheduling tasks considered in this paper, but this may be task-specific rather than a broadly generalizable conclusion. I therefore suggest adding a more in-depth discussion of why random data works well in these environments, including the state-action support structure, effective coverage density, and the potential limitations of this observation.

---

> > > > ### Author Response · Authors · 2026-07-22
> > > >
> > > > Dear Reviewer FhPp,
> > > >
> > > > Thank you for your response. We are pleased that the revision has addressed most of your concerns. We agree that our finding is specific to the JSP/FJSP formulations studied in this work and should not be interpreted as applying to scheduling problems in general. To clarify the scope, we have reviewed the paper and replaced the remaining overly broad references to “scheduling” with language that specifically refers to the JSP/FJSP setting studied here.
> > > >
> > > > The revised manuscript addresses the requested aspects in Section 5.2 and provides a more detailed analysis in Appendix F. Specifically, Section 5.2 quantifies the breadth of the empirical state–action support using SACo, while the reference-policy-anchored $k$-NN distance measures local support density. Appendix F.1 examines these results in greater detail; Appendix F.2 relates these properties to pessimism, intrinsic uncertainty, and trajectory stitching; and Appendix F.3 presents a controlled $\varepsilon$-PDR intervention. Moreover, Table 3 and Appendix L explicitly discuss the conditions and scheduling variants under which our findings may not transfer.

---

### Review · Reviewer_pwcn · 2026-07-09

**Summary Of Contributions:**

This paper proposes CDQAC, an offline reinforcement learning method for job shop scheduling (JSP/FJSP). The main claim is that, unlike conventional offline RL, randomly collected datasets can outperform expert datasets for scheduling because they provide better state-action coverage. The paper evaluates the proposed approach on multiple JSP and FJSP benchmarks, compares it with several online and offline baselines, and includes ablation studies and generalization experiments across different problem sizes.

**Additional Comments:**

Overall, I found the paper interesting and clearly written. The empirical results are promising, and the topic is relevant to the TMLR community. However, the main scientific claim—that randomly collected datasets outperform expert datasets because of better state-action coverage—is stronger than what is currently supported by the experimental evidence. Strengthening the experimental validation of this hypothesis and including comparisons with stronger offline RL baselines would significantly improve the paper.

**Audience:**

Yes

**Audience Explanation:**

The paper addresses an interesting problem at the intersection of offline reinforcement learning and combinatorial optimization. The finding that randomly collected data may outperform expert demonstrations in scheduling is potentially valuable to researchers working on offline RL, neural combinatorial optimization, and learning-based scheduling. Even if some of the empirical evidence could be strengthened, the overall direction is relevant to the TMLR community.

**Broader Impact Concerns:**

I do not identify any significant ethical concerns beyond those generally associated with learning-based scheduling methods. No additional broader impact discussion appears necessary.

**Claims And Evidence:**

No

**Claims Explanation:**

Strengths

1. The paper studies an interesting and relatively underexplored problem: offline reinforcement learning for scheduling without relying on expert demonstrations.

2. The proposed approach achieves competitive performance on several JSP and FJSP benchmarks.

3. The paper is generally well organized and easy to follow.

Weaknesses

1. The empirical evaluation is not sufficiently comprehensive. Although the paper compares against several scheduling-specific methods, it does not include comparisons with stronger or more recent general offline RL approaches. This makes it difficult to fully assess whether the observed gains come from the proposed algorithmic design.

2. The paper's central claim is that randomly collected datasets outperform expert datasets because they provide better state-action coverage. However, this explanation is not directly validated experimentally. The current results demonstrate improved performance but do not isolate or quantify the role of coverage. Additional analyses or controlled experiments would make this conclusion substantially more convincing.

3. While the paper evaluates several standard JSP/FJSP benchmarks, additional experiments on more diverse scheduling settings or distribution shifts would further strengthen the generalization claims.

**Requested Changes:**

Critical

1. Compare against stronger and more recent offline RL baselines to better position the proposed method relative to the current literature.

2. Provide more direct empirical evidence supporting the central hypothesis that improved state-action coverage, rather than other factors, is responsible for the performance gains observed with random datasets.

Would strengthen the paper

1. Include additional evaluations on more diverse scheduling settings or distribution shifts to further support the claimed generalization ability.

2. Provide additional analyses on dataset characteristics (e.g., coverage, diversity, or data quality) to better explain when random datasets are expected to outperform expert datasets.

---

> ### Author Response · Authors · 2026-07-17
> **Response to reviewer pwcn (Part 1/2)**
>
> We thank the reviewer for recognizing the relevance, organization, and empirical promise of the paper.
>
> ### C1. Stronger and more recent offline RL baselines
>
> Our baseline choice follows the standardized re-evaluation of Jackson et al. (2025), which identifies IQL and the TD3-derived ReBRAC as particularly strong general offline-RL baselines. ReBRAC and related TD3 methods are formulated around continuous action vectors, deterministic actors, Gaussian target-policy smoothing, and critics evaluated at perturbed continuous actions; they are therefore not directly applicable as published to the instance-dependent categorical action sets of JSP/FJSP. IQL is adaptable to this setting and is included, together with BC and both Offline-LD variants (mQRDQN and d-mSAC). All use the same DAN encoder, isolating the learning algorithm from the representation.
>
> The revision also strengthens the scheduling side with DOAGNN and PRISMA (Table 4; summary above). For distributional offline RL, Appendix K.3 provides the closest controlled comparison feasible in this action space: CODAC's distributional conservative term is transplanted into CDQAC while holding the encoder, actor, critic, data, and training protocol fixed. It does not improve on mean-CQL at any tested strength (Table 23). MCQ and CODAC themselves are continuous-action algorithms as published and would require a new discrete, variable-action formulation rather than a direct reimplementation.
>
> If the reviewer believes that we have overlooked a recent offline-RL method that is applicable to variable-size discrete action spaces, we would appreciate a specific suggestion.
>
> ### C2. Direct evidence for the role of state-action coverage
>
> We agree that this is the central empirical question, which we address in three complementary ways.
>
> **Measurement (Sect. 5.2; Tables 1-2).** The size-averaged dataset statistics are:
>
> | Dataset | SACo $\uparrow$ | TQ $\uparrow$ | $k$-NN distance $\downarrow$ |
> |:--|--:|--:|--:|
> | PDR | $1.00\pm0.00$ | $0.58\pm0.02$ | $0.033\pm0.011$ |
> | GA | $2.96\pm0.49$ | **$0.89\pm0.01$** | $0.030\pm0.009$ |
> | PDR-GA | $3.96\pm0.49$ | $0.81\pm0.01$ | **$0.029\pm0.009$** |
> | Random | **$7.70\pm0.77$** | $0.58\pm0.02$ | $0.031\pm0.009$ |
> | Spearman $\rho_s$ with gap | **$-0.946$** | $0.157$ | $0.578$ |
> | $p_s$-value | **$0.0006$** | $0.711$ | $0.140$ |
>
> The correlations jointly use the four primary datasets and the $\epsilon$-PDR mixtures. Random has the broadest coverage but is only the third-densest dataset around the high-quality reference trajectories; GA has the highest TQ; and PDR-GA is densest. Only SACo is significantly associated with the gap.
>
> **Controlled supporting intervention (Appendix F.3; Table 10).** This intervention was already part of the original submission; the revision extends Table 10 with TQ and $k$-NN density columns and folds it into the correlation analysis above. Each $\epsilon$-PDR trajectory takes a uniformly random feasible action with probability $\epsilon$ and otherwise follows a PDR, preserving substantial heuristic structure while changing coverage. The largest coverage deficit relative to Random occurs for $\epsilon$-PDR with $\epsilon=0.1$ (SACo 6.82 versus 7.70), which is also the only configuration that significantly underperforms Random ($p=0.006$). Conversely, $\epsilon$-PDR-GA with $\epsilon=0.2$ has the highest SACo (8.38) and the best mean gaps (10.45% greedy and 5.81% sampling), despite retaining all GA trajectories. Because the mixtures also differ in other properties, we interpret this as controlled supporting evidence, not proof that coverage is the sole cause.
>
> **Boundary conditions (Table 3).** We separately break the two proposed preconditions: early termination creates unequal trajectory lengths, while proxy and sparse rewards weaken objective alignment. In all three cases, the Random advantage shrinks or reverses and Random generally degrades most. Thus, the result is conditional on the studied constructive scheduling structure.
>
> Together, these analyses establish a significant association, provide an intervention consistent with it, and identify regimes in which it should not be expected. We do not claim causal identification.

---

> > ### Author Response · Authors · 2026-07-17
> > **Response to reviewer pwcn (Part 2/2)**
> >
> > ### WS1. More diverse scheduling settings and distribution shifts
> >
> > The evaluation already covers several substantial shifts: Brandimarte and the three Hurink families differ from the training generator and span different flexibility regimes; Demirkol processing times extend to 200 versus 100 in the training distributions; FJSP generalization is evaluated from $10\times5$ training to $30\times10$ and $40\times10$; and JSP evaluation reaches $100\times20$. Appendix J reports one-sided Wilcoxon tests against DANIEL: $p\approx0.018$ for the original FJSP comparison, $p\approx0.03$ after retraining DANIEL over four seeds, and $p\approx0.00022$ on JSP. Dynamic and stochastic scheduling variants are outside the scope of this work: like every learning-based method in our comparison (L2D, FJSP-DRL, DANIEL, Residual, DOAGNN, PRISMA) and every benchmark suite we evaluate on (Taillard, Demirkol, Brandimarte, Hurink), the paper targets the static, deterministic JSP/FJSP formulation. Appendix L now states this scope explicitly and identifies dynamic, stochastic, and multi-objective settings as future work.
> >
> > ### WS2. Additional dataset characteristics
> >
> > Revised Table 2 and Appendix F.1-3 jointly analyze support breadth (SACo), trajectory quality (TQ), local representation-space support, dataset thinning, pessimism, intrinsic uncertainty, and trajectory stitching. Table 9 gives SACo, TQ, and density at each training size; Table 10 adds TQ and density to the $\epsilon$-mixture experiments that were already part of the original submission. The conclusion is deliberately correlational: among the measured properties, only SACo is significantly associated with CDQAC's gap, with higher SACo consistently associated with lower gaps.
> >
> > ---
> > #### References
> > - Jackson, M., Berdica, U., Liesen, J., Whiteson, S., & Foerster, J. (2026). A clean slate for offline reinforcement learning. Advances in Neural Information Processing Systems, 38, 161052-161083.

---

### Author Response · Authors · 2026-07-17
**General response**

We thank all three reviewers for their careful and constructive feedback. All reviewers considered the paper relevant to at least part of the TMLR audience; the remaining concerns center on the breadth of the evidence, the interpretation of the random-data result, and the positioning of CDQAC. We revised the paper accordingly.

The main changes relative to the original submission are:

1. **Method overview.** We added an end-to-end overview of our approach, including offline data generation, CDQAC training, and schedule construction, in Fig. 1.

2. **Stronger scheduling baselines.** Table 4 now includes **DOAGNN** (Zhao et al., 2025), a recent online PPO method with separate GNNs for the machine-operation and operation-operation graphs, and **PRISMA** (Jesus et al., 2026), a hybrid multi-agent approach using CP-SAT demonstrations and CP-guided PPO training. Below we show the average over the four FJSP benchmarks. CDQAC obtains the best result in five of the eight benchmark-family/evaluation-mode combinations and lower gaps than PRISMA and DOAGNN in six and seven of eight comparisons, respectively. The submission already included OR-Tools CP-SAT, MIP, the hybrid 2SGA metaheuristic, GA, PDRs, and several specialized JSP methods.


| Method | Greedy gap (%) $\downarrow$ | Sampling gap (%) $\downarrow$ |
|:--|--:|--:|
| DOAGNN | 16.00 | 8.63 |
| PRISMA | 10.95 | 6.63 |
| **CDQAC** | **9.94** | **6.15** |



3. **Expanded dataset analysis.** The original Table 2 reported SACo only. Revised Sect. 5.2 and Table 2 add Trajectory Quality (TQ), an embedding-space $k$-NN local-support proxy, and Spearman correlations with CDQAC's gap, computed jointly over the four primary datasets and the $\epsilon$-PDR mixtures. The $\epsilon$-PDR coverage intervention itself (Appendix F.3/Table 10) was already part of the original submission; the revision extends Table 10 with TQ and $k$-NN density columns and integrates it into this correlation analysis. SACo is the only significant correlate ($\rho_s=-0.946$, $p_s=0.0006$); TQ ($\rho_s=0.157$, $p_s=0.711$) and density ($\rho_s=0.578$, $p_s=0.140$) are not significant. The negative SACo coefficient is expected because higher coverage is compared with a lower-is-better gap. Appendix F.1/Table 9 reports all metrics separately at each training size.

4. **Quantile-critic diagnostics.** New Appendix K and Fig. 10 visualize action-conditioned return distributions and analyze more than 270,000 reachable state-action pairs. Appendix K also quantifies mean/upper-tail alignment and quantile crossing and compares mean-CQL against CODAC's full-distribution penalty (Table 23).

5. **Sensitivity, deployment, and limitations.** Appendix I.1/Table 13 sweeps $\eta\in\{1,2,3,4,6,8\}$ and $\alpha_{\mathrm{CQL}}\in\{0,0.05,0.1\}$. New Appendix L reports data-generation, training, and inference costs, discusses omitted constraints, risk sensitivity, and generalization limits, and explicitly places dynamic/stochastic scheduling and multi-objective formulations outside the scope of this work, as future directions.

6. **More precise claims.** The abstract now says “matches or outperforms” rather than “consistently outperforms,” “outperforms recent” rather than “surpasses state-of-the-art,” and “depends more on” rather than “is governed mainly by.” Contribution 3 now reports a significant coverage-performance correlation rather than calling coverage a binding constraint. We also clarify that the methodological contributions are the specific separated-input distributional dueling critic and the adaptation of delayed actor updates to offline learning with variable-size discrete actions; novelty is not claimed merely from combining known components.

These additions expand the main paper slightly from 12 to 13 pages.